# Offline Reinforcement Learning of High-Quality Behaviors Under Robust Style Alignment

## Abstract

We study offline reinforcement learning of style-conditioned policies using explicit style supervision via subtrajectory labeling functions. In this setting, aligning style with high task performance is particularly challenging due to distribution shift and inherent conflicts between style and reward. Existing methods, despite introducing numerous definitions of style, often fail to reconcile these objectives effectively. To address these challenges, we propose a unified definition of behavior style and instantiate it into a practical framework. Building on this, we introduce Style-Conditioned Implicit Q-Learning (SCIQL), which leverages offline goal-conditioned RL techniques, such as hindsight relabeling and value learning, and combine it with a new Gated Advantage Weighted Regression mechanism to efficiently optimize task performance while preserving style alignment. Experiments demonstrate that SCIQL achieves superior performance on both objectives compared to prior offline methods.

## 1 Introduction

A task can often be performed through diverse means and approaches. As such, while the majority of the sequential decision making literature has focused on learning agents that seek to optimize task performance, there has been a growing interest in the development of diverse agents that display a variety of behavioral styles. While many previous works tackled diverse policy learning by relying on online interactions (Nilsson & Cully, 2021; Wu et al., 2023), the widespread availability of pre-recorded diverse behavior data (Hofmann, 2019; Mahmood et al., 2019b; Zhang et al., 2019; Fu et al., 2021; Lee et al., 2024a; Jia et al., 2024; Park et al., 2025) catalyzed much progress in the learning of policies from such data without further environment interactions, allowing the training of high-performing agents in a more sample-efficient, less time-consuming and safer way (Levine et al., 2020). Such methods can be divided into two categories: Imitation Learning (IL) methods (Pomerleau, 1991; Florence et al., 2021b; Chi et al., 2024b) mimic expert trajectories, while offline Reinforcement Learning (RL) methods (Kumar et al., 2020; Kostrikov et al., 2021; Fujimoto & Gu, 2021; Chen et al., 2021; Nair et al., 2021; Garg et al., 2023) target high-performing behaviors based on observed rewards. Although some recent work has focused on diverse policy learning in both offline IL (Zhan et al., 2020; Yang et al., 2024) and offline RL (Mao et al., 2024), several challenges and questions remain in the study and deployment of stylized policies.

**Challenge 1: Style definition.** Literature dealing with style alignment ranges from discrete trajectory labels (Zhan et al., 2020; Yang et al., 2024) to unsupervised clusters (Mao et al., 2024) and continuous latent encodings (Petitbois et al., 2025), with distinct trade-offs: unsupervised definitions are often uncontrollable and hard to interpret, while supervised ones rely on manual labels and incur significant labeling costs. Additionally, since play styles span multiple timescales, attributing each local step to a style is non-trivial and can take part in credit assignment problems. Furthermore, depending on the definition of style, assessing the alignment of an agent's behavior with respect to a target style may be difficult, which complicates alignment measurement and hinders policy controllability. As such, a key challenge is to derive a **general** definition that addresses **interpretability**, **labeling cost**, **alignment measurement**, and **credit assignment**.

**Challenge 2: Addressing distribution shift.** While offline IL and offline RL are known to suffer from distribution shift due to environment stochasticity and compounding errors (Levine et al., 2020), the addition of style conditioning can exacerbate the issue by creating mismatches at inference time between visited states and target styles. For instance, a running policy may trip and fall into an out-of-distribution state-style configuration without the ability to recalibrate. While some previous work addressed this issue (Petitbois et al., 2025), most of them lack mechanisms to perform robust style alignment. Consequently, an open question is how to achieve **robust style alignment** without relying on further environment interactions.

**Challenge 3: Solving task and style misalignment.** Style alignment and task performance are often incompatible. For instance, a crawling policy may not achieve the same speed as a running one. Optimizing conflicting objectives of style alignment and task performance has been explored in offline RL, either by directly seeking compromises between them (Lin et al., 2024a;b; Yuan et al., 2025), or by shifting optimal policies from one objective to the other (Mao et al., 2024), but always at the cost of style alignment. Consequently, ensuring **robust style alignment while optimizing task performance** remains an open problem.

In this paper, we address these challenges through the following contributions: (**1**) We propose a novel **general** view of the stylized policy learning problem as a generalization of the goal-conditioned RL (GCRL) problem (Park et al., 2025) and show that the style alignment corresponds to the optimization of a form of *style occupancy measure* (Dayan, 1993; Touati & Ollivier, 2021; Blier et al., 2021; Eysenbach et al., 2023). (**2**) We instantiate our definition within the supervised data-programming framework (Ratner et al., 2017) by using labeling functions as in Zhan et al. (2020); Yang et al. (2024) but on trajectory windows rather than full trajectories, capturing the multi-timescale nature of styles. This design choice mitigates high **credit assignment** challenges by design. The use of labeling functions also allows users to **quickly** program various **meaningful** style annotations for both training data and evaluation data, making the **alignment measurement** easier at inference. (**3**) We introduce Style-Conditioned-Implicit-Q-Learning (SCIQL), a style-conditioned offline RL algorithm inspired by IQL (Kostrikov et al., 2021) which leverages advantage signals to guide the policy towards the activation of target styles, making efficient use of **style-relabeling** (Petitbois et al., 2025) and trajectory stitching (Char et al., 2022) to allow for **robust style alignment**. (**4**) Making use of the casting of stylized policy learning problem as a RL problem, we introduce the notion of **Gated Advantage Weighted Regression (GAWR)** in the stylized policy learning context by using advantage functions as gates to allow **style-conditioned task performance optimization**. (**5**) We provide diverse clean implementations of stylized RL tasks on which we demonstrate through a set of experiments that our method effectively outperforms previous work on both **style alignment** and **style-conditioned task performance optimization**, along with various ablation studies. We provide links to clean implementations of our algorithms in JAX (Bradbury et al., 2018) along with the datasets in the following project page: `https://sciql-iclr-2026.github.io/`.

## 2 RELATED WORK

**IL and offline RL.** Imitation Learning seeks to learn policies by mimicking expert demonstrations, usually stored as trajectory datasets, and can be grouped into different categories, including Behavior Cloning, classical Inverse RL (IRL), and Apprenticeship / Adversarial IRL. Behavior Cloning (BC) (Pomerleau, 1991) performs supervised regression of actions given states but suffers from compounding errors and distribution shifts (Ross et al., 2011). Classical IRL (Ng & Russell, 2000; Fu et al., 2018; Arora & Doshi, 2020) infers a reward under which the demonstration policy is optimal to optimize it via online RL. It is robust to distribution shifts but requires environment interactions. Apprenticeship / Adversarial IRL (e.g., GAIL (Ho & Ermon, 2016)) learns policies directly via implicit rewards, combining IRL's robustness with BC's direct learning, but typically requires online interactions. On the other hand, offline RL does not assume optimal demonstrations. It uses reward signals to train policies offline and tackles distribution shifts via sequence modeling (Chen et al., 2021), biased BC (Nair et al., 2021; Fujimoto & Gu, 2021), policy conservativeness (Kumar et al., 2020), expectile regression (Kostrikov et al., 2021), or Q-value exponential weighting (Garg et al., 2023). In this work, we leverage offline RL techniques to jointly optimize behavior styles and task performance from reward signals, without assuming demonstration optimality.

**Diverse policy learning.** Capturing diverse behavior from a pre-recorded dataset has been addressed in the literature under various scopes. Several methods aim to capture a demonstration dataset's multimodality at the action level through imitation learning techniques (Florence et al., 2021a; Shafiullah et al., 2022; Pearce et al., 2023; Chi et al., 2024a; Lee et al., 2024b) while other methods aim to learn higher-timescale behavior diversity by learning to capture various behavior styles in both an unsupervised and supervised approach. In the IRL setting, InfoGAIL (Li et al., 2017), Intention-GAN (Hausman et al., 2017) and DiverseGAIL (Wang et al., 2017) aim it dentify various behavior styles from demonstration data and train policies to reconstruct them using IRL techniques. Tirinzoni et al. (2025) aim to learn a forward-backward representation of a state successor measure (Dayan, 1993; Touati & Ollivier, 2021) to learn through IRL a policy optimizing a high variety of rewards with a bias towards a demonstration dataset. In a BC setting, WZBC (Petitbois et al., 2025) learns a latent space of trajectories to employ trajectory-similarity-weighted-regression to improve robustness to compounding errors in trajectory reconstruction. Further, SORL (Mao et al., 2024) learns a set of diverse representative policies through the EM algorithm and enhances them to perform stylized offline RL. In the supervised setting, CTVAE (Zhan et al., 2020) augments trajectory variational auto-encoders with trajectory style labels to perform imitation learning under style calibration, while BCPMI (Yang et al., 2024) performs a behavior cloning regression weighted by mutual information estimates between state-action pairs and style labels. Our method falls into the offline supervised learning category as in CTVAE and BCPMI as we employ supervised style labels to derive style reward signals for our policy to optimize. However, we consider styles defined on subtrajectories unlike CTVAE and BCPMI which consider full trajectory styles, which can create high credit assignment issues for very long trajectories. Additionally, unlike CTVAE, our method is model-free and unlike BCPMI, we use reinforcement learning signals to enhance the robustness of our method to distribution shift and allow for both task performance and style alignment optimization.

**Goal-Conditioned RL.** Goal-Conditioned RL (GCRL) (Kaelbling, 1993; Liu et al., 2022; Park et al., 2025) encompasses methods that learn policies to achieve diverse goals efficiently and reliably. As our style alignment objective consists in visiting state-action pairs of high-probability to contribute to a given style, it shares with GCRL the same challenges of sparse rewards, long-term decision making and trajectory stitching. To address these challenges, Ghosh et al. (2019); Yang et al. (2022) combine imitation learning with Hindsight Experience Replay (HER) (Andrychowicz et al., 2017), while Chebotar et al. (2021); Kostrikov et al. (2021); Park et al. (2024); Canesse et al. (2024); Kobanda et al. (2025) additionally learn goal-conditioned value functions to extract policies using offline RL techniques. Unlike GCRL, which focuses on achieving specific goals, our framework addresses performing RL tasks under stylistic constraints. This can be viewed as a generalization from goal-reaching to executing diverse RL tasks while maintaining stylistic alignment. Specifically, we distinguish between Style-Conditioned RL (SCRL), the problem of reaching state–action pairs with high style alignment, and Style-Conditioned Task Performance Optimization (SCTPO), which involves performing a task under style alignment constraints.

## 3 Preliminaries

**Markov decision process.** In this work, we consider a $\gamma$-discounted Markov Decision Process (MDP) defined by $\mathcal{M} = (\mathcal{S}, \mathcal{A}, \mu, p, \gamma)$ where $\mathcal{S}$ is the state space, $\mathcal{A}$ the action space, $\mu \in \Delta(\mathcal{S})$ the initial state distribution, $p : \mathcal{S} \times \mathcal{A} \to \Delta(\mathcal{S})$ the transition kernel and $\gamma \in [0, 1)$ a discount factor. In this setting, an agent is modeled by a policy $\pi : S \to \Delta(A)$ which interacts sequentially with the environment. At first the environment is initialized according to $\mu$ in a state $s_0$. At each timestep $t$, the agent observes a state $s_t \in \mathcal{S}$ and generates an action $a_t \in \mathcal{A}$ to transition via $p$ towards a new state $s_{t+1} \in \mathcal{S}$ leading to a trajectory $\tau = (s_0, a_0, s_1, a_1, ...)$. In practice, this interactive process can repeat itself until an eventual terminal state $s_T$ is reached (termination) at timestep $T$, or until a maximal timestep is reached (truncation), to generate a trajectory $\tau = \{(s_t, a_t, r_t)\}_{t=0}^{T-1} \cup \{s_T\} \in \mathcal{T}$. We assume that we have access to a finite dataset $\mathcal{D}$ of such trajectories collected by an unknown set of policies, typically corresponding to humans or synthetic policies.

**Style and diversity in imitation learning.** To train a policy towards a target behavior, traditional IL methods leverage $\mathcal{D}$ by mimicking its behaviors under the assumption of the combined expertise and homogeneity of its trajectories. In contrast, we assume that $\mathcal{D}$'s behaviors can possibly display a high amount of heterogeneity. Previous literature (Zhan et al., 2020; Mao et al., 2024; Yang et al.,

2024) describes this heterogeneity through various definitions of behavior styles. Denoting $\tilde{\mathcal{T}}$ as the set of (overlapping) subtrajectories, we can generalize those definitions by defining a style as the **labeling** of a subtrajectory $\tau_{t:t+h} \in \tilde{\mathcal{T}}$ given a comparison **criterion** towards a **task** to perform. Hence, a style translates into a specific way to carry out a given task given a criterion. A **task** in the MDP framework is generally defined through a reward function $r : \mathcal{S} \times \mathcal{A} \to [r_{\min}, r_{\max}]$ to maximize along the trajectory. Given a task, an agent can display a range of behaviors that varies greatly. A **criterion** $\lambda : \tilde{\mathcal{T}} \to \mathcal{L}(\lambda)$ is a tool to describe such variations. It can range from *"the vector of an unsupervised learned trajectory encoder"* to *"the speed class of my agent"* and projects any sub-trajectory into a **label** in $\mathcal{L}(\lambda)$. For instance, we can have $z \in \mathcal{L}(\lambda) = \mathbb{R}^d$ or *"fast"* $\in \mathcal{L}(\lambda) = \{$*"slow"*, *"fast"*$\}$. A **behavior style** can consequently be defined in the most general sense as the set of subtrajectories that verify a certain label, given a criterion and a task.

**Style labeling and data programming.** The various definitions of behavior styles in the literature can be divided into unsupervised settings (Li et al., 2017; Hausman et al., 2017; Wang et al., 2017; Mao et al., 2024; Petitbois et al., 2025) and supervised settings (Zhan et al., 2020; Yang et al., 2024). In particular, following Zhan et al. (2020); Yang et al. (2024), we focus on the data programming (Ratner et al., 2017) paradigm, using labeling functions as the criterion. However, unlike Zhan et al. (2020); Yang et al. (2024), which define their labeling functions on full trajectories given any criterion $\lambda$, we define ours as hard-coded functions on subtrajectories $\lambda : \tilde{\mathcal{T}} \to [\![0, |\lambda| - 1]\!]$, with $|\lambda|$ the number of categories of $\lambda$. Using such labeling functions has several benefits. As noted in Zhan et al. (2020), labeling functions are simple to specify yet highly flexible. They reduce **labeling cost** by eliminating manual annotation, which is often time-consuming and expensive, and, crucially, they enhance interpretability, a key limitation of unsupervised approaches, thereby enabling clearer notions of **interpretability** and more direct **alignment measurement**. While previous works as Zhan et al. (2020); Yang et al. (2024) have focused on trajectory-level labels $\lambda(\tau)$, we argue that relying on per-timestep labeling functions, defined in our framework as labels of windows, is a more pragmatic choice. Indeed, as various styles can have various timescales, styles can in fact vary across a trajectory, which can lead to avoidable **credit assignment** issues. As such, given a labeling function $\lambda$, we annotate the dataset $\mathcal{D}$ by marking each state-action pair $(s_t, a_t)$ of each of its trajectories $\tau$ as "contributing" to the style of its corresponding window of radius $w(\lambda)$: $\lambda(\mathcal{D}) = \{(s_t, a_t, z_t), t \in \{0, \ldots, |\tau|\}, \tau \in \mathcal{D}\}$ with $\forall(\tau, t)$ and $z_t = \lambda(\tau_{t-w(\lambda)+1:t+w(\lambda)})$. We illustrate several of such styles in Appendix A.

**Standard performance metrics.** Our goal is to learn a policy $\pi : \mathcal{S} \times \mathcal{L}(\lambda) \to \Delta(\mathcal{A})$ which performs a specific task defined by a given reward $r$, while displaying behaviors calibrated toward given styles. Traditionally, the RL problem corresponds to the maximization of the **task performance metric**, defined as the expected discounted cumulated sum of rewards:

$$J(\pi) = \mathbb{E}_\pi \left[ \sum_{t=0}^\infty \gamma^t r(s_t, a_t) \right] \tag{1}$$

Furthermore, within our framework, given a criterion $\lambda$, playing within a style labeled as $z \in \mathcal{L}(\lambda)$ naturally translates into the maximization of the activation of this style label within the generated trajectory, which corresponds the maximization of the **style alignment metric**, defined as the expected accuracy of the styles:

$$S^{\mathbb{1}}(\pi, \lambda, z) = \mathbb{E}_\pi \left[ \sum_{t=0}^\infty \gamma^t \mathbb{1}\{\lambda(\tau_{t-w(\lambda)+1:t+w(\lambda)}) = z\} \right] \tag{2}$$

$S^{\mathbb{1}}(\pi, \lambda, z)$ cannot be directly optimized within a reinforcement learning framework as $\mathbb{1}\{\lambda(\tau_{t-w(\lambda)+1:t+w(\lambda)})\}$ depends on future states. However, through its annotations, the criterion $\lambda$ defines a distribution $p_\pi^\lambda(z|s, a)$ which corresponds to the probability of the surrounding style being of label $z$ when performing $(s, a)$ under $\pi$. Hence, using $p_\lambda^\pi(z|s, a)$, we propose to optimize instead the following **probabilistic style alignment metric**:

$$S^p(\pi, \lambda, z) = \mathbb{E}_\pi \left[ \sum_{t=0}^\infty \gamma^t p_\pi^\lambda(z|s_t, a_t) \right] \tag{3}$$

This objective corresponds to a Style Conditioned RL (SCRL) problem under the reward $p_\pi^\lambda(z|s, a)$. In practice, estimating $p_\pi^\lambda(z|s, a)$ is challenging and its dependency on $\pi$ makes the optimization of

$S^p(\pi, \lambda, z)$ difficult. As such, we optimize instead $p_{\pi_\mathcal{D}}^\lambda(z|s, a)$ with $\pi_\mathcal{D}$ the sampling policy which we will note $p(z|s, a)$.

**Style alignment as an occupancy measure.** Given a policy $\pi$, its discounted state-action occupancy measure $\rho_\pi : \mathcal{S} \times \mathcal{A} \to \mathbb{R}$ is defined as $\rho_\pi(s, a) = \pi(a|s) \sum_{t=0}^\infty \gamma^t \mathbb{P}(s_t = s|\pi)$. It can be interpreted as the discounted distribution of state-action pairs that the agent will encounter while interacting with $\mathcal{M}$ with $\pi$. For any reward function $r : \mathcal{S} \times \mathcal{A} \to \mathbb{R}$, occupancy measures can allow us to write:

$$J(\pi) = \sum_{s,a} \rho_\pi(s, a) r(s, a) \tag{4}$$

This objective translates into visiting the state-action pairs that yield the most rewards. From this, we can derive the state-action-style occupancy measure for any policy $\pi$ as: $\rho_\pi(s, a, z) = p(z|s, a)\pi(a|s) \sum_{t=0}^\infty \gamma^t \mathbb{P}(s_t = s|\pi)$ and consecutively we can define the style occupancy measure as: $\rho_\pi(z) = \sum_{s,a} \rho_\pi(s, a, z)$. The style occupancy measure corresponds to the discounted distribution of the styles that the agent will encounter while interacting with $\mathcal{M}$ and following $\pi$. We can directly see that:

$$S^p(\pi, \lambda, z) = \sum_{s,a} \rho_\pi(s, a) p(z|s, a) = \sum_{s,a} \rho_\pi(s, a, z) = \rho_\pi(z) \tag{5}$$

Hence, optimizing the style alignment metric directly relates to optimizing style occupancy measure, i.e. to visit the state-action pairs which are the most likely to contribute to the given target style. In the following, we will present a new method to effectively optimize the **style alignment metric** while allowing good **style-conditioned task performance optimization**.

## 4 OPTIMIZING TASK PERFORMANCE UNDER STYLE ALIGNMENT

In this section, we first present in subsection 4.1 the challenges that arise when optimizing the style alignment metric (Equation 3). Then, we describe the methods we use to optimize the task performance (Equation 1) and the style alignment (Equation 3) in the subsections 4.2 and 4.3 respectively. Finally, we introduce our style conditioned task performance optimization method in subsection 4.4.

### 4.1 MOTIVATION

Figure 1: **Long term decision making and stitch challenges for style alignment optimization.** Consider two tasks: **halfcheetah**, where an agent controls a halfcheetah body (Towers et al., 2024) to run along the horizontal axis, and **circle2d**, where the goal is to draw circles in a 2D plane. Each admits style criteria (e.g., running speed, circle position). Achieving styles such as high-speed running or top-right circles requires navigating through zero-signal transitions, demanding **long-term decision marking**, while trajectories in $\mathcal{D}$ may not cover the full MDP, calling for **trajectory stitching**.

As illustrated in Figure 1, solving SCRL problems need for algorithms capable of long-term decision making and stitching, as illustrated in Figure 1, a property lacking in many previous works (Yang et al., 2024; Mao et al., 2024). In the following, we detail the design of our algorithm, motivated by these requirements.

## 4.2 LEARNING TO OPTIMIZE THE TASK PERFORMANCE

The first cornerstone of our objective is to extract from $\mathcal{D}$ a policy $\pi^{r,*} : \mathcal{S} \to \Delta(\mathcal{A})$ that maximizes task performance $J(\pi)$. For this, we employ the well-known IQL algorithm (Kostrikov et al., 2021), which mitigates value overestimation by estimating the optimal value function through expectile regression:

$$\mathcal{L}_{V^r}(\phi^r) = \mathbb{E}_{(s_t,a_t)\sim p^{\mathcal{D}}(s,a)} \left[ \ell_2^\kappa \big( Q_{\theta^r}^r(s_t, a_t) - V_{\phi^r}^r(s_t) \big) \right] \tag{6}$$

$$\mathcal{L}_{Q^r}(\theta^r) = \mathbb{E}_{(s_t,a_t,s_{t+1})\sim p^{\mathcal{D}}(s,a,s')} \left[ \big( r(s_t, a_t) + \gamma V_{\phi^r}^r(s_{t+1}) - Q_{\theta^r}^r(s_t, a_t) \big)^2 \right] \tag{7}$$

where $\ell_2^\kappa(u) = |\kappa - \mathbb{1}\{u < 0\}|u^2, \kappa \in [0.5, 1)$ is the expectile loss, an asymmetric squared loss that biases $V_{\phi^r}^r$ toward the upper tail of the $Q_{\theta^r}^r$ distribution, and $p^{\mathcal{D}}$ defines the uniform distribution of $\mathcal{D}$. The trained $V_{\phi^r}^r$ and $Q_{\theta^r}^r$ are then used to learn a policy network $\pi_{\psi^r}^r$ via Advantage-Weighted Regression (AWR) (Peng et al., 2019):

$$J_{\pi^r}(\psi^r) = \mathbb{E}_{(s_t,a_t)\sim p^{\mathcal{D}}(s,a)} \left[ \exp(\beta^r \cdot A_{\theta^r,\phi^r}^r(s_t, a_t)) \log \pi_{\psi^r}^r(a_t|s_t) \right] \tag{8}$$

with $\beta \in (0, \infty]$ an inverse temperature and advantage: $A_{\theta^r,\phi^r}^r(s_t, a_t) = Q_{\theta^r}^r(s_t, a_t) - V_{\phi^r}^r(s_t)$, which measures how much better or worse action $a_t$ in state $s_t$ is compared to the baseline value. This procedure corresponds to cloning dataset state–action pairs with a bias toward actions with higher advantages.

## 4.3 LEARNING TO OPTIMIZE STYLE ALIGNMENT

To optimize for style alignment, we introduce SCIQL, a simple adaptation of IQL which employs the same principles of relabeling as the GCRL literature (Park et al., 2025) to optimize for any given criterion $\lambda$ the style-conditioned alignment objective: $\pi^{\lambda,*} : \mathcal{S} \to \Delta(\mathcal{A}) \in \arg\max_\pi S(\pi, z), \forall z \in \mathcal{L}(\lambda)$. As in IQL, SCIQL first fits the optimal style-conditioned value functions through neural networks $V_{\phi_\lambda}^\lambda$ and $Q_{\theta_\lambda}^\lambda$ using expectile regression:

$$\mathcal{L}_{V^\lambda}(\phi^\lambda) = \mathbb{E}_{(s_t,a_t)\sim p^{\lambda(\mathcal{D})}(s,a),\, z_t\sim p_{\mathrm{m}}^{\lambda(\mathcal{D})}(z|s_t,a_t)} \left[ \ell_\kappa^2 \big( Q_{\theta^\lambda}^\lambda(s_t, a_t, z_t) - V_{\phi^\lambda}^\lambda(s_t, z_t) \big) \right] \tag{9}$$

$$\mathcal{L}_{Q^\lambda}(\theta^\lambda) = \mathbb{E}_{(s_t,a_t,s_{t+1})\sim p^{\lambda(\mathcal{D})}(s,a,s'),\, z_t\sim p_{\mathrm{m}}^{\lambda(\mathcal{D})}(z|s_t,a_t)} \Big[ \big( \chi_{\omega^\lambda}^\lambda(s_t, a_t, z_t) + \gamma V_{\phi^\lambda}^\lambda(s_{t+1}, z_t) \tag{10}$$
$$- Q_{\theta^\lambda}^\lambda(s_t, a_t, z_t) \big)^2 \Big]$$

with $\chi_{\theta_\chi}(s, a, z)$ an estimator of $p(z|s, a)$. Comparing between several strategies, we empirically found (see Appendix E.1) that taking $\chi_{\omega^\lambda}^\lambda(s_t, a_t, z_t) = \mathbb{1}(z_t = z_{\mathrm{c}})$ with $z_{\mathrm{c}}$ the associated label within $\lambda(\mathcal{D})$ to be one of the best performing methods, which we kept for its simplicity. We sample styles from a mixture $p_{\mathrm{m}}^{\lambda(\mathcal{D})}(z|s, a)$ of a set of sampling distributions: $p_{\mathrm{c}}^{\lambda(\mathcal{D})}(z|s, a)$ which corresponds to the Dirac distribution of the style label associated to $(s, a)$ within its trajectory in $\lambda(\mathcal{D})$, $p_{\mathrm{f}}^{\lambda(\mathcal{D})}(z|s, a)$ which corresponds to the uniform distribution on the styles associated to the future state-actions pairs within $\lambda(\mathcal{D})$ starting from $(s, a)$ and $p_{\mathrm{r}}^{\lambda(\mathcal{D})}(z)$ which corresponds to the uniform distribution of the style labels over the entire dataset $\lambda(\mathcal{D})$. This sampling of styles outside the joint distribution $p^{\lambda(\mathcal{D})}(s, a, z)$ enables to address **distribution-shift**. After that, we extract a style-conditioned policy $\pi_{\psi^\lambda}^\lambda$ through AWR by optimizing:

$$J_{\pi^\lambda}(\psi^\lambda) = \mathbb{E}_{(s_t,a_t)\sim p^{\mathcal{D}}(s,a),\, z_t\sim p_{\mathrm{m}}^{\mathcal{D}}(z|s_t,a_t)} \left[ \exp(\beta^\lambda \cdot A_{\theta^\lambda,\phi^\lambda}^\lambda(s_t, a_t, z_t)) \log \pi_{\psi^\lambda}^\lambda(a_t|s_t, z_t) \right] \tag{11}$$

This objective drives $\pi_{\psi^\lambda}^\lambda$ to copy the dataset's actions with a bias toward actions likely to lead in the future to the visitation of state-actions pairs of high likelihood of contribution to the style in conditioning. This formulation effectively works with styles outside of the joint distribution and leads as we see in the experiment section 5.2 to a more **robust style alignment**.

## 4.4 LEARNING TO PERFORM STYLE-CONDITIONED TASK PERFORMANCE OPTIMIZATION

Most of the time, task performance for the reward $r$ and style alignment for the criterion $\lambda$ are partially incompatible objectives. SORL (Mao et al., 2024) addresses this by optimizing diverse policies using stylized advantage-weighted regression, which seeks to maximize the task performance of anchor policies while constraining updates to prevent collapse toward a single expert policy. Nevertheless, these changes can still induce shifts in the learned policies, hurting style alignment and thus controllability. Consequently, we instead aim to design a method which optimizes the task performance while still preserving style alignment as much as possible. Meanwhile, the advantage is defined as $A(s,a) = Q(s,a) - V(s)$ and quantifies how much better or worse action $a$ is in state $s$ under policy $\pi$. Given it has zero expectation under $\pi$, if $A(s,a) > 0$, taking $a$ in state $s$ improves the expected discounted return compared to sampling from $\pi$, making $(s,a)$ beneficial, while if $A(s,a) < 0$, it lowers it, making $(s,a)$ detrimental. As such, to perform style-conditioned task performance optimization, we propose to use advantages not only as a learning signal to maximize, but also as a mask to filter detrimental transitions when trying to maximize the task performance objective under style alignment constraints. For this, we introduce Gated Advantage Weighted Regression (GAWR), which computes a gated advantage function:

$$\xi^{r|\lambda}(A^{\lambda}, A^{r})(s,a,z) = A^{\lambda}(s,a,z) + \sigma(A^{\lambda}(s,a,z)) \cdot A^{r}(s,a) \tag{12}$$

to train policy $\pi^{r|\lambda}$ for task performance while preserving style alignment:

$$J_{\pi^{r|\lambda}}(\psi^{r|\lambda}) = \mathbb{E}_{(s_t,a_t) \sim p^{\mathcal{D}}(s,a),\, z_t \sim p_m^{\mathcal{D}}(z|s_t,a_t)} \Big[ \exp(\beta^{r|\lambda} \cdot \xi^{r|\lambda}(A^{\lambda}_{\bar{\theta}^{\lambda},\phi^{\lambda}}, A^{r}_{\bar{\theta}^{r},\phi^{r}})(s_t,a_t,z_t))$$
$$\cdot \log \pi^{r|\lambda}_{\psi^{r|\lambda}}(a_t \mid s_t, z_t) \Big] \tag{13}$$

Unlike in SORL, gated advantages can transmit learning signals within non aligned state-action pairs thanks to the advantage summation, filtering detrimental samples instead of non-aligned ones.

We display the pseudocode of the full training pipeline of SCIQL in Algorithm 1. Since the value functions can be learned independently, it is possible to perform these steps in parallel before the policy extraction stage to reduce training time. Furthermore, in practice, similarly to prior IQL and related algorithms (Kostrikov et al., 2021; Park et al., 2024), both value learning and policy extraction are performed simultaneously within a single global training loop.

---

**Algorithm 1** Style-Conditioned Implicit Q-Learning with Gated Advantage Weighted Regression.

---

**Input:** offline dataset $\mathcal{D}$, labeling function $\lambda$
Initialize $\phi^{\lambda}, \theta^{r}, \bar{\theta}^{r}, \theta^{\lambda}, \bar{\theta}^{\lambda}, \psi^{r|\lambda}$
**while** not converged **do**                                    # Train the task value functions
    $\phi^{r} \leftarrow \phi^{r} - \nu_{V^r} \nabla \mathcal{L}_{V^r}(\phi^r)$ according to Equation 6
    $\theta^{r} \leftarrow \theta^{r} - \nu_{Q^r} \nabla \mathcal{L}_{Q^r}(\theta^r)$ according to Equation 7
    $\bar{\theta}^{r} \leftarrow (1 - \upsilon_{\text{Polyak}})\bar{\theta}^{r} + \upsilon_{\text{Polyak}}\theta^{r}$
**end while**
**while** not converged **do**                                    # Train the style value functions
    $\phi^{\lambda} \leftarrow \phi^{\lambda} - \nu_{V^{\lambda}} \nabla \mathcal{L}_{V^{\lambda}}(\phi^{\lambda})$ according to Equation 9
    $\theta^{\lambda} \leftarrow \theta^{\lambda} - \nu_{Q^{\lambda}} \nabla \mathcal{L}_{Q^{\lambda}}(\theta^{\lambda})$ according to Equation 10
    $\bar{\theta}^{\lambda} \leftarrow (1 - \upsilon_{\text{Polyak}})\bar{\theta}^{\lambda} + \upsilon_{\text{Polyak}}\theta^{\lambda}$
**end while**
**while** not converged **do**                                    # Train the policy $\pi^{\lambda}_{\psi^{\lambda}}$ through GAWR
    $\psi^{r|\lambda} \leftarrow \psi^{r|\lambda} + \nu_{\pi^{r|\lambda}} \nabla J_{\pi^{r|\lambda}}(\psi^{r|\lambda})$ according to Equation 13
**end while**

---

# 5 EXPERIMENTS

## 5.1 EXPERIMENTAL SETUP

After introducing environments in section 5.1.1, we tackle the following experimental questions:

1. How does SCIQL compare to previous work on style alignment?

2. Does GAWR help SCIQL perform style conditioned task performance optimization?

3. How does SCIQL compare to previous work on style conditioned task performance optimization?

### 5.1.1 ENVIRONMENTS, TASKS, LABELS AND DATASETS

**Circle2d** (see Figure 1) is a modified version of the environment from Li et al. (2017) and consists of a 2D plane where an agent can roam within a confined square to draw a target circle. For this environment, we define the labels: **position**, **movement_direction**, **turn_direction**, **radius**, **speed**, and **curvature_noise**. We generate two datasets using a hard-coded agent that draws circles with various centers and radii, orientations (clockwise and counter-clockwise), speeds, and action noise levels. The first dataset, **circle2d-inplace-v0**, is obtained by drawing the circle directly from the start position, while the **circle2d-navigate-v0** dataset is obtained by navigating to a target position before drawing the circle. **HalfCheetah** (Todorov et al., 2012) (see Figure 1) is a task where the objective is to control a planar 6-DoF robot to move as far as possible in the forward direction. For this environment, we define the labels: **speed**, **angle**, **torso_height**, **backfoot_height**, and **front-foot_height**. We train a diverse set of HalfCheetah policies using SAC (Haarnoja et al., 2018) to generate three datasets: **halfcheetah-fixed-v0**, where the policy is fixed throughout the trajectory; **halfcheetah-stitch-v0**, where trajectories are split into short segments; and **halfcheetah-vary-v0**, where the policy changes during the trajectory. **HumEnv** (Tirinzoni et al., 2025) is a higher dimensional task consisting in controlling a SMPL skeleton (Loper et al., 2023) with 358-dimensional observations through a 69-dimensional action space to move as fast as possible in a flat plane. In **humenv-simple-v0**, the humanoid is initialized in a standing position. We generate a stylized dataset using the Metamotivo-M1 model provided in Tirinzoni et al. (2025), leading to various ways of moving at different heights and speeds and focus on a **head_height** criterion of 2 labels, **low** and **high**. In **humenv-complex-v0**, the humanoid is initialized in a lying down position, and the dataset is generated as in **humenv-simple-v0**, but with style variations within the trajectory. Also, in **humenv-complex-v0**, we define a **speed** criterion of 3 labels: **immobile**, **slow** and **fast**, and a finer **head_height** criterion of 3 labels: **low**, **medium** and **high**. Further details about each environment, task, labeling function and dataset are provided in Appendix A.

### 5.1.2 BASELINES AND MODEL DETAILS

We compare SCIQL against external state-of-the-art algorithms and a hierarchy of ablations designed to isolate the contributions of SCIQL's components. For the ablations, we begin with standard **BC** Pomerleau (1991) as a non-conditioned reference. We then introduce **Conditioned BC (CBC)**, which incorporates style conditioning using the current trajectory style. Finally, to analyze the benefits of style relabeling, we introduce **SCBC**, an IL variant of **SCIQL** which performs hindsight style relabeling by sampling style labels from the future trajectory, but without value functions. For external comparisons, we evaluate against **BCPMI** (Yang et al., 2024), which extends CBC via mutual-information weighting, and an adapted version of **SORL** (Mao et al., 2024) (see Appendix C), which serves as the primary benchmark for optimizing task performance under style constraints. Further details on architectures and hyperparameters are provided in Appendix C and Appendix B.

## 5.2 RESULTS ON STYLE ALIGNMENT

Our first set of experiments evaluates the capability of SCIQL to achieve style alignment compared to baselines. For each style label $z \in \mathcal{L}(\lambda)$ of each criterion $\lambda$, we perform 10 rollouts across 5 seeds, conditioned on $z$ (except BC, which does not support label conditioning). Each generated trajectory $\tau = \{(s_t, a_t), t \in \{0, \ldots, |\tau| - 1\}\}$ is then annotated as $\lambda(\tau) = \{(s_t, a_t, z_t), t \in \{0, \ldots, |\tau| - 1\}\}$ with $z_t = \lambda(\tau_{t-w(\lambda)+1:t+w(\lambda)}), \forall t \in \{0, \ldots, |\tau| - 1\}$. For each annotated trajectory, we compute its empirical normalized undiscounted style alignment:

$$\hat{S}^{\mathbb{1}}(\lambda(\tau), z) = \frac{1}{|\tau|} \sum_{t=0}^{|\tau|-1} \mathbb{1}\{z_t = z\}, \tag{14}$$

where the normalization by the trajectory length $|\tau|$ ensures that $\hat{S}^{\mathbb{1}}(\lambda(\tau), z) \in [0, 1]$, which hence represents the fraction of timesteps labeled as contributing to the target label. We then average align-

ments over 10 episodes to compute the empirical normalized undiscounted style alignment of our policy, $\hat{S}^{\mathbb{1}}(\pi, \lambda, z)$, which can be seen as the analogue of a GCRL success rate in the SCRL context. Because of the multiplicity of criteria and labels (see Appendix D), we report average alignments across all criteria and labels in Table 1, with full results provided in Appendix D. Standard deviations are computed as the average across 5 seeds for the different tested $(\lambda, z)$. We observe that SCIQL achieves the best style alignment performance by a large margin compared to previous baselines for every dataset, highlighting its effectiveness in long-term decision making and stitching, unlike prior methods. In particular, the performance gap between BC and CBC underscores the necessity of style conditioning. Moreover, the similar performance of SORL in imitation mode ($\beta = 0$), BCPMI, and CBC can be explained by the similarity of their objectives (see Appendix C), all corresponding to a weighted CBC without style relabeling. The performance gap between SCBC and the previous baselines further highlights the importance of integrating trajectory stitching and style relabeling within stylized policies, while the dominance of SCIQL demonstrates the additional benefits of value learning, which augments relabeling by integrating randomly sampled styles during training and enables more effective policy extraction overall. Additionally, SCIQL does not suffer from a drop in alignment in halfcheetah-vary-v0 compared to the previous baselines. CBC exhibits higher variance, while BCPMI, SORL, and SCBC show a decrease in average alignment. This highlights SCIQL's robustness to noisier trajectories, as variations in style during trajectory generation can produce noisy learning signals. In particular, style variations can make SCBC consider the wrong actions as beneficial when sampling future styles for relabeling at train time. A deeper analysis for can be found in Appendix D.

Table 1: **Style alignment results**

| Dataset | BC | CBC | BCPMI | SORL ($\beta = 0$) | SCBC | SCIQL |
|---|---|---|---|---|---|---|
| circle2d-inplace-v0 | 29.1 ± 6.3 | 58.6 ± 2.3 | 58.9 ± 2.6 | 58.9 ± 2.7 | 68.6 ± 2.0 | **74.6 ± 9.3** |
| circle2d-navigate-v0 | 29.1 ± 5.3 | 58.9 ± 2.7 | 59.9 ± 2.3 | 60.0 ± 3.3 | 67.2 ± 1.8 | **75.5 ± 4.7** |
| halfcheetah-fixed-v0 | 30.0 ± 5.9 | 51.2 ± 9.0 | 58.1 ± 8.4 | 53.1 ± 10.6 | 58.0 ± 5.3 | **78.0 ± 1.8** |
| halfcheetah-stitch-v0 | 30.0 ± 6.8 | 52.1 ± 7.6 | 58.9 ± 11.3 | 48.4 ± 12.5 | 57.4 ± 4.7 | **78.0 ± 1.1** |
| halfcheetah-vary-v0 | 30.0 ± 4.5 | 52.0 ± 12.0 | 52.6 ± 17.2 | 46.7 ± 9.5 | 31.7 ± 4.2 | **78.9 ± 0.7** |
| humenv-simple-v0 | 50.0 ± 44.4 | 89.1 ± 22.0 | 79.2 ± 26.7 | 79.4 ± 26.9 | **99.6 ± 0.0** | **99.6 ± 0.0** |
| humenv-complex-v0 | 33.3 ± 4.0 | 47.1 ± 12.8 | 44.6 ± 18.4 | 47.7 ± 6.9 | 33.2 ± 3.5 | **83.5 ± 6.2** |

## 5.3 RESULTS ON STYLE-CONDITIONED TASK PERFORMANCE OPTIMIZATION

To evaluate the capability of SCIQL to perform style-conditioned task performance optimization, we plot the average style alignments and normalized returns of SCIQL without GAWR ($\lambda$), with a style-based GAWR ($\lambda > r$), and with a reward-based GAWR ($r > \lambda$) for reference. We compare against SORL with various temperatures $\beta$, which control the importance of task performance in the SORL objective (see Appendix C). First, we observe in Table 2 that while increasing the importance of task performance raises the returns for both SORL and SCIQL, SCIQL ($\lambda > r$) achieves better style alignment than all SORL variants while significantly improving its task performance over SCIQL ($\lambda$). In particular, while increasing task performance importance in SORL results in a significant decrease in style alignment, GAWR enables SCIQL ($\lambda > r$) to better maintain alignment for the majority of the dataset. Finally, GAWR can also be used for task-conditioned style alignment optimization, allowing SCIQL ($r > \lambda$) to achieve task performance on par with or better than SORL across tasks. To quantify the trade-offs between task and style, we compute complementary metrics in addition to standard evaluations. First, we compute the Hypervolumes (HV) of both approaches and observe that SCIQL achieves a substantial improvement of +41.2% to +163.9% (see Figure 2), indicating that it achieves a better overall task-performance to style-alignment tradeoff than SORL. In particular, SCIQL($\lambda > r$) lies closer to the ideal point $(100, 100)$, corresponding to a reduction in Euclidean distance to the ideal point of 18-28%. This shows that SCIQL reaches a stronger compromise between objectives, effectively shifting the Pareto frontier closer to theoretical optimality. Furthermore, because we consider task performance and style reward asymmetrically and aim to improve task performance while maintaining strong style alignment, we observe that SCIQL($\lambda > r$) preserves the style alignment of SCIQL($\lambda$) in nearly all environments and datasets (excluding halfcheetah-stitch), while substantially improving task performance. A deeper analysis for can be found in Appendix D.

Table 2: **Style-conditioned task performance optimization results.**

| Dataset | Metric | SORL ($\beta = 0$) | SORL ($\beta = 1$) | SORL ($\beta = 3$) | SCIQL ($\lambda$) | SCIQL ($\lambda > r$) | SCIQL ($r > \lambda$) |
|---|---|---|---|---|---|---|---|
| circle2d-inplace-v0 | Style | $58.9 \pm 2.7$ | $54.5 \pm 4.6$ | $53.9 \pm 4.2$ | $74.6 \pm 9.3$ | $71.6 \pm 4.8$ | $47.9 \pm 9.3$ |
| | Task | $16.6 \pm 6.2$ | $70.4 \pm 3.8$ | $73.6 \pm 3.3$ | $6.6 \pm 2.8$ | $68.6 \pm 6.9$ | $89.1 \pm 3.3$ |
| circle2d-navigate-v0 | Style | $60.0 \pm 3.3$ | $58.0 \pm 5.2$ | $57.6 \pm 4.0$ | $75.5 \pm 4.7$ | $76.5 \pm 2.9$ | $56.7 \pm 6.1$ |
| | Task | $18.5 \pm 7.3$ | $69.7 \pm 4.6$ | $72.7 \pm 3.9$ | $7.9 \pm 4.6$ | $66.2 \pm 6.5$ | $87.7 \pm 3.8$ |
| halfcheetah-fix-v0 | Style | $53.1 \pm 10.6$ | $44.4 \pm 6.1$ | $41.3 \pm 4.1$ | $78.0 \pm 1.8$ | $78.1 \pm 1.5$ | $49.7 \pm 5.4$ |
| | Task | $32.1 \pm 8.4$ | $72.7 \pm 5.6$ | $80.6 \pm 3.1$ | $47.6 \pm 2.3$ | $56.5 \pm 2.5$ | $76.6 \pm 5.5$ |
| halfcheetah-stitch-v0 | Style | $48.4 \pm 12.5$ | $41.1 \pm 4.8$ | $42.1 \pm 4.9$ | $78.0 \pm 1.1$ | $60.8 \pm 6.0$ | $33.8 \pm 6.2$ |
| | Task | $31.9 \pm 10.3$ | $81.3 \pm 3.1$ | $78.3 \pm 5.6$ | $47.0 \pm 2.3$ | $70.0 \pm 6.0$ | $80.4 \pm 9.0$ |
| halfcheetah-vary-v0 | Style | $46.7 \pm 9.5$ | $37.0 \pm 3.0$ | $31.1 \pm 2.0$ | $78.9 \pm 0.7$ | $77.8 \pm 1.0$ | $41.8 \pm 5.0$ |
| | Task | $35.9 \pm 9.0$ | $79.0 \pm 3.2$ | $82.6 \pm 3.1$ | $50.6 \pm 1.3$ | $58.0 \pm 1.7$ | $84.6 \pm 3.2$ |
| humenv-simple-v0 | Style | $79.4 \pm 26.9$ | $99.1 \pm 0.9$ | $99.4 \pm 0.4$ | $99.6 \pm 0.0$ | $99.6 \pm 0.1$ | $99.5 \pm 0.2$ |
| | Task | $14.6 \pm 14.5$ | $16.0 \pm 7.5$ | $20.0 \pm 12.5$ | $19.1 \pm 7.1$ | $31.7 \pm 4.8$ | $36.5 \pm 0.4$ |
| humenv-complex-v0 | Style | $47.7 \pm 6.9$ | $25.4 \pm 11.0$ | $23.5 \pm 15.0$ | $83.5 \pm 6.2$ | $90.8 \pm 9.1$ | $33.3 \pm 4.3$ |
| | Task | $5.1 \pm 2.7$ | $29.7 \pm 5.2$ | $27.1 \pm 8.8$ | $11.0 \pm 2.2$ | $15.9 \pm 2.5$ | $41.0 \pm 3.2$ |
| mean_relative_change | Style (%) | +0.0 | -6.9 | -13.9 | +0.0 | -3.1 | -37.2 |
| | Task (%) | +0.0 | +214.5 | +233.1 | +0.0 | +269.9 | +402.1 |

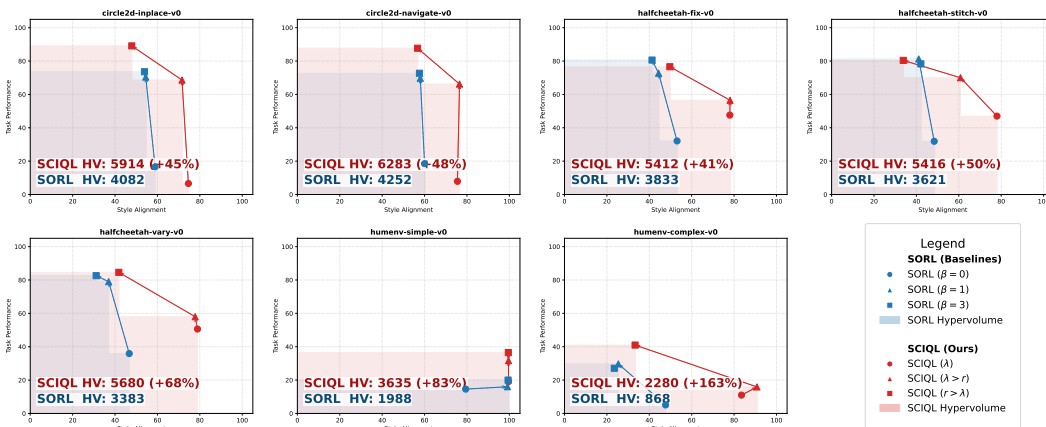

Figure 2: **Pareto fronts and hypervolumes of SORL and SCIQL.**

## 6 CONCLUSION

We propose a novel general definition of behavior styles within the sequential decision making framework and instantiate it by the use of labeling functions to learn **interpretable** styles with a low **labeling cost** and easy **alignment measurement** while effectively avoiding unnecessary **credit assignment** issues by relying on subtrajectories labeling. We then present the SCIQL algorithm which leverages Gated AWR to solve long-term decision making and trajectory stitching challenges while providing superior performance in both style alignment and style-conditioned task performance compared to previous work.

We think that our framework opens the door to several interesting research directions. First, an interesting next step would be to find ways to scale it to a multiplicity of criteria. Furthermore, finding mechanisms to enhance the representation span of labeling functions could also be interesting. Finally, integrating zero-shot capabilities to generate on the fly style-conditioned reinforcement learning policies would be worthwhile to explore.

# 7 REPRODUCIBILITY STATEMENT

To ensure the reproducibility of our work, we detail our environments, tasks labels and datasets in Appendix A, the choice of architecture and hyperparameter in Appendix B and the baselines we use in Appendix C. Moreover, we provide links to clean implementations of our algorithms in JAX (Bradbury et al., 2018) along with the datasets in the following project page: `https://sciql-iclr-2026.github.io/`.

# 8 LLM USE

The writing of this paper has been aided by an LLM for the following purposes: **(1)** Performing searches to help verify the completeness of our related work. **(2)** Checking the grammar and wording of the paper. **(3)** Providing assistance with code debugging and utilities under our close supervision.

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

# A ENVIRONMENTS, TASKS, LABELS AND DATASETS

In this section, we detail our environments, tasks, labels and datasets.

## A.1 CIRCLE2D

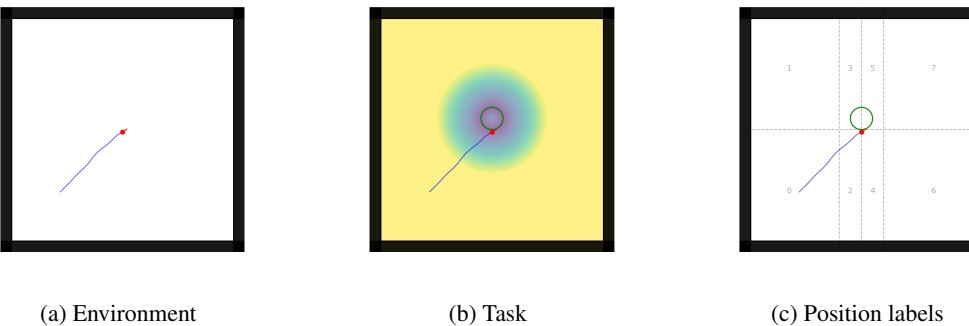

(a) Environment                    (b) Task                    (c) Position labels

Figure 3: **Circle2d environment visualizations.**

**Environment** The Circle2d environment consists in a 2d plane where an agent can roam around within a confined square. Its state space $\mathcal{S}$ corresponds to the history of the 4 previous $(x_{\text{agent}}, y_{\text{agent}}, \theta_{\text{agent}}) \in [[x_{min}, x_{max}] \times [y_{min}, y_{max}] \times [\theta_{min}, \theta_{max}]] = [-50.0, 50.0] \times [-50.0, 50.0] \times [-\pi, \pi]$, padded if needed by repeating to oldest triplet (namely for the beginning of the trajectory). Its action space $\mathcal{A}$ is $[-1, 1]^2$ where the first dimension maps onto a angular shift $\Delta\theta \in [\Delta\theta_{min}, \Delta\theta_{max}] = [-\pi, \pi]$ in radians and the second dimension maps onto a speed in $[v_{min}, v_{max}] = [0.5, 3.0]$. At first, the environment is initialized by sampling a random position from $[[0.7 \cdot x_{min}, 0.7 \cdot x_{max}] \times [0.7 \cdot y_{min}, 0.7 \cdot y_{max}]]$ and a random orientation from $[-\pi, \pi]$. At each timestep $t$, given a state $s_t$ and an action $a_t$, the agent rotates by the corresponding $\Delta\theta_t$ before moving by the displacement vector $\Delta v_t$. The episode is truncated after 1000 timesteps have been reached. We display a minimal visual example of our environment in Figure 3a.

**Task** In Circle2D, we define the task as drawing a target circle given its center $xy_{\text{target}}$ and its radius $radius_{\text{target}}$ and encode it by a reward: $r(s_t, a_t) = -|||xy_{\text{agent}} - xy_{\text{target}}||_2^2 - radius_{\text{target}}|$. In this work, we consider the same fixed circle target along experiments and we display its associated reward colormap in Figure 3b.

**Datasets** We generate for this environment two datasets by using a hard-coded agent which draws circles of various centers and radius, with different orientations (clockwise and counter-clockwise) and different speed and noise levels on the actions. The first dataset **circle2d-inplace-v0** is obtained by directly performing the circle at start position, while the **circle2d-navigate-v0** dataset is obtained by moving around a target position before drawing the circle. We plot in Figure 4 the datasets trajectories.

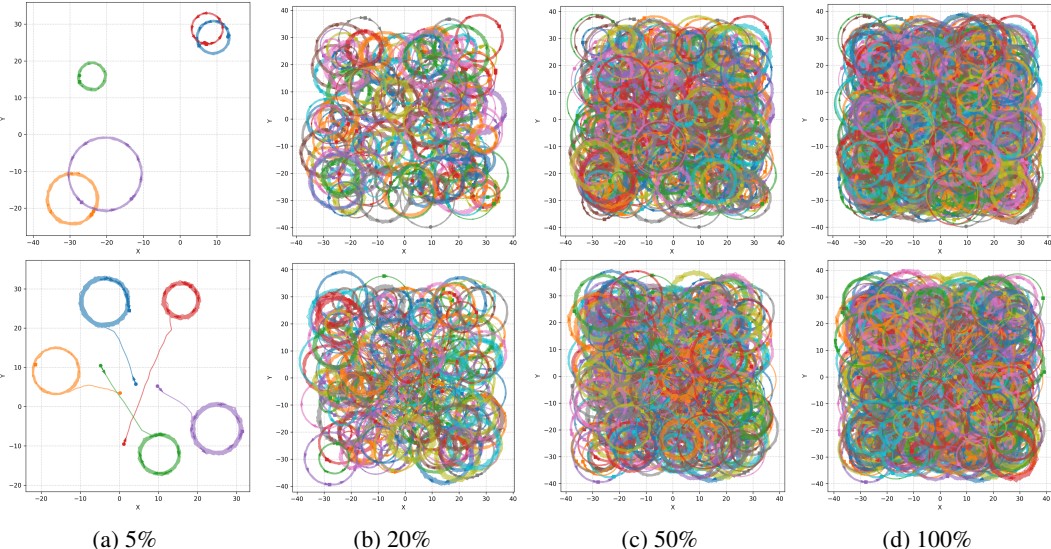

(a) 5%      (b) 20%      (c) 50%      (d) 100%

Figure 4: **Circle2d datasets trajectory visualizations at different percentages.** The top row corresponds to the **circle2d-inplace-v0** while the bottom row corresponds to the **circle2d-navigate-v0**

**Criteria and labels**    We present below the various labeling function we designed for Circle2d.

• **position**: The position labeling function $\lambda_{\text{position}}$ partitions the 2D plane into a fixed grid and assigns to each timestep the index of the cell containing the current position. Concretely, the $x$-axis range $[-30, 30]$ (real units) is split uniformly into 4 bins and the $y$-axis is split at $0$ into 2 bins, yielding $4 \times 2 = 8$ areas. At timestep $t$, with window size $w$, we read every $(x_{t'}, y_{t'})$ in the window $\tau_{t-w+1:t+w}$ and set the label as the majority area. The label set is $\mathcal{L}(\lambda) = [\![0, 7]\!]$. In practice, we take $w = 1$ to mitigate unnecessary credit assignment issues. We plot in Figure 5 the corresponding visuals and histograms.

• **movement direction**: The movement–direction labeling function $\lambda_{\text{move}}$ discretizes the instantaneous displacement direction. For each timestep $t'$, we compute $\Delta p_{t'} = p_{t'+1} - p_{t'}$ and $\theta_{t'} = \text{atan2}(\Delta y_{t'}, \Delta x_{t'})$, and uniformly quantize $[-\pi, \pi)$ into $K = 8$ bins. With window size $w$, the label at $t$ is the majority direction bin over $\{\theta_{t'}\}_{t' \in \tau_{t-w+1:t+w}}$. If $\|\Delta p_{t'}\| < 0.1$ (real units) for a frame, it contributes an undetermined class $u$ (non-promptable). Thus $\mathcal{L}(\lambda) = [\![0, 8]\!]$, with promptable bins $0..7$ and $8 = u$. In practice we use $w = 1$ to mitigate unnecessary credit assignment issues. See Figure 6 for visuals and histograms.

• **turn direction**: The turn–direction labeling function $\lambda_{\text{turn}}$ inherently operates on a centered temporal window to estimate local angular velocity. Let $(\theta_t)_t$ be the unwrapped heading; on an odd window $W_t$ (default size 11), we form $\Delta \theta_{t'} = \theta_{t'+1} - \theta_{t'}$ and compute $\bar{\omega}_t = \frac{1}{|W_t|} \sum_{t' \in W_t} \Delta \theta_{t'}$. If $|\bar{\omega}_t| < 0.1$ rad/step we label "straight," else "left" if $\bar{\omega}_t > 0$ (counter-clockwise) and "right" if $\bar{\omega}_t < 0$ (clockwise). We set $\mathcal{L}(\lambda) = \{0, 1, 2\}$ with $0 = $ right, $1 = $ left, $2 = $ straight (non-promptable). We plot in Figure 7 its visuals and histograms.

• **radius category**: The radius labeling function $\lambda_{\text{radius}}$ also works directly on centered windows. First, on a short window $W_t^{\text{str}}$ (default size 11) we test straightness via the mean absolute heading increment; if it is below $0.1$ rad/step, the label is "straight." Otherwise, on a larger window of positions $W_t^{\text{rad}}$ (default size 51) we fit a circle by least squares and take its radius $r_t$. We uniformly partition $[2, 11]$ (real units) into $K = 3$ bins and assign the corresponding bin; the straight case is encoded as bin $K$. Thus $\mathcal{L}(\lambda) = [\![0, K]\!]$, where $0..K - 1$ denote increasing-radius curved motion and $K$ denotes straight (non-promptable). See Figure 8.

• **speed category**: The speed labeling function $\lambda_{\text{speed}}$ bins the scalar speed. For each timestep $t'$ we compute the speed $v_{t'}$ and uniformly partition $[0.5, 3.0]$ (real units) into $K = 3$ bins. With window size $w$, the label at $t$ is the majority speed bin over $\{v_{t'}\}_{t' \in \tau_{t-w+1:t+w}}$. Hence $\mathcal{L}(\lambda) = [\![0, K - 1]\!]$.

In practice we take $w = 1$ to mitigate unnecessary credit assignment issues. We plot in Figure 9 the corresponding visuals and histograms.

• **curvature noise**: The curvature-noise labeling function $\lambda_{\text{noise}}$ computes a variability statistic on a centered window. With unwrapped heading $(\theta_t)_t$, we define $\Delta\theta_{t'} = \theta_{t'+1} - \theta_{t'}$ and $\Delta^2\theta_{t'} = \Delta\theta_{t'+1} - \Delta\theta_{t'}$. On an odd window $W_t$ (default size 51), we take $\sigma_t = \text{std}\big(\{\Delta^2\theta_{t'}\}_{t' \in W_t}\big)$ and uniformly bin $\sigma_t$ into $K = 3$ categories over $[0.0, 0.8]$. Hence $\mathcal{L}(\lambda) = [\![0, K-1]\!]$. We plot in Figure 10 its visuals and histograms.

*Notes.* For all labels that use windows, the implementation ensures an odd, centered window around $t$; where relevant, "straight"/"undetermined" classes are excluded from promptable labels but kept in $\mathcal{L}(\lambda)$ for completeness. Bin edges are uniform by default and configurable through the class constructors.

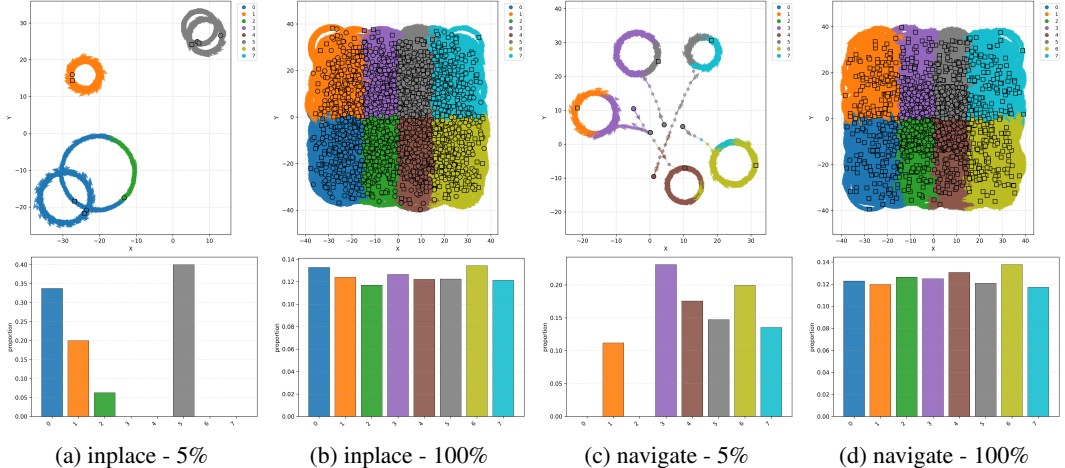

| (a) inplace - 5% | (b) inplace - 100% | (c) navigate - 5% | (d) navigate - 100% |

Figure 5: **Circle2d position label visualizations at different percentages.**

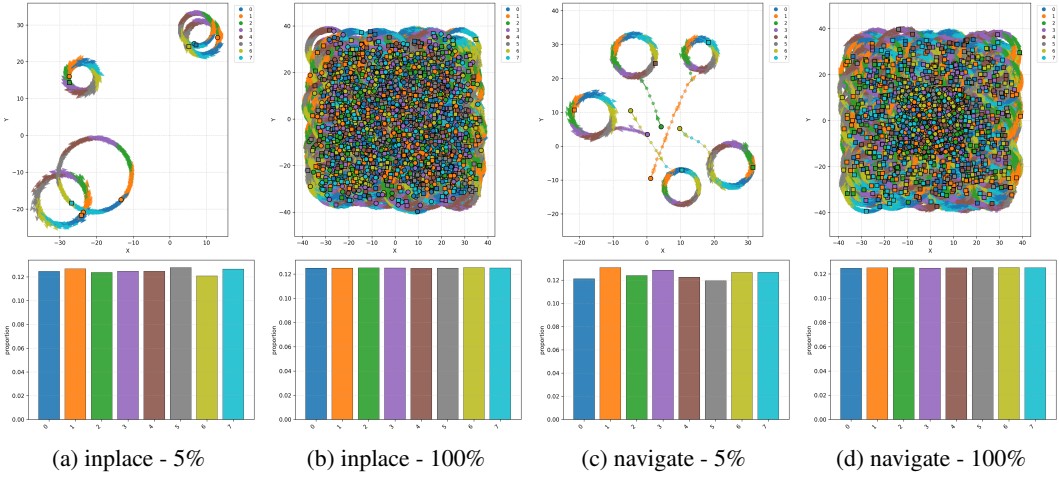

| (a) inplace - 5% | (b) inplace - 100% | (c) navigate - 5% | (d) navigate - 100% |

Figure 6: **Circle2d movement direction label visualizations at different percentages.**

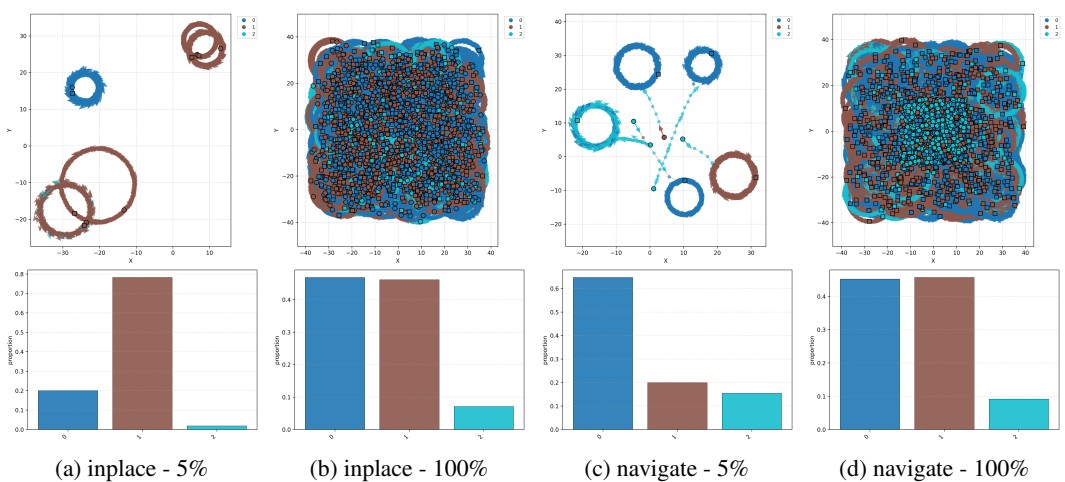

Figure 7: **Circle2d turn direction label visualizations at different percentages.**

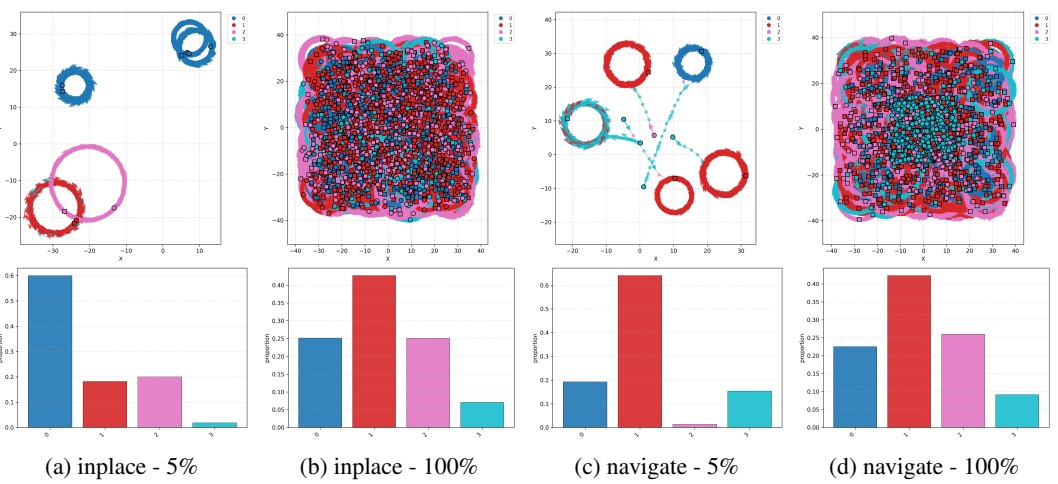

Figure 8: **Circle2d radius label visualizations at different percentages.**

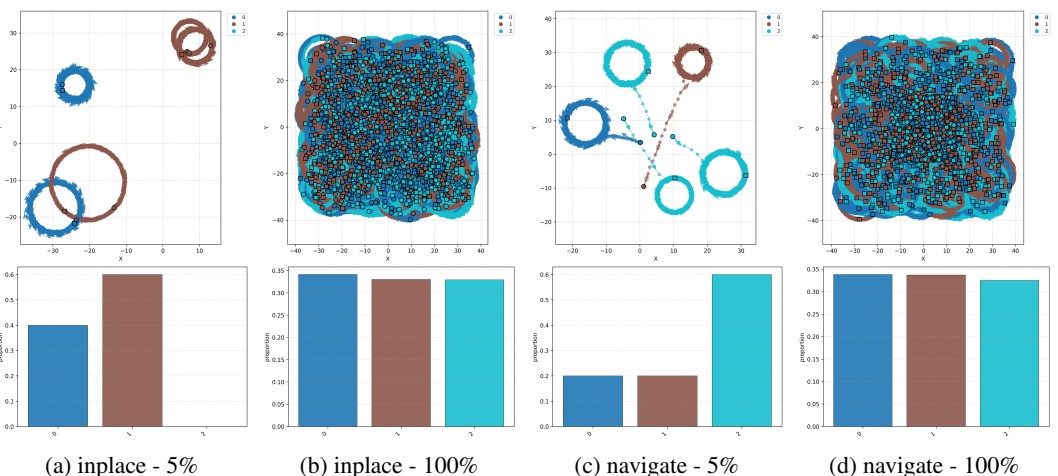

Figure 9: **Circle2d speed label visualizations at different percentages.**

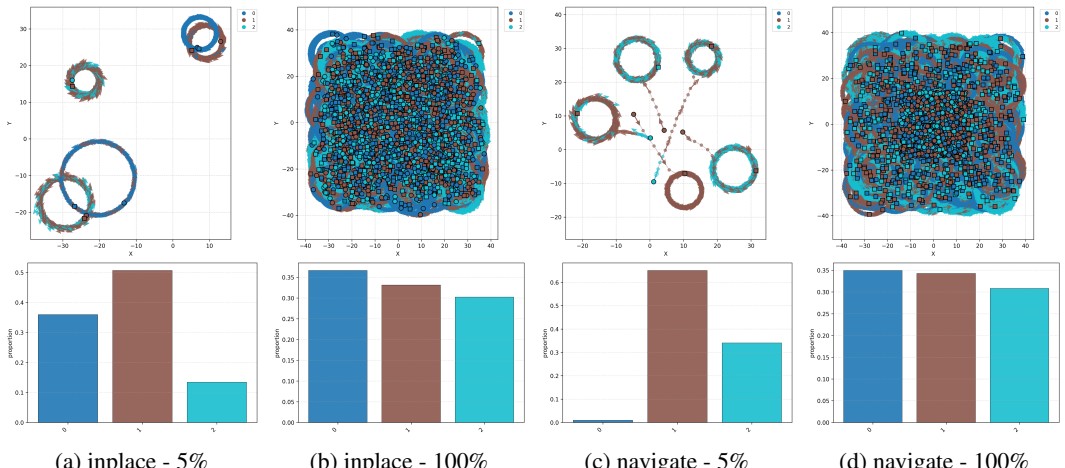

(a) inplace - 5%    (b) inplace - 100%    (c) navigate - 5%    (d) navigate - 100%

Figure 10: **Circle2d curvature noise visualizations at different percentages.**

### A.2  HALFCHEETAH

**Environment**  HalfCheetah (Todorov et al., 2012; Towers et al., 2024) is an environment consisting in controlling a 6-DoF 2-dimensional robot composed of 9 body parts and 8 joints connecting them. The environment as a time limit of 1000 timesteps. Details about this environment can be read in Towers et al. (2024).

**Task**  As implemented in Towers et al. (2024), at each timestep $t$, the agent applies continuous control actions $\mathbf{a}_t \in \mathbb{R}^d$ that drive the joints of the cheetah. The environment evaluates performance using a reward which encourages rapid forward progress while penalizing excessive control effort. Formally, the forward velocity of the torso is

$$v_t = \frac{x_{t+1} - x_t}{\Delta t},$$

where $x_t$ is the torso position along the horizontal axis and $\Delta t$ is the simulator timestep. The reward combines a positive term proportional to forward velocity with a quadratic control penalty:

$$r_t \;=\; w_f\, v_t \;-\; w_c \sum_{i=1}^{d} a_{t,i}^2,$$

where $w_f$ is the forward-reward weight and $w_c$ is the control-cost weight. Thus, the agent must learn to run efficiently: moving forward quickly while keeping joint torques as small as possible.

**Datasets**  To generate the datasets, we train a diverse set of HalfCheetah policies through SAC (Haarnoja et al., 2018). We construct several *archetype* policies defined by Gaussian-shaped reward functions that bias behavior toward specific styles. The **Height** archetype rewards the torso maintaining a target vertical position $z_{\text{torso}}$ at specified values, thereby inducing qualitatively distinct gaits: *crawling* ($z \approx 0.5$ with $\sigma = 0.04$), *normal running* ($z \approx 0.6$ with $\sigma = 0.04$), or *upright running* ($z \approx 0.7$ with $\sigma = 0.04$). The **Speed** archetype rewards locomotion close to a desired forward velocity, producing policies that move at *slow pace* ($v \approx 1.5$), *medium pace* ($v \approx 5.0$), or *fast pace* ($v \approx 10.0$). Finally, the **Angle** archetype shapes behavior around the torso pitch angle, leading to policies that prefer *upright* ($\theta \approx -0.2$ with $\sigma = 0.05$), *flat* ($\theta \approx 0.0$ with $\sigma = 0.05$), or *crouched* ($\theta \approx 0.2$ with $\sigma = 0.05$) postures while still advancing forward. These archetypes yield a diverse collection of locomotion styles that serve as structured variations of the base HalfCheetah task. Then, we generate three datasets: **halfcheetah-fixed-v0**, where the archetype policy is fixed during the trajectory; **halfcheetah-stitch-v0**, where the trajectories are cut into shorter segments from the **halfcheetah-fixed-v0** dataset; and **halfcheetah-vary-v0**, where the policy archetype changes within the same trajectory. Each dataset contain $10^6 = 1000(\text{episodes}) * 1000(\text{timesteps})$ steps, with the stitch datasets containing more episodes as it cuts the fix dataset episodes.

**Criteria and labels**   We present below the various labeling functions we designed for HalfCheetah. Each labeling function $\lambda$ maps raw environment signals to a discrete label sequence, optionally smoothed by a majority vote over a window $\tau_{t-w+1:t+w}$. In practice, we take $w = 1$ to mitigate unnecessary credit assignment issues.

• **speed**: The speed labeling function $\lambda_{\text{speed}}$ discretizes the forward velocity magnitude $|v_t|$. We define a range $[v_{\min}, v_{\max}] = [0.1, 10.0]$ (real units) and split it uniformly into $K = 3$ bins, yielding the labels $\mathcal{L}(\lambda_{\text{speed}}) = [\![0, 2]\!]$. At timestep $t$, we assign the bin index corresponding to $|v_t|$, and take the majority bin across the window. See Figure 11.

• **angle**: The angle labeling function $\lambda_{\text{angle}}$ discretizes the torso pitch $\theta_t$. We define $[\theta_{\min}, \theta_{\max}] = [-0.3, 0.3]$ (radians) and split uniformly into $K = 3$ bins, yielding the label set $\mathcal{L}(\lambda_{\text{angle}}) = [\![0, 2]\!]$. At timestep $t$, we assign the bin index of $\theta_t$, and take the majority label over the window. See Figure 12.

• **torso height**: The torso–height labeling function $\lambda_{\text{torso}}$ discretizes the vertical torso position $h_t$. We define $[h_{\min}, h_{\max}] = [0.4, 0.8]$ (real units) and split into $K = 3$ bins, giving $\mathcal{L}(\lambda_{\text{torso}}) = [\![0, 2]\!]$. Labels are assigned per timestep and smoothed by majority vote. See Figure 13.

• **back-foot height**: The back-foot labeling function $\lambda_{\text{bf}}$ discretizes the vertical position of the back foot $h_t^{\text{bf}}$. We define $[h_{\min}, h_{\max}] = [0.0, 0.3]$ and split into $K = 4$ bins, giving $\mathcal{L}(\lambda_{\text{bf}}) = [\![0, 3]\!]$. Labels are taken per timestep and majority-voted. See Figure 14.

• **front-foot height**: The front-foot labeling function $\lambda_{\text{ff}}$ discretizes the vertical position of the front foot $h_t^{\text{ff}}$ in the same manner as the back-foot: $[0.0, 0.3]$ split into $K = 4$ bins, yielding $\mathcal{L}(\lambda_{\text{ff}}) = [\![0, 3]\!]$. See Figure 15.

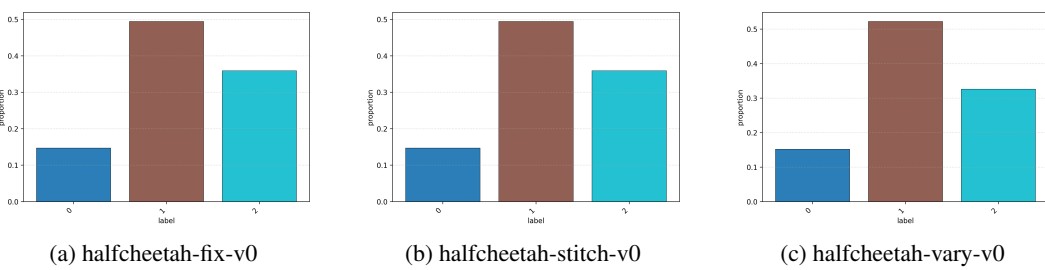

(a) halfcheetah-fix-v0             (b) halfcheetah-stitch-v0             (c) halfcheetah-vary-v0

Figure 11: **HalfCheetah speed label histograms.**

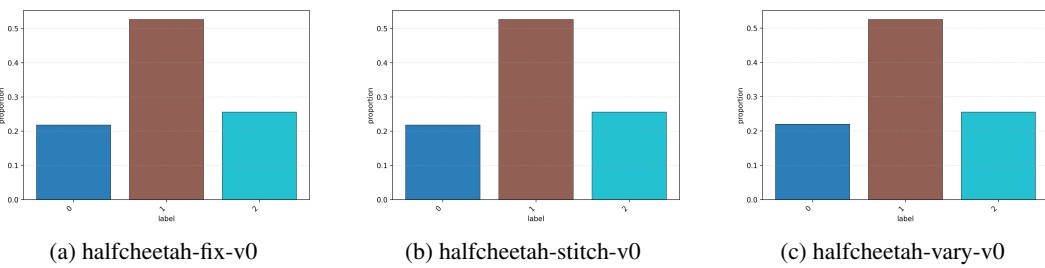

(a) halfcheetah-fix-v0             (b) halfcheetah-stitch-v0             (c) halfcheetah-vary-v0

Figure 12: **HalfCheetah angle label histograms.**

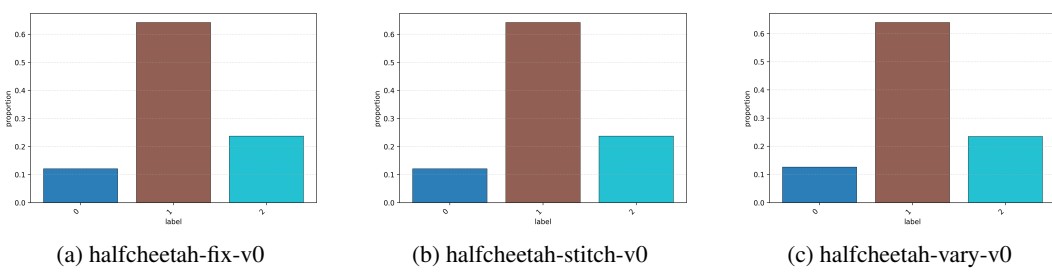

(a) halfcheetah-fix-v0      (b) halfcheetah-stitch-v0      (c) halfcheetah-vary-v0

Figure 13: **HalfCheetah torso height label histograms.**

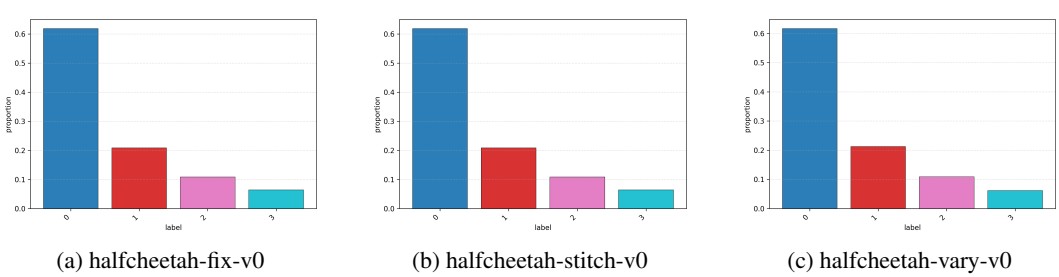

(a) halfcheetah-fix-v0      (b) halfcheetah-stitch-v0      (c) halfcheetah-vary-v0

Figure 14: **HalfCheetah backfoot height label histograms.**

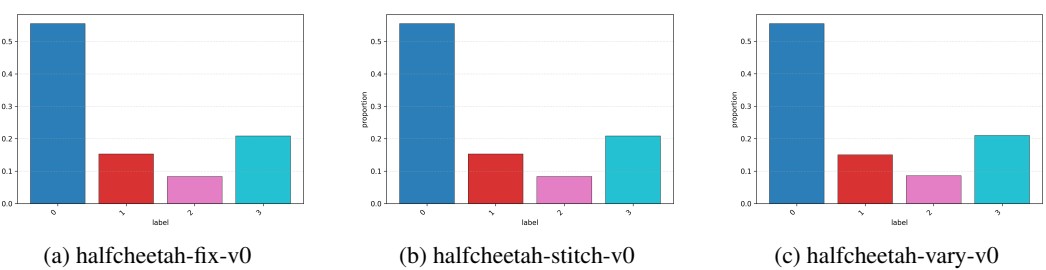

(a) halfcheetah-fix-v0      (b) halfcheetah-stitch-v0      (c) halfcheetah-vary-v0

Figure 15: **HalfCheetah frontfoot height label histograms.**

### A.3 HUMENV

**Environment** The HumEnv environment (Tirinzoni et al., 2025) is built on the SMPL skeleton (Loper et al., 2023), which consists of 24 rigid bodies, among which 23 are actuated. This SMPL skeleton is widely used in character animation and is well suited for expressing natural human-like stylized behaviors. HumEnv's observations consist in the concatenation of the body poses (70 D), body rotations (144 D) and angular velocities (144D) resulting in a 358-dimensional vector. It moves the body using a proportional derivative controller resulting in a 69-dimensional action space. This task has consequently a higher dimensionality of (358, 69) compared to HalfCheetah's (17, 6) dimensionality. We consider two types of HumEnv environments, HumEnv-Simple, which initializes the humanoid in a standing position, and HumEnv-Complex, which initializes the humanoid in a lying down position.

**Task** At each timestep $t$, the agent applies continuous control actions $\mathbf{a}_t \in \mathbb{R}^d$. The environments evaluate performance using a reward that encourages high-speed movement in the horizontal plane, modulated by a control efficiency term. Formally, let $\mathbf{v}_{t,xy}$ denote the velocity vector of the center of mass projected onto the horizontal plane (ignoring vertical movement). The reward is defined as the norm of this velocity, scaled by a multiplicative control factor:

$$r_t = \alpha(\mathbf{a}_t) \cdot \|\mathbf{v}_{t,xy}\|_2,$$

where $\alpha(\mathbf{a}_t) \in [0.8, 1.0]$ is a smoothness coefficient derived from a quadratic tolerance function on the control inputs $\mathbf{a}_t$ provided in Tirinzoni et al. (2025).

**Datasets** We generated for each environment a stylized dataset using the Metamotivo-M1 model provided in Tirinzoni et al. (2025), using various ways of moving at different heights and speeds. Since, the Metamotivo-M1 model was trained with a regularization towards the AMASS motion-capture dataset (Mahmood et al., 2019a), it provides more natural and human-like stylized behaviors.

**Criteria and labels** We present below the various labeling functions we designed for the HumEnv environments. Each labeling function $\lambda$ maps raw environment signals to a discrete label sequence, optionally smoothed by a majority vote over a window $\tau_{t-w+1:t+w}$. In practice, we take $w = 1$ to mitigate unnecessary credit assignment issues.

• **simple - head height**: For HumEnv-Simple, we focused our study on a single **head_height** criterion of two labels, namely **low** and **high**. The simple - head_height labeling function discretizes the vertical head position $h_t$ using a single threshold at 1.2. This results in $K = 2$ bins ($h_t < 1.2$ and $h_t \geq 1.2$), yielding the label set $\mathcal{L}(\lambda_{\text{simple\_head}}) = [\![0, 1]\!]$. See Figure 16a.

• **complex - speed**: For the HumEnv-Complex, we added a new **speed** criterion. The speed labeling function $\lambda_{\text{speed}}$ discretizes the center-of-mass velocity magnitude $|v_t|$. Based on the agent's movement capabilities, we define three distinct regimes: immobile ($|v_t| < 0.2$), slow ($0.2 \leq |v_t| \leq 3.0$), and fast ($|v_t| > 3.0$). This yields $K = 3$ bins with labels $\mathcal{L}(\lambda_{\text{speed}}) = [\![0, 2]\!]$. See Figure 16b.

• **complex - head height**: For the HumEnv-Complex, we also complexified the **complex - head_height** criteria by adding a new label for a total of 3 labels. The head-height labeling function $\lambda_{\text{complex\_head}}$ discretizes the vertical position of the agent's head $h_t$. We define thresholds at 0.4 and 1.2 to capture different postures: lying down, crouching and standing. The space is split into $K = 3$ bins: $h_t < 0.4$, $0.4 \leq h_t \leq 1.2$, and $h_t > 1.2$, yielding $\mathcal{L}(\lambda_{\text{complex\_head}}) = [\![0, 2]\!]$. See Figure 16c.

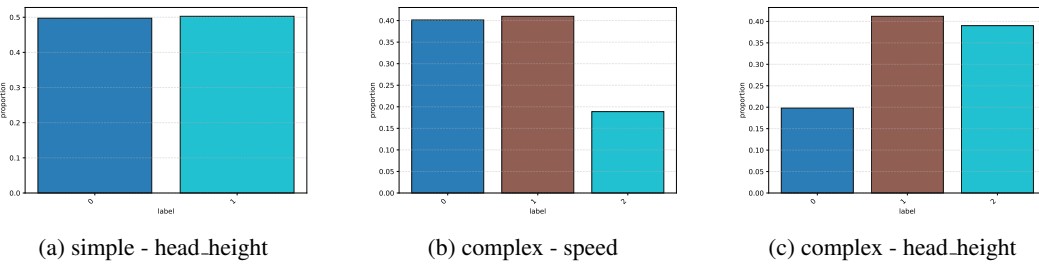

(a) simple - head_height          (b) complex - speed          (c) complex - head_height

Figure 16: **HumEnv label histograms.**

# B  ARCHITECTURES AND HYPERPARAMETERS

**Optimization:** For all baselines, when necessary, labels are encoded as latent variables of dimension 16 via an embedding matrix. We optimize all networks using the Adam optimizer with a learning rate of $3 \cdot 10^{-3}$, employing cosine learning-rate decay for the policies, a batch size of 256, and $10^5$ gradient steps for the $\chi$ estimators and $10^6$ for the other networks. Value functions $V$ additionally use layer normalization. Unless otherwise specified, we use the IQL hyperparameters $\beta = 3$, $\kappa = 0.7$, and $\gamma = 0.99$, and perform Polyak averaging on the $Q$-networks with coefficient 0.005.

**Architectures:** For Circle2d and HalfCheetah, the policies $\pi$, value networks $V, Q$, and estimators $\chi$ are MLPs with hidden size $[256, 256]$ and ReLU activations. For HumEnv, the policies are MLPs with hidden size $[1024, 1024, 1024]$ and ReLU activations.

**Relabeling:** In SCIQL, we use $p_r^{\lambda(\mathcal{D})}$ as $p_m^{\lambda(\mathcal{D})}$ for all criteria of all environments.

**Implementations:** Our implementations are written in JAX (Bradbury et al., 2018), and take inspiration from Nishimori (2024), allowing little training durations. In Circle2D and HalfCheetah, we get for BC ($\approx 2$min), CBC ($\approx 3$min), BCPMI ($\approx 4$min), SORL ($\approx 15$min), SCBC ($\approx 3$min) and SCIQL ($\approx 35$min) on a NVIDIA V100 GPU for training runs. In HumEnv, we get for BC ($\approx 5$min), CBC ($\approx 5$min), BCPMI ($\approx 6$min), SORL ($\approx 23$min), SCBC ($\approx 5$min) and SCIQL ($\approx 45$min) on a NVIDIA A100 GPU for training runs. Our code and datasets can be found in our project website: https://sciql-iclr-2026.github.io/.

## C  BASELINES

In this subsection, we describe in more details our baselines.

**Behavior Cloning (BC).**  BC (Pomerleau, 1991) is the simplest of our baselines and learns by maximizing the likelihood of actions given states through supervised learning on $\mathcal{D}$:

$$J_{\mathrm{BC}}(\pi) = \mathbb{E}_{(s,a)\sim p^{\mathcal{D}}(s,a)}[\log \pi(a|s)]. \tag{15}$$

We use this baseline as a reference for style alignment performance without conditioning.

**Conditioned Behavior Cloning (CBC).**  CBC is the simplest style-conditioned method of our baselines and consists in concatenating to BC's states their associated label within $\lambda(\mathcal{D})$:

$$J_{\mathrm{CBC}}(\pi) = \mathbb{E}_{(s,a)\sim p^{\mathcal{D}}(s,a),\, z\sim p^{\mathcal{D}}_{\mathrm{cur}}(z|s,a)}[\log \pi(a|s,z)] \tag{16}$$

This baseline serves as a reference to test the various benefits of subsequent methods to better perform style alignment optimization.

**Behavior Cloning with Pointwise Mutual Information weighting (BCPMI).**  BCPMI (Yang et al., 2024) seeks to address credit assignment issues between state–action pairs and style labels by relying on their mutual information estimates. For this, BCPMI uses Mutual Information Neural Estimation (MINE). In the information-theoretic setting, let $S$, $A$, and $Z$ be random variables corresponding to states, actions, and styles, respectively. The mutual information between state–action pairs $(S, A)$ and styles $Z$ can be written as the Kullback–Leibler (KL) divergence between the joint distribution $P_{S,A,Z}$ and the product of their marginals $P_{S,A} \otimes P_Z$:

$$I(S, A; Z) = D_{KL}(P_{S,A,Z} \,\|\, P_{S,A} \otimes P_Z). \tag{17}$$

As directly estimating this mutual information is difficult, MINE relies on the Donsker–Varadhan lower bound:

$$I(S, A; Z) \geq \sup_{T\in\mathcal{F}} \mathbb{E}_{(s,a,z)\sim P_{S,A,Z}}[T(s,a,z)] - \log\left(\mathbb{E}_{(s,a,z)\sim P_{S,A}\otimes P_Z}[e^{T(s,a,z)}]\right), \tag{18}$$

where $\mathcal{F}$ denotes a class of functions $T : \mathcal{S} \times \mathcal{A} \times \mathcal{Z} \to \mathbb{R}$. According to Donsker & Varadhan (1975), optimizing this bound yields

$$T^*(s, a, z) = \log \frac{p(s, a, z)}{p(s, a)p(z)} = \log \frac{p(z|s, a)}{p(z)}. \tag{19}$$

BCPMI trains a neural network to approximate $T^*(s, a, z)$ and uses it to weight CBC's learning objective, increasing the impact of transitions with high style relevance while reducing that of less relevant ones:

$$J_{\mathrm{MINE}}(T) = \mathbb{E}_{(s,a)\sim p^{\lambda(\mathcal{D})}(s,a),\, z\sim p_c^{\lambda(\mathcal{D})}(z|s,a)}[T(s,a,z)] - \log\left(\mathbb{E}_{(s,a)\sim p^{\mathcal{D}}(s,a),\, z\sim p_r^{\lambda(\mathcal{D})}(z)}[e^{T(s,a,z)}]\right), \tag{20}$$

$$J_{\mathrm{BC-PMI}}(\pi) = \mathbb{E}_{(s,a)\sim p^{\lambda(\mathcal{D})}(s,a),\, z\sim p_c^{\lambda(\mathcal{D})}(z|s,a)}[\exp(T^*(s,a,z))\log \pi(a|s,z)]. \tag{21}$$

This baseline is notable as it constitutes a first step toward addressing the credit assignment challenges in style-conditioned policy learning. However, as it strictly focuses on imitation learning rather than task performance, it does not support style mixing and is therefore not designed to address distribution shifts at inference time, unlike our method.

**Stylized Offline Reinforcement Learning (SORL):**  SORL (Mao et al., 2024) is an important baseline to consider since it both addresses the optimization of policy diversity and task performance. Initially designed within a unsupervised learning setting, SORL is a two step algorithm which aims to learn a diverse set of high-performing policies from $\mathcal{D}$. First, SORL uses the Expectation-Maximisation (EM) algorithm to first learn a finite set of diverse policies $\{\mu^{(i)}\}$ to capture the heterogeneity of $\mathcal{D}$. The E step aims to fit an estimate $\hat{p}(z = i|\tau)$ the posteriors $p(z = i|\tau)$,

associating each trajectory to a given style among $N$ styles. The M step aims to train the stylized policies $\{\mu^{(i)}\}$ according to their associated style through $\hat{p}(z = i|\tau)$:

$$\underline{\text{E step:}} \; \forall i \in \{0, ..., N-1\}, \hat{p}(z = i|\tau) \approx \frac{1}{Z} \sum_{(s,a) \in \tau} \mu^{(i)}(a|s) \tag{22}$$

$$\underline{\text{M step:}} \; \forall i \in \{0, ..., N-1\}, J_{\text{SORL - M step}}(\mu^{(i)}) = \frac{1}{|\mathcal{D}|} \sum_{\tau \in \mathcal{D}} \sum_{i=1}^{m} \hat{p}(z = i|\tau) \sum_{(s,a) \in \tau} \log \mu^{(i)}(a|s)$$
$$\tag{23}$$

Then, to perform task performance optimization while preserving a certain amount of diversity, SORL proposes to train from $\{\mu^{(i)}\}$ a set of policies $\{\pi^{(i)}\}$ by solving the following constrained problem:

$$\forall i \in \{0, ..., N-1\}, \quad \pi^{(i)} = \arg\max_{\pi^{(i)}} J(\pi^{(i)}) \tag{24}$$

$$\text{s.t.} \quad \mathbb{E}_{s \sim \rho_{\mu^{(i)}}(s)} D_{KL}\big(\pi^{(i)}(\cdot|s) \, \| \, \mu^{(i)}(\cdot|s)\big) \leq \epsilon, \quad \int_a \pi^{(i)}(a|s) \, da = 1, \, \forall s. \tag{25}$$

By using its associated Lagrangian optimization problem, Mao et al. (2024) show that this problem can be casted into a Stylized Advantage Weighted Regression (SAWR) objective:

$$\forall i \in \{0, ..., N-1\}, J_{\text{SORL - SAWR}}(\pi^{(i)}) = \mathbb{E}_{\tau \sim \mathcal{D}} \hat{p}(z = i|\tau) \sum_{(s,a) \in \tau} \log \pi^{(i)}(a|s) \exp\left(\frac{1}{\alpha} A^r(s,a)\right). \tag{26}$$

In our supervised setting, the first step translates into the learning of a style conditioned policy $\mu^{\lambda,*} : \mathcal{S} \to \Delta(\mathcal{A}) \in \arg\max_{\pi} S(\mu, z), \forall z \in \mathcal{L}(\lambda)$ by optimizing the style alignment objective while the second step translates into optimizing $\mu^{\lambda,*}$'s performance by learning under the solution $\pi^{r,\lambda,*}$ of the following constrained problem:

$$\forall z \in \mathcal{L}(\lambda), \pi^{r,\lambda,*}(\cdot|\cdot, z) = \underset{\pi(\cdot|\cdot,z)}{\arg\max} \, J(\pi(\cdot|\cdot, z)) \tag{27}$$

$$\text{s.t.} \; \mathbb{E}_{s \sim \rho_{\mu(\cdot|\cdot,z)}(s)} D_{KL}(\pi(\cdot|s, z)||\mu(\cdot|s, z)) \leq \varepsilon, \int_a \pi(\cdot|s, z) = 1, \forall s \tag{28}$$

Let $z \in \mathcal{L}(\lambda)$ be a style label. Following a similar path as Peng et al. (2019) and Mao et al. (2024), we can state that maximizing $J(\pi(\cdot|\cdot, z))$ is similar as maximizing the expected improvement $\eta(\pi(\cdot|\cdot, z)) = J(\pi(\cdot|\cdot, z)) - J(\mu(\cdot|\cdot, z))$, which can be express as Schulman et al. (2017) show as:

$$\eta(\pi(\cdot|\cdot, z)) = \mathbb{E}_{s \sim \rho_{\pi(\cdot|\cdot,z)}(s)} \mathbb{E}_{a \sim \pi(\cdot|s,z)}[A^{\mu(\cdot|\cdot,z)}(s,a)] \tag{29}$$

Like Peng et al. (2019) showed, we can substitute $\rho_{\pi(\cdot|\cdot,z)}$ to $\rho_{\mu(\cdot|\cdot,z)}$ to simplify this optimization problem as the resulting error has been shown to be bounded by $D_{KL}(\pi(\cdot|\cdot, z)||\mu(\cdot|\cdot, z))$ Schulman et al. (2017). Furthermore, Peng et al. (2019) and Mao et al. (2024) approximate $A^{\mu(\cdot|\cdot,z)}(s,a)$ by the advantage $A^{\mu}(s,a)$ where $\mu$ represents the policy distribution of the dataset. In our setting, we will use the advantage $A^r(s,a)$ estimated through IQL to be coherent with SCIQL. Consequently, SORL's stylized advantage weighted regression becomes in our context:

$$\pi^{r,\lambda,*}(\cdot|\cdot, z) = \underset{\pi(\cdot|\cdot,z)}{\arg\max} \, \mathbb{E}_{s \sim \rho_{\mu(\cdot|\cdot,z)}(s)} \mathbb{E}_{a \sim \pi(\cdot|s,z)}[A^r(s,a)] \tag{30}$$

$$\text{s.t.} \; \mathbb{E}_{s \sim \rho_{\mu(\cdot|\cdot,z)}(s)} D_{KL}(\pi(\cdot|s, z)||\mu(\cdot|s, z)) \leq \varepsilon, \int_a \pi(\cdot|s, z) = 1, \forall s \tag{31}$$

As Peng et al. (2019) and Mao et al. (2024), we compute the corresponding Lagrangian of this optimization problem:

$$L(\pi(\cdot|\cdot, z), \alpha^\mu, \boldsymbol{\alpha}^\pi) = \mathbb{E}_{s \sim \rho_{\mu(\cdot|\cdot, z)}}\Big[\mathbb{E}_{a \sim \pi(\cdot|s, z)} A^r(s, a) \tag{32}$$

$$+ \alpha^\mu\big(\varepsilon - D_{KL}(\pi(\cdot|s, z) \,\|\, \mu(\cdot|s, z)))\big)\Big] \tag{33}$$

$$+ \int_s \boldsymbol{\alpha}_s^\pi\Big(1 - \int_a \pi(a|s, z)\, da\Big) ds \tag{34}$$

$$= \int_s \rho_{\mu(\cdot|\cdot, z)}(s) ds\Big[\int_a \pi(a|s, z) da A^r(s, a) \tag{35}$$

$$+ \alpha^\mu\Big(\varepsilon - \int_a \pi(a|s, z) \log \frac{\pi(a|s, z)}{\mu(a|s, z)} da\Big] \tag{36}$$

$$+ \int_s \boldsymbol{\alpha}_s^\pi\Big(1 - \int_a \pi(a|s, z)\, da\Big) ds \qquad\qquad = \tag{37}$$

with $\alpha^\mu \geq 0$ and $\boldsymbol{\alpha}^\pi = \{\boldsymbol{\alpha}_s^\pi \in \mathbb{R}, s \in \mathcal{S}\}$ the Lagrange multipliers. We differentiate $L(\pi(\cdot|\cdot, z), \alpha^\mu, \boldsymbol{\alpha}^\pi)$ as:

$$\frac{\partial L}{\partial \pi(a|s, z)} = \rho_{\mu(\cdot|s, z)}(s)\Big[A^r(s, a) - \alpha^\mu \log \pi(a|s, z) + \alpha^\mu \log \mu(a|s, z) - \alpha^\mu\Big] - \boldsymbol{\alpha}_s^\pi \tag{38}$$

Setting this derivative to zero yields the following closed-form solution:

$$\pi^*(a|s, z) = \frac{1}{Z(s, z)} \mu(a|s, z) \exp\Big(\frac{1}{\alpha^\mu} A^r(s, a)\Big), \tag{39}$$

where $Z(s, z)$ is the normalization term defined as:

$$Z(s, z) = \exp\Big(\frac{1}{\rho_{\mu(\cdot|\cdot, z)}(s)} \frac{\boldsymbol{\alpha}_s^\pi}{\alpha^\mu} + 1\Big). \tag{40}$$

Finally, as Peng et al. (2019) and Mao et al. (2024), we estimate $\pi^*(\cdot|\cdot, z)$ with a neural network policy $\pi_\psi(\cdot|\cdot, z)$ by solving:

$$\arg\min_\psi \mathbb{E}_{s \sim p^{\lambda(\mathcal{D})}(s|z)}\Big[D_{KL}\big(\pi^*(\cdot|s, z) \,\|\, \pi_\psi(\cdot|s, z)\big)\Big] \tag{41}$$

$$= \arg\min_\psi \mathbb{E}_{s \sim p^{\lambda(\mathcal{D})}(s|z)}\Big[\int_a \big(\pi^*(a|s, z) \log \pi^*(a|s, z) - \pi^*(a|s, z) \log \pi_\psi(a|s, z)\big) da\Big] \tag{42}$$

$$= \arg\min_\psi -\mathbb{E}_{s \sim p^{\lambda(\mathcal{D})}(s|z)}\Big[\int_a \pi^*(a|s, z) \log \pi_\psi(a|s, z)\, da\Big] \tag{43}$$

$$= \arg\min_\psi -\mathbb{E}_{s \sim p^{\lambda(\mathcal{D})}(s|z)}\Big[\int_a \frac{1}{Z(s, z)} \mu(a|s, z) \exp\big(\tfrac{1}{\alpha^\mu} A^r(s, a)\big) \log \pi_\psi(a|s, z)\, da\Big] \tag{44}$$

$$= \arg\min_\psi -\mathbb{E}_{(s,a) \sim p^{\lambda(\mathcal{D})}(s,a|z)}\Big[\frac{1}{Z(s, z)} \exp\big(\tfrac{1}{\alpha^\mu} A^r(s, a)\big) \log \pi_\psi(a|s, z)\Big] \tag{45}$$

$$= \arg\min_\psi -\mathbb{E}_{(s,a) \sim p^{\lambda(\mathcal{D})}(s,a)}\Big[p(z|s, a) \frac{1}{Z(s, z)} \exp\big(\tfrac{1}{\alpha^\mu} A^r(s, a)\big) \log \pi_\psi(a|s, z)\Big] \tag{46}$$

By neglecting the absorbing constant as Peng et al. (2019); Mao et al. (2024), we can finally express the SORL objective in our supervised version:

$$\arg\min_\psi -\mathbb{E}_{(s,a) \sim p^{\lambda(\mathcal{D})}(s,a)}\big[p(z|s, a)\, \exp\big(\tfrac{1}{\alpha^\mu} A^r(s, a)\big) \log \pi_\psi(a|s, z)\big] \tag{47}$$

As we want to optimize this objective for all $z \in \mathcal{L}(\lambda)$, we write below the general objective:

$$\arg\min_\psi -\mathbb{E}_{(s,a) \sim p^{\lambda(\mathcal{D})}(s,a)}\Big[\frac{1}{|\lambda|} \sum_{z=0}^{|\lambda|-1} p(z|s, a)\, \exp\big(\tfrac{1}{\alpha^\mu} A^r(s, a)\big) \log \pi_\psi(a|s, z)\Big] \tag{48}$$

As in SCIQL, we can employ several strategies to estimate $p(z|s, a)$ through an estimator $\chi(s, a, z)$ which we all detail in appendix E.1. Additionally, the advantage functions can be learned offline through IQL as in SCIQL. Hence, we can obtain our adapted SORL objectives by taking $\beta = 1/\alpha^\mu$:

$$\mathcal{L}_{\text{SORL}}(V_r) = \mathbb{E}_{(s,a)\sim p^{\mathcal{D}}(s,a)}[\ell_\kappa^2(\bar{Q}_r(s, a) - V_r(s))] \tag{49}$$

$$\mathcal{L}_{\text{SORL}}(Q_r) = \mathbb{E}_{(s,a,s')\sim p^{\mathcal{D}}(s,a,s')}[r(s, a) + \gamma V_r(s') - Q_r(s, a))^2] \tag{50}$$

$$J_{\text{SORL}}(\pi) = \mathbb{E}_{(s,a)\sim p^{\mathcal{D}}(s,a)}\frac{1}{|\lambda|}\sum_{z=0}^{|\lambda|-1}\chi(s, a, z)e^{\beta A^r(s,a)}\log\pi(a|s, z) \tag{51}$$

**Style-Conditioned Behavior Cloning (SCBC):** SCBC corresponds to a simpler behavior cloning version of SCIQL whose objective can be written as:

$$J_{\text{SCBC}}(\pi) = \mathbb{E}_{(s,a)\sim p^{\mathcal{D}}(s,a), z\sim p_{\text{f}}^{\mathcal{D}}(z|s,a)}[\log\pi(a|s, z)] \tag{52}$$

This baseline is interesting as it shows both how style mixing with hindsight relabeling can be beneficial to style alignment while highlighting the impact of value learning when compared to SCIQL. For instance, value learning allows for relabeling outside of $p_{\text{f}}^{\lambda(\mathcal{D})}$ on top of optimizing the policy.

## D  ADDITIONAL TABLES

Table 3: **Experiment complexity**

| Environment | Criterion | $n_{\text{labels}}$ | $n_{\text{datasets}}$ | $n_{\text{seeds}}$ | Total trainings | $n_{\text{eval\_episodes}}$ | Total evals episodes |
|---|---|---|---|---|---|---|---|
| circle2d | position | 8 | 2 | 5 | 80 | 10 | 800 |
| | movement_direction | 8 | 2 | 5 | 80 | 10 | 800 |
| | turn_direction | 2 | 2 | 5 | 20 | 10 | 200 |
| | radius | 15 | 2 | 5 | 150 | 10 | 1500 |
| | speed | 15 | 2 | 5 | 150 | 10 | 1500 |
| | curvature_noise | 3 | 2 | 5 | 45 | 10 | 450 |
| halfcheetah | speed | 3 | 3 | 5 | 45 | 10 | 450 |
| | angle | 3 | 3 | 5 | 45 | 10 | 450 |
| | torso_height | 3 | 3 | 5 | 45 | 10 | 450 |
| | backfoot_height | 4 | 3 | 5 | 60 | 10 | 600 |
| | frontfoot_height | 4 | 3 | 5 | 60 | 10 | 600 |
| humenv-simple | head_height | 2 | 1 | 5 | 10 | 10 | 100 |
| humenv-complex | speed | 3 | 1 | 5 | 15 | 10 | 150 |
| | head_height | 3 | 1 | 5 | 15 | 10 | 150 |
| all | 14 criteria | 76 | - | - | 820 | - | 8200 |

In this section, we display the full results for both style alignment and style-conditioned task performance optimization. These tables are computed for each environment and criterion $\lambda$ by averaging performance across 5 seeds and all labels in $\mathcal{L}(\lambda)$. Table 3 reports the evaluation complexity statistics of our experiments, which, for each algorithm variant, requires 820 training runs and 8200 evaluation episodes. Normalized per seed, this corresponds to $820/5 = 164$ runs per algorithm, which justifies our use of averages in Table 1, Table 2, Table 4, and Table 5. In the following, we write additional remarks about the full results tables.

**Style alignment:** In Table 4, SCIQL achieves better style alignment on most criteria, while being slightly lower on the **turn_direction**, **radius**, and **speed** criteria of Circle2d. This can be explained by the fact that these criteria do not require relabeling, and we show in Appendix E.2 that optimal performance can be recovered by changing the sampling distribution from $p_{\text{r}}^{\lambda(\mathcal{D})}$ that we globally use to $p_{\text{c}}^{\lambda(\mathcal{D})}$ for those particular criteria. Additionally, methods that do not perform style relabeling perform worse in inplace than in navigate for styles corresponding to specific subsets of the state space, such as position, highlighting the importance of style relabeling for alignment. For halfcheetah, SCIQL largely dominates all baselines demonstrating SCIQL's robustness to noisier trajectories. Namely, in the halfcheetah-vary-v0, SCIQL dominates even more the baselines. In particular, we recall from Appendix C that SCBC sees a important decrease in its style alignment. This can be explained by the nature of the relabeling used in SCBC. For a given observed state-action pair in the dataset $(s, a)$, SCBC samples a futur style $z_{\text{f}}$ from the future of the trajectory and considers $(s, a, z_{\text{f}})$ as expert behavior. Indeed, for SCBC, every action is expert to reach the styles in the future of its trajectory. However, when style variations occur within the trajectory, for instance when alternating low and high speeds $(z_{\text{slow}}, ..., z_{\text{fast}}, ..., z_{\text{slow}}, ...)$ an action conditributing to high speed $(s, a, z_t)$ with $z_{\text{f}} = z_{\text{fast}}$ could be relabeled as $(s, a, z_{\text{slow}})$, provoking the learning of an action for high speeds while being conditioned on $z_{\text{slow}}$. SCIQL solves this problem by adding an advantage weighted regression mechanism to always strives to reach as fast as possible style alignment. consequently lowering thus the weights of wrong labels.

Table 4: **Style alignment results (full).**

| Dataset | BC | CBC | BCPMI | SORL ($\beta = 0$) | SCBC | SCIQL |
|---|---|---|---|---|---|---|
| circle2d-inplace-v0 - position | 12.5 ± 6.9 | 15.0 ± 10.3 | 16.3 ± 13.5 | 14.9 ± 11.6 | 65.9 ± 11.5 | 98.0 ± 0.3 |
| circle2d-inplace-v0 - movement_direction | 12.5 ± 0.2 | 4.4 ± 1.6 | 4.1 ± 1.4 | 5.3 ± 4.2 | 12.5 ± 0.3 | 20.5 ± 4.4 |
| circle2d-inplace-v0 - turn_direction | 50.0 ± 25.1 | 100.0 ± 0.0 | 100.0 ± 0.1 | 100.0 ± 0.1 | 100.0 ± 0.0 | 82.6 ± 26.3 |
| circle2d-inplace-v0 - radius | 33.3 ± 1.2 | 99.1 ± 2.0 | 99.7 ± 0.6 | 99.8 ± 0.4 | 100.0 ± 0.0 | 96.1 ± 5.3 |
| circle2d-inplace-v0 - speed | 33.3 ± 4.2 | 99.9 ± 0.1 | 99.9 ± 0.0 | 99.9 ± 0.0 | 99.9 ± 0.0 | 91.6 ± 13.3 |
| circle2d-inplace-v0 - curvature_noise | 33.3 ± 0.0 | 33.3 ± 0.0 | 33.3 ± 0.1 | 33.3 ± 0.0 | 33.3 ± 0.0 | 59.1 ± 6.1 |
| circle2d-inplace-v0 - all | 29.1 ± 6.3 | 58.6 ± 2.3 | 58.9 ± 2.6 | 58.9 ± 2.7 | 68.6 ± 2.0 | **74.6 ± 9.3** |
| circle2d-navigate-v0 - position | 12.5 ± 7.4 | 16.7 ± 9.5 | 24.0 ± 11.8 | 22.3 ± 14.8 | 58.5 ± 9.5 | 98.4 ± 0.2 |
| circle2d-navigate-v0 - movement_direction | 12.5 ± 0.2 | 5.7 ± 4.9 | 3.2 ± 0.2 | 4.9 ± 3.7 | 12.5 ± 0.2 | 27.0 ± 5.7 |
| circle2d-navigate-v0 - turn_direction | 50.0 ± 13.4 | 100.0 ± 0.0 | 100.0 ± 0.0 | 100.0 ± 0.1 | 99.6 ± 0.1 | 96.0 ± 5.7 |
| circle2d-navigate-v0 - radius | 33.3 ± 10.6 | 98.1 ± 1.7 | 98.8 ± 1.4 | 99.7 ± 0.4 | 99.2 ± 0.9 | 95.8 ± 5.6 |
| circle2d-navigate-v0 - speed | 33.3 ± 0.0 | 99.9 ± 0.0 | 99.9 ± 0.0 | 99.6 ± 0.7 | 99.9 ± 0.0 | 96.0 ± 4.5 |
| circle2d-navigate-v0 - curvature_noise | 33.3 ± 0.0 | 33.3 ± 0.1 | 33.3 ± 0.3 | 33.3 ± 0.0 | 33.4 ± 0.1 | 40.0 ± 6.7 |
| circle2d-navigate-v0 - all | 29.1 ± 5.3 | 58.9 ± 2.7 | 59.9 ± 2.3 | 60.0 ± 3.3 | 67.2 ± 1.8 | **75.5 ± 4.7** |
| halfcheetah-fixed-v0 - speed | 33.3 ± 11.2 | 73.9 ± 11.8 | 77.6 ± 9.0 | 73.0 ± 20.3 | 95.9 ± 1.2 | 96.0 ± 1.6 |
| halfcheetah-fixed-v0 - angle | 33.3 ± 4.5 | 57.7 ± 15.5 | 68.0 ± 11.3 | 60.0 ± 15.5 | 55.2 ± 7.4 | 99.1 ± 1.1 |
| halfcheetah-fixed-v0 - torso_height | 33.3 ± 6.0 | 70.9 ± 11.1 | 82.2 ± 10.0 | 73.2 ± 8.9 | 79.3 ± 8.3 | 96.8 ± 3.5 |
| halfcheetah-fixed-v0 - backfoot_height | 25.0 ± 2.5 | 26.9 ± 2.6 | 29.6 ± 3.9 | 28.4 ± 2.8 | 32.4 ± 6.8 | 47.5 ± 2.0 |
| halfcheetah-fixed-v0 - frontfoot_height | 25.0 ± 5.5 | 26.5 ± 3.9 | 33.3 ± 7.8 | 30.7 ± 5.7 | 27.0 ± 3.0 | 50.5 ± 0.8 |
| halfcheetah-fixed-v0 - all | 30.0 ± 5.9 | 51.2 ± 9.0 | 58.1 ± 8.4 | 53.1 ±10.6 | 58.0 ± 5.3 | **78.0 ± 1.8** |
| halfcheetah-stitch-v0 - speed | 33.3 ± 8.7 | 79.9 ± 8.0 | 70.1 ± 17.7 | 57.1 ± 23.2 | 92.0 ± 3.3 | 96.3 ± 0.5 |
| halfcheetah-stitch-v0 - angle | 33.3 ± 8.0 | 50.4 ± 14.2 | 72.1 ± 18.9 | 55.0 ± 20.4 | 60.8 ± 5.8 | 99.5 ± 0.2 |
| halfcheetah-stitch-v0 - torso_height | 33.3 ± 9.9 | 72.6 ± 7.2 | 87.1 ± 7.7 | 71.5 ± 10.7 | 80.1 ± 6.8 | 96.9 ± 1.4 |
| halfcheetah-stitch-v0 - backfoot_height | 25.0 ± 3.8 | 28.6 ± 2.7 | 30.0 ± 6.3 | 28.0 ± 3.4 | 27.3 ± 3.9 | 47.0 ± 2.4 |
| halfcheetah-stitch-v0 - frontfoot_height | 25.0 ± 3.6 | 29.1 ± 5.9 | 35.3 ± 6.0 | 30.2 ± 5.0 | 27.0 ± 3.5 | 50.3 ± 0.8 |
| halfcheetah-stitch-v0 - all | 30.0 ± 6.8 | 52.1 ± 7.6 | 58.9 ±11.3 | 48.4 ±12.5 | 57.4 ± 4.7 | **78.0 ± 1.1** |
| halfcheetah-vary-v0 - speed | 33.3 ± 6.9 | 63.3 ± 15.5 | 56.4 ± 23.2 | 54.3 ± 14.3 | 37.8 ± 5.8 | 96.7 ± 0.1 |
| halfcheetah-vary-v0 - angle | 33.3 ± 4.6 | 59.2 ± 24.2 | 46.4 ± 22.1 | 39.7 ± 10.8 | 34.8 ± 3.9 | 99.2 ± 0.6 |
| halfcheetah-vary-v0 - torso_height | 33.3 ± 7.6 | 79.3 ± 10.9 | 92.6 ± 7.5 | 77.0 ± 11.8 | 36.2 ± 6.1 | 98.8 ± 0.3 |
| halfcheetah-vary-v0 - backfoot_height | 25.0 ± 1.7 | 29.6 ± 4.5 | 32.9 ± 27.3 | 31.8 ± 5.3 | 25.1 ± 2.2 | 49.5 ± 1.4 |
| halfcheetah-vary-v0 - frontfoot_height | 25.0 ± 1.8 | 28.7 ± 5.1 | 34.9 ± 5.7 | 30.6 ± 5.3 | 24.8 ± 2.8 | 50.4 ± 1.0 |
| halfcheetah-vary-v0 - all | 30.0 ± 4.5 | 52.0 ±12.0 | 52.6 ±17.2 | 46.7 ± 9.5 | 31.7 ± 4.2 | **78.9 ± 0.7** |
| humenv-simple-v0 - head_height | 50.0 ± 44.4 | 89.1 ± 22.0 | 79.2 ± 26.7 | 79.4 ± 26.9 | **99.6 ± 0.0** | **99.6 ± 0.0** |
| humenv-simple-v0 - all | 50.0 ± 44.4 | 89.1 ± 22.0 | 79.2 ± 26.7 | 79.4 ± 26.9 | **99.6 ± 0.0** | **99.6 ± 0.0** |
| humenv-complex-v0 - speed | 33.3 ± 5.2 | 32.6 ± 7.1 | 32.1 ± 13.6 | 34.3 ± 4.7 | 34.1 ± 5.8 | **83.7 ± 5.9** |
| humenv-complex-v0 - head_height | 33.3 ± 2.7 | 61.6 ± 18.5 | 57.1 ± 23.3 | 61.1 ± 9.2 | 32.4 ± 1.3 | **83.3 ± 6.6** |
| humenv-complex-v0 - all | 33.3 ± 4.0 | 47.1 ± 12.8 | 44.6 ± 18.4 | 47.7 ± 6.9 | 33.2 ± 3.5 | **83.5 ± 6.2** |

**Style-conditioned task performance optimization results:** We see in Table 5 that choosing SORL's temperature $\beta_{\text{SORL}}$ is challenging, as finding a good balance between style alignment and task performance is highly sensitive to its value. For instance, in halfcheetah-vary-v0 - speed, as in many other settings, increasing $\beta_{\text{SORL}}$ from 0 to 1 leads to an immediate drop in style alignment. In halfcheetah-vary-v0 - torso height, the decreases occur more gradually, with drops appearing both when moving from $\beta_{\text{SORL}} = 0$ to $\beta_{\text{SORL}} = 1$ and from $\beta_{\text{SORL}} = 1$ to $\beta_{\text{SORL}} = 3$. In contrast, SCIQL shows no such degradation. These examples highlight that tuning SORL's temperature for style-conditioned task performance optimization can be troublesome, as it requires precise adjustment and the optimal value may vary across styles. SCIQL's temperature parameter $\beta_{\text{SCIQL}}$ differs fundamentally: it does not encode the trade-off between style alignment and task performance. Instead, it is inherited directly from IQL's temperature parameter $\beta_{\text{IQL}}$, while the trade-off itself is handled by the Gated Advantage Weighted Regression. Experimentally, we find that setting $\beta_{\text{SCIQL}}$ equal to the values of $\beta_{\text{IQL}}$ commonly used in the literature, typically chosen as 1.0, 3.0, and 10.0 (Kostrikov et al., 2021; Park et al., 2024; 2025), is an effective heuristic. Hence, SCIQL maintains strong alignment by design while significantly improving task performance, without requiring precise fine-tuning.

Table 5: **Style-conditioned task performance optimization results (full).**

| Dataset | Metric | SORL ($\beta = 0$) | SORL ($\beta = 1$) | SORL ($\beta = 3$) | SCIQL ($\lambda$) | SCIQL ($\lambda > r$) | SCIQL ($r > \lambda$) |
|---|---|---|---|---|---|---|---|
| circle2d-inplace-v0 - all | Style | 58.9 ± 2.7 | 54.5 ± 4.6 | 53.9 ± 4.2 | 74.6 ± 9.3 | 71.6 ± 4.8 | 47.9 ± 9.3 |
| circle2d-inplace-v0 - all | Task | 16.6 ± 6.2 | 70.4 ± 3.8 | 73.6 ± 3.3 | 6.6 ± 2.8 | 68.6 ± 6.9 | 89.1 ± 3.3 |
| circle2d-inplace-v0 - position | Style | 14.9 ± 11.6 | 15.5 ± 5.5 | 12.1 ± 3.2 | 98.0 ± 0.3 | 96.1 ± 1.9 | 31.5 ± 6.8 |
| circle2d-inplace-v0 - position | Task | 12.8 ± 7.4 | 79.2 ± 8.8 | 80.4 ± 7.7 | 2.6 ± 0.6 | 17.3 ± 4.1 | 69.3 ± 7.8 |
| circle2d-inplace-v0 - movement_direction | Style | 5.3 ± 4.2 | 5.5 ± 3.4 | 4.7 ± 1.7 | 20.5 ± 4.4 | 14.5 ± 2.3 | 12.5 ± 0.8 |
| circle2d-inplace-v0 - movement_direction | Task | 0.5 ± 0.1 | 0.6 ± 0.1 | 0.6 ± 0.2 | 1.3 ± 0.2 | 80.8 ± 11.3 | 93.4 ± 3.3 |
| circle2d-inplace-v0 - turn_direction | Style | 100.0 ± 0.1 | 98.2 ± 1.3 | 97.9 ± 2.2 | 82.6 ± 26.3 | 85.5 ± 11.3 | 64.0 ± 16.9 |
| circle2d-inplace-v0 - turn_direction | Task | 14.3 ± 3.2 | 88.4 ± 1.7 | 90.1 ± 3.1 | 6.9 ± 5.8 | 90.8 ± 3.7 | 95.0 ± 1.9 |
| circle2d-inplace-v0 - radius_category | Style | 99.8 ± 0.4 | 77.1 ± 12.2 | 72.6 ± 5.3 | 96.1 ± 5.3 | 99.9 ± 0.1 | 57.1 ± 16.3 |
| circle2d-inplace-v0 - radius_category | Task | 28.3 ± 10.0 | 78.0 ± 4.6 | 87.4 ± 2.3 | 6.5 ± 3.2 | 53.9 ± 10.4 | 90.2 ± 2.2 |
| circle2d-inplace-v0 - speed_category | Style | 99.9 ± 0.0 | 97.4 ± 4.8 | 96.2 ± 5.0 | 91.6 ± 13.3 | 94.5 ± 7.6 | 88.4 ± 14.7 |
| circle2d-inplace-v0 - speed_category | Task | 21.0 ± 8.2 | 86.3 ± 3.6 | 91.8 ± 2.4 | 19.5 ± 6.2 | 91.5 ± 2.1 | 93.2 ± 2.0 |
| circle2d-inplace-v0 - curvature_noise | Style | 33.3 ± 0.0 | 33.5 ± 0.3 | 39.8 ± 8.0 | 59.1 ± 6.1 | 38.9 ± 5.5 | 33.6 ± 0.3 |
| circle2d-inplace-v0 - curvature_noise | Task | 22.8 ± 8.0 | 89.6 ± 4.2 | 91.3 ± 4.2 | 2.6 ± 0.8 | 77.5 ± 9.7 | 93.3 ± 2.4 |
| circle2d-navigate-v0 - all | Style | 60.0 ± 3.3 | 58.0 ± 5.2 | 57.6 ± 4.0 | 75.5 ± 4.7 | 76.5 ± 2.9 | 56.7 ± 6.1 |
| circle2d-navigate-v0 - all | Task | 18.5 ± 7.3 | 69.7 ± 4.6 | 72.7 ± 3.9 | 7.9 ± 4.6 | 66.2 ± 6.5 | 87.7 ± 3.8 |
| circle2d-navigate-v0 - position | Style | 22.3 ± 14.8 | 15.7 ± 4.5 | 13.9 ± 3.1 | 98.4 ± 0.2 | 96.0 ± 2.2 | 35.9 ± 10.4 |
| circle2d-navigate-v0 - position | Task | 19.8 ± 10.2 | 63.3 ± 13.8 | 69.4 ± 13.1 | 2.8 ± 0.9 | 20.1 ± 2.8 | 64.1 ± 9.3 |
| circle2d-navigate-v0 - movement_direction | Style | 4.9 ± 3.7 | 5.8 ± 5.4 | 5.6 ± 4.1 | 27.0 ± 5.7 | 18.4 ± 4.0 | 12.6 ± 0.8 |
| circle2d-navigate-v0 - movement_direction | Task | 0.4 ± 0.0 | 0.7 ± 0.6 | 0.4 ± 0.1 | 1.1 ± 0.1 | 63.3 ± 13.4 | 94.5 ± 1.3 |
| circle2d-navigate-v0 - turn_direction | Style | 100.0 ± 0.1 | 99.6 ± 0.4 | 99.8 ± 0.1 | 96.0 ± 5.7 | 100.0 ± 0.0 | 81.9 ± 6.3 |
| circle2d-navigate-v0 - turn_direction | Task | 18.4 ± 11.4 | 92.5 ± 3.2 | 93.4 ± 2.6 | 2.7 ± 1.3 | 94.4 ± 2.4 | 95.4 ± 1.4 |
| circle2d-navigate-v0 - radius_category | Style | 99.7 ± 0.4 | 91.2 ± 7.0 | 91.3 ± 11.5 | 95.8 ± 5.6 | 99.7 ± 0.1 | 77.1 ± 16.8 |
| circle2d-navigate-v0 - radius_category | Task | 30.9 ± 9.4 | 83.0 ± 2.8 | 88.0 ± 1.8 | 16.3 ± 7.4 | 64.3 ± 8.4 | 87.1 ± 3.8 |
| circle2d-navigate-v0 - speed_category | Style | 99.6 ± 0.7 | 97.1 ± 6.3 | 99.6 ± 0.8 | 96.0 ± 4.5 | 99.2 ± 1.1 | 99.0 ± 1.8 |
| circle2d-navigate-v0 - speed_category | Task | 21.6 ± 5.0 | 89.8 ± 3.6 | 90.6 ± 3.4 | 15.3 ± 8.7 | 92.7 ± 4.5 | 95.3 ± 2.2 |
| circle2d-navigate-v0 - curvature_noise | Style | 33.3 ± 0.0 | 38.9 ± 7.9 | 35.4 ± 4.6 | 40.0 ± 6.7 | 45.8 ± 9.8 | 33.6 ± 0.7 |
| circle2d-navigate-v0 - curvature_noise | Task | 19.7 ± 7.7 | 88.8 ± 3.6 | 94.5 ± 2.1 | 9.0 ± 9.7 | 62.4 ± 7.5 | 89.9 ± 4.7 |
| halfcheetah-fix-v0 - all | Style | 53.1 ± 10.6 | 44.4 ± 6.1 | 41.3 ± 4.1 | 78.0 ± 1.8 | 78.1 ± 1.5 | 49.7 ± 5.4 |
| halfcheetah-fix-v0 - all | Task | 32.1 ± 8.4 | 72.8 ± 5.6 | 80.6 ± 3.1 | 47.6 ± 2.3 | 56.5 ± 2.5 | 76.6 ± 5.5 |
| halfcheetah-fix-v0 - speed | Style | 73.0 ± 20.3 | 31.9 ± 9.4 | 34.6 ± 2.2 | 96.0 ± 1.6 | 95.6 ± 3.1 | 37.4 ± 6.5 |
| halfcheetah-fix-v0 - speed | Task | 42.5 ± 13.2 | 72.5 ± 10.7 | 84.1 ± 2.4 | 48.1 ± 1.7 | 51.6 ± 1.9 | 87.5 ± 5.9 |
| halfcheetah-fix-v0 - angle | Style | 60.0 ± 15.5 | 41.4 ± 10.7 | 30.9 ± 2.7 | 99.1 ± 1.1 | 99.5 ± 0.1 | 69.9 ± 8.9 |
| halfcheetah-fix-v0 - angle | Task | 26.2 ± 5.3 | 68.4 ± 9.9 | 83.2 ± 4.2 | 38.0 ± 2.0 | 48.9 ± 1.9 | 68.0 ± 6.3 |
| halfcheetah-fix-v0 - torso_height | Style | 73.2 ± 8.9 | 89.7 ± 4.7 | 84.0 ± 7.9 | 96.8 ± 3.5 | 98.0 ± 1.9 | 63.8 ± 5.1 |
| halfcheetah-fix-v0 - torso_height | Task | 33.8 ± 8.9 | 73.1 ± 1.4 | 73.9 ± 1.7 | 50.3 ± 1.2 | 51.5 ± 1.0 | 68.8 ± 6.2 |
| halfcheetah-fix-v0 - backfoot_height | Style | 28.4 ± 2.8 | 34.7 ± 3.4 | 31.0 ± 4.6 | 47.5 ± 2.0 | 49.2 ± 1.2 | 37.6 ± 2.8 |
| halfcheetah-fix-v0 - backfoot_height | Task | 34.7 ± 6.6 | 85.4 ± 1.5 | 86.4 ± 1.9 | 63.1 ± 5.0 | 76.2 ± 1.6 | 82.3 ± 4.4 |
| halfcheetah-fix-v0 - frontfoot_height | Style | 30.7 ± 5.7 | 24.1 ± 2.4 | 26.0 ± 3.0 | 50.5 ± 0.8 | 48.2 ± 1.2 | 39.9 ± 3.8 |
| halfcheetah-fix-v0 - frontfoot_height | Task | 23.5 ± 7.9 | 64.4 ± 4.6 | 75.4 ± 5.3 | 38.3 ± 1.7 | 54.5 ± 5.9 | 76.3 ± 4.9 |
| halfcheetah-stitch-v0 - all | Style | 48.4 ± 12.5 | 41.1 ± 4.8 | 42.1 ± 4.9 | 78.0 ± 1.1 | 60.8 ± 6.0 | 33.8 ± 6.2 |
| halfcheetah-stitch-v0 - all | Task | 31.9 ± 10.3 | 81.3 ± 3.1 | 78.3 ± 5.6 | 47.0 ± 2.3 | 70.0 ± 6.0 | 80.4 ± 9.0 |
| halfcheetah-stitch-v0 - speed | Style | 57.1 ± 23.2 | 34.0 ± 2.3 | 38.1 ± 4.7 | 96.3 ± 0.5 | 47.6 ± 11.2 | 32.6 ± 5.2 |
| halfcheetah-stitch-v0 - speed | Task | 32.7 ± 14.3 | 83.3 ± 3.0 | 81.3 ± 5.0 | 47.2 ± 0.7 | 78.7 ± 8.5 | 84.0 ± 8.5 |
| halfcheetah-stitch-v0 - angle | Style | 55.0 ± 20.4 | 31.5 ± 3.3 | 34.7 ± 6.5 | 99.5 ± 0.2 | 92.5 ± 6.1 | 38.0 ± 6.0 |
| halfcheetah-stitch-v0 - angle | Task | 25.5 ± 8.8 | 83.4 ± 4.2 | 79.7 ± 9.7 | 41.1 ± 4.2 | 54.8 ± 6.6 | 79.7 ± 7.1 |
| halfcheetah-stitch-v0 - torso_height | Style | 71.5 ± 10.7 | 83.0 ± 10.6 | 77.7 ± 5.9 | 96.9 ± 1.4 | 85.1 ± 7.4 | 44.5 ± 8.3 |
| halfcheetah-stitch-v0 - torso_height | Task | 33.7 ± 10.9 | 74.1 ± 1.3 | 69.8 ± 4.1 | 48.3 ± 2.2 | 59.5 ± 5.5 | 82.1 ± 7.5 |
| halfcheetah-stitch-v0 - backfoot_height | Style | 28.0 ± 3.4 | 30.6 ± 5.0 | 32.0 ± 3.7 | 47.0 ± 2.4 | 39.1 ± 3.8 | 29.0 ± 6.3 |
| halfcheetah-stitch-v0 - backfoot_height | Task | 41.2 ± 9.2 | 87.0 ± 1.8 | 84.6 ± 4.5 | 60.7 ± 3.7 | 80.8 ± 6.4 | 76.2 ± 9.8 |
| halfcheetah-stitch-v0 - frontfoot_height | Style | 30.2 ± 5.0 | 26.5 ± 2.9 | 28.0 ± 3.6 | 50.3 ± 0.8 | 39.5 ± 1.3 | 24.8 ± 5.0 |
| halfcheetah-stitch-v0 - frontfoot_height | Task | 26.5 ± 8.3 | 78.5 ± 5.3 | 76.1 ± 4.9 | 37.8 ± 0.8 | 76.3 ± 3.2 | 79.8 ± 12.0 |
| halfcheetah-vary-v0 - all | Style | 46.7 ± 9.5 | 37.0 ± 3.0 | 31.1 ± 2.0 | 78.9 ± 0.7 | 77.8 ± 1.0 | 41.8 ± 5.0 |
| halfcheetah-vary-v0 - all | Task | 35.9 ± 9.0 | 79.0 ± 3.2 | 82.6 ± 3.1 | 50.6 ± 1.3 | 58.0 ± 1.7 | 84.6 ± 3.2 |
| halfcheetah-vary-v0 - speed | Style | 54.3 ± 14.3 | 33.3 ± 0.3 | 33.4 ± 0.2 | 96.7 ± 0.1 | 96.9 ± 0.4 | 40.7 ± 6.1 |
| halfcheetah-vary-v0 - speed | Task | 42.7 ± 9.3 | 88.2 ± 2.4 | 88.7 ± 2.2 | 48.1 ± 1.3 | 50.7 ± 0.9 | 84.1 ± 5.2 |
| halfcheetah-vary-v0 - angle | Style | 39.7 ± 10.8 | 32.9 ± 4.2 | 31.8 ± 2.0 | 99.2 ± 0.6 | 98.7 ± 1.8 | 44.3 ± 5.2 |
| halfcheetah-vary-v0 - angle | Task | 19.0 ± 7.4 | 83.1 ± 3.6 | 84.7 ± 2.3 | 48.0 ± 2.1 | 55.3 ± 1.1 | 84.8 ± 3.0 |
| halfcheetah-vary-v0 - torso_height | Style | 77.0 ± 11.8 | 60.7 ± 4.1 | 36.9 ± 3.2 | 98.8 ± 0.3 | 98.8 ± 0.3 | 59.3 ± 7.1 |
| halfcheetah-vary-v0 - torso_height | Task | 37.3 ± 11.7 | 68.2 ± 2.9 | 74.0 ± 3.0 | 50.5 ± 0.5 | 50.9 ± 1.3 | 87.2 ± 1.9 |
| halfcheetah-vary-v0 - backfoot_height | Style | 31.8 ± 5.3 | 32.8 ± 3.8 | 27.4 ± 3.5 | 49.5 ± 1.4 | 45.7 ± 1.2 | 28.2 ± 2.9 |
| halfcheetah-vary-v0 - backfoot_height | Task | 48.1 ± 7.5 | 80.3 ± 2.9 | 82.6 ± 4.7 | 69.0 ± 1.7 | 75.0 ± 1.8 | 87.9 ± 1.6 |
| halfcheetah-vary-v0 - frontfoot_height | Style | 30.6 ± 5.3 | 25.4 ± 2.8 | 25.9 ± 1.3 | 50.4 ± 1.0 | 48.7 ± 1.2 | 36.5 ± 3.6 |
| halfcheetah-vary-v0 - frontfoot_height | Task | 32.4 ± 8.9 | 75.4 ± 4.0 | 83.0 ± 3.1 | 37.5 ± 1.1 | 58.0 ± 3.2 | 79.0 ± 4.3 |
| humenv-simple-v0 - head_height | Style | 79.4 ± 26.9 | 99.1 ± 0.9 | 99.4 ± 0.4 | 99.6 ± 0.0 | 99.6 ± 0.1 | 99.5 ± 0.2 |
| humenv-simple-v0 - head_height | Task | 14.6 ± 14.5 | 16.0 ± 7.5 | 20.0 ± 12.5 | 19.1 ± 7.1 | 31.7 ± 4.8 | 36.5 ± 0.4 |
| humenv-simple-v0 - all | Style | 79.4 ± 26.9 | 99.1 ± 0.9 | 99.4 ± 0.4 | 99.6 ± 0.0 | 99.6 ± 0.1 | 99.5 ± 0.2 |
| humenv-simple-v0 - all | Task | 14.6 ± 14.5 | 16.0 ± 7.5 | 20.0 ± 12.5 | 19.1 ± 7.1 | 31.7 ± 4.8 | 36.5 ± 0.4 |
| humenv-complex-v0 - speed | Style | 34.3 ± 4.7 | 28.8 ± 8.4 | 22.6 ± 11.1 | 83.7 ± 5.9 | 91.6 ± 8.9 | 33.3 ± 3.7 |
| humenv-complex-v0 - speed | Task | 5.7 ± 1.6 | 39.7 ± 5.2 | 33.6 ± 8.2 | 12.0 ± 1.6 | 16.2 ± 2.3 | 40.0 ± 2.6 |
| humenv-complex-v0 - head_height | Style | 61.1 ± 9.2 | 22.0 ± 13.6 | 24.3 ± 18.9 | 83.3 ± 6.6 | 90.1 ± 9.3 | 33.3 ± 4.9 |
| humenv-complex-v0 - head_height | Task | 4.5 ± 3.8 | 19.6 ± 5.3 | 20.5 ± 9.3 | 10.0 ± 2.8 | 15.7 ± 2.6 | 41.9 ± 3.8 |
| humenv-complex-v0 - all | Style | 47.7 ± 6.9 | 25.4 ± 11.0 | 23.5 ± 15.0 | 83.5 ± 6.2 | 90.8 ± 9.1 | 33.3 ± 4.3 |
| humenv-complex-v0 - all | Task | 5.1 ± 2.7 | 29.7 ± 5.2 | 27.1 ± 8.8 | 11.0 ± 2.2 | 15.9 ± 2.5 | 41.0 ± 3.2 |

## E ABLATIONS

### E.1 HOW DO WE NEED TO ESTIMATE $p(z|s,a)$?

Estimating $p(z|s,a)$ relates to estimating the correspondence between a state-action pair and a style which and is a key component of our problematic. We tested for this purpose four distinct strategies to form an estimator $\chi(s,a,z)$ of $p(z|s,a)$. A first strategy noted **ind** consists in taking as the estimator the indicator of $\{z = z_c\}$ with $z_c$ the associated label of $(s,a)$ within $\lambda(\mathcal{D})$:

$$\forall(s,a,z_c) \in \lambda(\mathcal{D}), \chi_{\mathrm{ind}}(s,a,z) = \chi_{\mathrm{ind}}(z_c,z) = \mathbb{1}(z = z_c) \tag{53}$$

As $\lambda$ can attribute several labels to $(s,a)$ within $\mathcal{D}$, we can state that:

$$\forall(s,a) \in \mathcal{D}, \mathbb{E}_{z_c \sim p_c^{\lambda(\mathcal{D})}(z|s,a)}[\chi_{\mathrm{ind}}(z_c,z)] = \mathbb{E}_{z_c \sim p_c^{\lambda(\mathcal{D})}(z|s,a)}[\mathbb{1}(z=z_c)] \approx p(z|s,a) \tag{54}$$

as the expectation of an indicator variable is the probability of its associated event. Hence, using $\chi_{\mathrm{ind}}$ can be justified when relying on a sufficient number of samples during training.

Another approach noted **MINE** is to use the MINE estimator described in Appendix **??** to estimate:

$$T^*(s,a,z) = \log \frac{p(s,a,z)}{p(s,a)p(z)} = \log \frac{p(z|s,a)}{p(z)} \tag{55}$$

by optimizing:

$$J_{\mathrm{MINE}}(T) = \mathbb{E}_{(s,a) \sim p^{\lambda(\mathcal{D})}(s,a), z \sim p_c^{\lambda(\mathcal{D})}(z|s,a)}[T(s,a,z)] - \log\left(\mathbb{E}_{(s,a) \sim p^{\lambda(\mathcal{D})}(s,a), z \sim p_r^{\mathcal{D}}(z)}\left[e^{T(s,a,z)}\right]\right) \tag{56}$$

and taking:

$$\chi_{\mathrm{MINE}}(s,a,z) = p_r^{\mathcal{D}}(z)e^{T(s,a,z)} \tag{57}$$

$$\approx p_r^{\mathcal{D}}(z)e^{\log \frac{p(z|s,a)}{p(z)}} \tag{58}$$

$$\approx p_r^{\mathcal{D}}(z)\frac{p(z|s,a)}{p(z)} \tag{59}$$

$$\approx p(z|s,a) \tag{60}$$

Also, as we seek to approximate $p(z|s,a) \in [0,1]$ with discrete labels, we propose to train directly a neural network $\chi(s,a,z)$ within the MINE objective, taking $p_r^{\lambda(\mathcal{D})}(z)$ as an approximation of $p(z)$:

$$J_{\mathrm{MINE}}(\chi) = \mathbb{E}_{(s,a) \sim p^{\lambda(\mathcal{D})}(s,a), z \sim p_c^{\lambda(\mathcal{D})}(z|s,a)}[\log \frac{\chi(s,a,z)}{p_r^{\lambda(\mathcal{D})}(z)}] - \log\left(\mathbb{E}_{(s,a) \sim p^{\lambda(\mathcal{D})}(s,a), z \sim p_r^{\lambda(\mathcal{D})}(z)}\left[e^{\log \frac{\chi(s,a,z)}{p_r^{\lambda(\mathcal{D})}(z)}}\right]\right) \tag{61}$$

with $\chi$'s output activations taken as a sigmoid and a softmax to define the **sigmoid** and **softmax** strategies respectively. We evaluate the impact of each strategy on style alignment and report the results in Table 6 and Figure 17. For SORL, both **MINE** and **softmax** achieve the best performance, while for SCIQL the best results are obtained with **ind** and **softmax**. Accordingly, in our experiments we adopt **softmax** for SORL and **ind** for SCIQL.

Table 6: **Style alignments for different** $p(z|s,a)$ **estimation strategies.**

| Dataset | SORL (ind) | SORL (MINE) | SORL (sigmoid) | SORL (softmax) | SCIQL (ind) | SCIQL (MINE) | SCIQL (sigmoid) | SCIQL (softmax) |
|---|---|---|---|---|---|---|---|---|
| mujoco_halfcheetah-fix | 30.3 ± 3.4 | 52.6 ± 12.4 | 44.0 ± 11.7 | 53.1 ± 10.6 | 78.0 ± 1.8 | 67.4 ± 8.1 | 69.0 ± 7.1 | 77.9 ± 1.1 |
| mujoco_halfcheetah-stitch | 30.0 ± 4.5 | 52.7 ± 10.8 | 43.0 ± 10.7 | 48.4 ± 12.5 | 78.0 ± 1.1 | 67.4 ± 8.0 | 69.5 ± 6.0 | 77.8 ± 1.5 |
| mujoco_halfcheetah-vary | 29.7 ± 4.3 | 47.0 ± 10.1 | 42.7 ± 11.5 | 46.7 ± 9.5 | 78.9 ± 0.7 | 73.6 ± 5.7 | 67.1 ± 6.4 | 78.8 ± 0.9 |
| random_circles-inplace-v0 | 29.4 ± 3.5 | 59.1 ± 2.7 | 46.6 ± 11.9 | 58.9 ± 2.7 | 74.7 ± 9.3 | 74.3 ± 2.0 | 53.6 ± 19.8 | 73.7 ± 7.7 |
| random_circles-navigate-v0 | 29.1 ± 6.1 | 59.9 ± 3.2 | 46.9 ± 8.7 | 60.0 ± 3.3 | 75.5 ± 4.7 | 75.5 ± 4.6 | 62.1 ± 12.9 | 75.4 ± 4.3 |
| all_datasets | 29.8 ± 4.0 | **53.8 ± 8.6** | 44.9 ± 11.3 | 53.2 ± 8.5 | **77.2 ± 3.2** | 70.9 ± 6.0 | 64.9 ± 10.1 | **76.9 ± 2.8** |

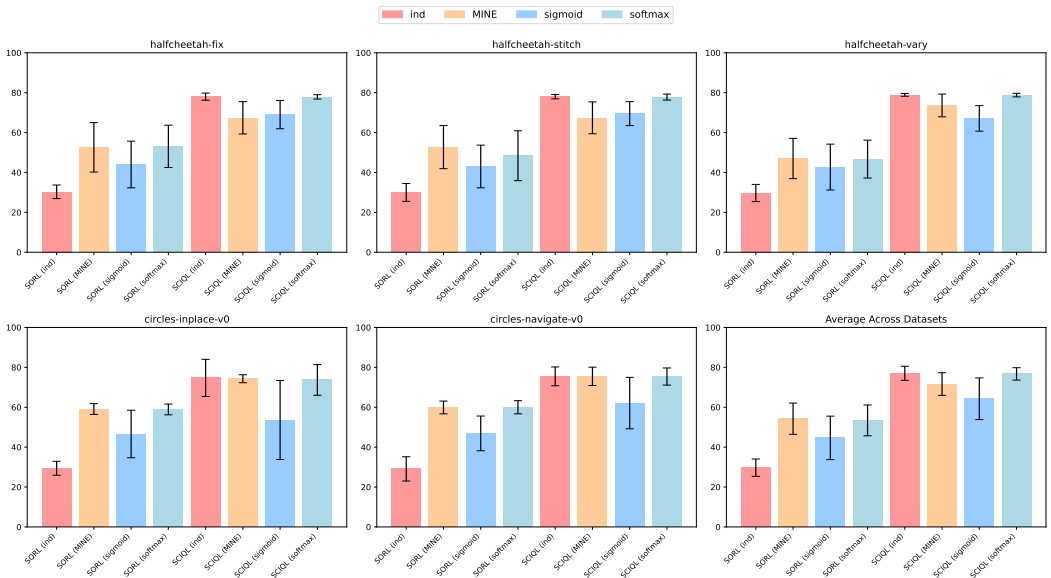

Figure 17: **Style alignments histograms for different** $p(z|s, a)$ **estimation strategies.**

## E.2 What is the impact of the choice of $p_{\mathrm{m}}^{\lambda(\mathcal{D})}$?

To address the lower performance of SCIQL on the **turn_direction**, **radius**, and **speed** criteria of Circle2d, we evaluated SCIQL by sampling styles from $p_{\mathrm{c}}^{\lambda(\mathcal{D})}$ rather than $p_{\mathrm{r}}^{\lambda(\mathcal{D})}$. As shown in the histogram in Figure 18, using $p_{\mathrm{c}}^{\lambda(\mathcal{D})}$ improves style alignment to its maximum score, highlighting both SCIQL's flexibility in varying its style sampling distributions and the potential importance of this choice when optimizing style alignment.

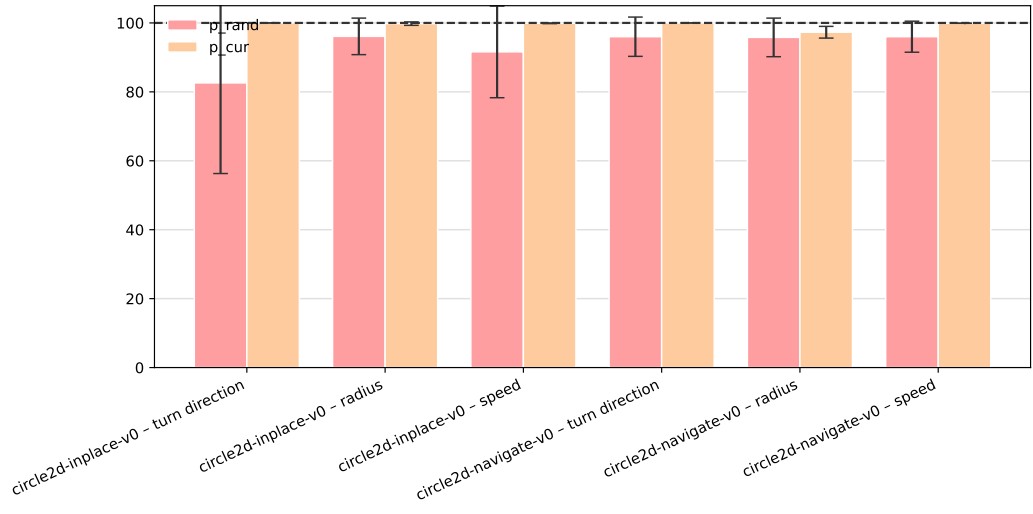

Figure 18: **SCIQL performance under $p_{\mathrm{r}}^{\lambda(\mathcal{D})}$ vs $p_{\mathrm{c}}^{\lambda(\mathcal{D})}$?**

### E.3 How robust is SCIQL to imperfect style annotations?

While relying on labeling functions allows for explainable and precise style annotations, style annotations could in practice be imperfect due to the noisiness of domain experts. For instance, alternative labeling approaches such as human generated labels or VLMs could provide noisy labels due to biases, stochasticity and unclear cuts between style transitions. All those imperfections can have an important impact on style alignment. Hence, to measure the robustness of SCIQL in comparison to the baselines, we simulate labeling imperfections by modifying the labeling procedure such that for a given criterion $\lambda$, each state-action-style triplet $(s_t, a_t, z_t)$ of $\lambda(\mathcal{D})$ is polluted with a probability $\zeta$ by changing its label $z_t$ to another label $\tilde{z}_t$ sampled uniformly among other available labels of $\mathcal{L}(\lambda)$. We plot in Figure 19 the evolution of the style alignment of the different baselines for the halfcheetah-fix-v0 - speed in Subfigure 19a, halfcheetah-fix-v0 - angle tasks in Subfigure 19b and the average of those evolutions as halfcheetah-fix-v0 - speed + angle in Subfigure 19c.

First, for noise levels going from $0.0$ to $0.6$, we see that SCIQL maintains a very good style alignment. More precisely, SCIQL is on average (i.e. in speed_label + angle_label) better aligned with a noise level of $0.6$ than all of the other baselines with no noise. The other baselines lose all their alignment even for small noise levels such as $0.2$, obtaining style alignments equal to BC's, which means that the baselines consider any noisy label as uninformative noise and ignore them, losing all conditioning capabilities. This shows that **SCIQL is significantly more robust to label noise than any test baseline**, highlighting the benefits of integrating RL signals to style alignment training.

Second, above a certain noise threshold $\bar{\zeta}$, we see that SCIQL's alignment plummets towards $0$, which is infact a good feature. A possible intuition is that this threshold corresponds to the noise level above which the true labeling of each state-action pair is no longer majoritary in the noisy dataset. Beyond this threshold, for SCIQL, **the best outcome for alignment is to reach wrong labels**. Indeed, for each state-action pair $(s, a)$, the probability of labeling to the right label $z$ is $p_{\text{right}} = 1 - \zeta$, while the probability of choosing a wrong label is $p_{\text{wrong}} = \zeta$. Since wrong labels are sampled uniformly, each individual wrong label $\tilde{z}_i \in Z_{\text{wrong}} = \mathcal{L}(\lambda) \backslash \{z\}$ has a probability $p_i = \frac{\zeta}{|\lambda| - 1}$ to be selected, $|\lambda|$ being the total numbers of labels in $\mathcal{L}(\lambda)$. Consequently, for the right label to maintain the majority position, the threshold needs to verify:

$$\forall \tilde{z}_i \in Z_{\text{wrong}}, p_{\text{true}} > p_i \Leftrightarrow p_{\text{true}} > \max_{\tilde{z}_i \in Z_{\text{wrong}}} p_i \Leftrightarrow 1 - \zeta > \frac{\zeta}{|\lambda| - 1} \Leftrightarrow \frac{|\lambda| - 1}{|\lambda|} > \zeta \qquad (62)$$

Also, as described in Appendix A, both speed and angle criteria have the same number of $|\lambda| = 3$ labels each and as such, for both labels $\bar{\zeta} = \frac{|\lambda| - 1}{|\lambda|} = \frac{2}{3}$, which **corresponds to the observed threshold and consequently supports our intuition**.

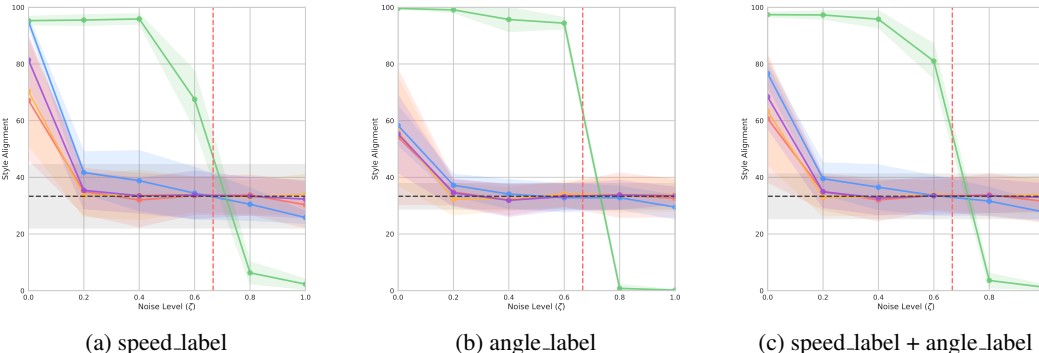

(a) speed_label        (b) angle_label        (c) speed_label + angle_label

Figure 19: **Evolution of style alignment under noisy labels.** For noise labels $\zeta \in \{0.0, 0.1, ..., 1.0\}$, we compare the evolution of style alignment of **BC** (--), **CBC** (-•-), **BC-PMI** (-•-), **SCBC** (-•-), **SORL** (-•-) and **SCIQL** (-•-). We see that SCIQL maintains an overall better alignment before the noise threshold (vertical --) where the true label is majoritary, and then misaligns itself beyond the noise treshold, which corresponds to following intentionally the wrong styles accordingly to the noisy labeling.

