# OpenReview forum: "Offline Reinforcement Learning of High-Quality Behaviors Under Robust Style Alignment"
_ICLR.cc/2026/Conference — Submitted to ICLR 2026_

### Official Review · Reviewer_re3p · 2025-10-25

**Soundness:** 3
**Presentation:** 2
**Contribution:** 2
**Rating:** 6
**Confidence:** 4

**Summary:**

This paper proposes a method for style alignment in the offline RL setting using implicit Q learning and advantage weighted regression. Styles are defined using hard coded functions which is then used as a reward to learn a style value function. This value function is combined with a task value function (independent of style) to train a style aligned policy. Experiments are conducted on the circle and halfcheetah environments, showing significant performance advantage over baselines such as SORL and SCBC. Ablation experiments demonstrate how different temperature parameters prioritize task performance and style alignment.

**Strengths:**

* The proposed solution is quite simple and sound.
* The effectiveness of the proposed method is clear.

**Weaknesses:**

* I think the presentation can be improved if the authors moved some of the plots in the appendix to the main paper.
* Some details in the method can be better explained.
* I find the need to tune the temperature parameter and its sensitivity a downside of the proposed method.

**Questions:**

* In (12), can you add a text explanation of the equation? Is the gating saying that if the style advantage is high enough such that the sigmoid output is 1 then you can incorporate task advantage? In theory the advantage function at optimality is zero $\max_{a}Q(s, a) = V(s)$, the sigmoid output is 0.5, so you are still using a small weight on the task reward advantage.
* In the results, you did not include an in-depth explanation of the different datasets. Can you explain how you expect the method to behave differently for different datasets? From Table 1, it looks like halfcheetah-vary performs worse on the baseline methods than the other halfcheetah datasets. Why?
* Can you comment on the sensitivity of the temperature parameters?
* I would suggest moving some of the plots in the appendix to the main paper so that people understand what style means.
* (Minor) there are a lot of typos in the paper. Please fix.

---

> ### Author Response · Authors · 2025-11-21
> **Answer to Reviewer re3p (1)**
>
> We thank **R-re3p** for the review and constructive feedback. We appreciate the recognition of the simplicity, soundness, and effectiveness of our methods.
>
> > I think the presentation can be improved if the authors moved some of the plots in the appendix to the main paper.
> > I would suggest moving some of the plots in the appendix to the main paper so that people understand what style means.
>
> We thank **R-re3p** for this suggestion. To address it to the best of our ability, and given space constraints, we added a reference on l.191 to the descriptive section presenting the environments, datasets, and labeling functions, so that readers can more easily visualize and understand our labels. To further improve clarity, we now also updated the project website, showcasing videos of all styles from the dataset (bottom part).
>
> > Some details in the method can be better explained.
>
> > Is the gating saying that if the style advantage is high enough such that the sigmoid output is 1 then you can incorporate task advantage?
>
> This statement is exact. The goal of the gating mechanism is to increase task performance while mitigating the drop in style alignment. Using the sigmoid of the style advantage as gating ensures that the task-advantage term is incorporated only when the style advantage is positive. This effectively filters the $(s,a,z)$ triplets to keep only those that are style-aligned.
>
> > In theory the advantage function at optimality is zero , the sigmoid output is $0.5$, so you are still using a small weight on the task reward advantage.
>
> We agree with the reviewer that, in theory, if one uses the optimal advantage, then for any $(s,z)$, sampling actions from the dataset yields $A(s,a,z)≤0$ for all actions. In this case, the sigmoid would indeed saturate at best around $0.5$.
>
> However, we do not use the optimal advantage. Instead, we rely on the advantage estimated by the IQL [1] algorithm, which corresponds to the value functions of the behavior policy (the policy underlying the dataset), biased toward high-value states through expectile regression. This biased advantage has been shown in the literature to provide substantial improvements over the behavior-policy advantage that can be learned through Monte Carlo rollouts, as done in the AWR [2] paper.
>
> In practice, during training we observe that the minimum and maximum of the advantages are indeed negative and positive respectively. Moreover, these advantages span a range much larger than the variation range of the sigmoid, enabling a clear filtering of extreme values and yielding smoother outputs only for ambiguous $(s,a,z)$ pairs (i.e., when advantages are close). Finally, although not strictly necessary, our training pipeline allows for  normalizing the advantages using a moving average, which can help rescaling advantages in some datasets that would otherwise have a too small magnitude.
>
> > I find the need to tune the temperature parameter and its sensitivity a downside of the proposed method.
> > Can you comment on the sensitivity of the temperature parameters?
>
> Both SORL and SCIQL have a temperature parameter. However, they serve different purposes.
>  - SCIQL’s temperature parameter $β_{SCIQL}$ does not encode the style alignment and task performance tradeoff, but corresponds to a hyperparameter inherited from IQL’s temperature parameter $β_{IQL}$, while the tradeoff is tackled by the Gated Advantage Weighted Regression. We did not perform hyperparameter search on $β_{SCIQL}$, and just reused the $β_{IQL}$ used by IQL authors [1] in their halfcheetah experiments. Besides, we also reused these hyperparameters for our circle-2d experiments to attain best performance for both environments.
>  - In SORL, however, the temperature parameter encodes the tradeoff between style alignment and task performance. A low $β_{SORL}$ encourages style alignment while a high $β_{SORL}$ benefits to task performance. We added an analysis of the impact of SORL’s temperature parameter in Appendix D. We observe that selecting an appropriate temperature $β_{SORL}$ for SORL is challenging, as it requires a delicate balance between style alignment and task performance and depends on the dataset.
>
> Consequently, in contrast to SORL, SCIQL maintains by design a strong alignment while substantially improving task performance, without requiring fine-grained hyperparameter tuning of its temperature.

---

> ### Author Response · Authors · 2025-11-21
> **Answer to Reviewer re3p (2)**
>
> > In the results, you did not include an in-depth explanation of the different datasets. Can you explain how you expect the method to behave differently for different datasets? From Table 1, it looks like halfcheetah-vary performs worse on the baseline methods than the other halfcheetah datasets. Why?
>
> We acknowledge that providing more granular detail on dataset-specific behaviors would strengthen the analysis, as our original text focused primarily on SCIQL's aggregate superiority.
>
> We expect SCIQL to show the largest gains on datasets requiring **trajectory stitching** (incomplete trajectories) or **long-horizon planning** (where styles are state-dependent, like position labels), as SCIQL actively navigates toward aligned states while baselines like BCPMI and SORL lack these mechanisms. We also believe that this planning capability can make our method more robust to environment stochasticity or compounding errors which can both shift the policy outside of aligned states at inference time. Regarding halfcheetah-vary-v0, where styles change along the trajectory, we observe a significant performance drop for baselines (CBC shows high variance, BCPMI, SORL, and SCBC degrade in alignment). This is likely because varying styles induce noisy learning signals that impair imitation-based approaches. SCIQL’s robustness here underscores the benefit of value learning, which effectively enhances robustness to maintaining consistent alignment. We have expanded Subsection 5.2 and Appendix D to include this detailed breakdown.
>
> We thank **R-re3p** for pointing out this gap in our experimental section. We now feature a similar analysis in the updated manuscript in Subsection 5.2, with an additional link to a more detailed analysis in the Appendix D.
>
> > (Minor) there are a lot of typos in the paper. Please fix.
>
> We acknowledge the reviewer remark and have addressed it by correcting several typos in the revised version of the paper.
>
> [1] Kostrikov & al., Offline Reinforcement Learning with Implicit Q-Learning, ICLR 2022.
>
> [2] Peng & al., Advantage-Weighted Regression: Simple and Scalable Off-Policy Reinforcement Learning, Arxiv 2019.
>
> [3] Park & al., HIQL: Offline Goal-Conditioned RL with Latent States as Actions, Neurips 2023.
>
> [4] Park & al., OGBench: Benchmarking Offline Goal-Conditioned RL, ICLR 2025.

---

> > ### Comment · Reviewer_re3p · 2025-11-26
> >
> > Thank the authors for addressing my questions and highlighting their changes in blue which made it easy to review. I have no other questions.

---

### Official Review · Reviewer_KRET · 2025-10-27

**Soundness:** 2
**Presentation:** 2
**Contribution:** 2
**Rating:** 2
**Confidence:** 3

**Summary:**

This paper proposes SCIQL, an offline reinforcement learning algorithm designed to learn policies that optimize task reward while exhibit specific behavioral styles. Building upon IQL, SCIQL extends it to the style conditioned setting and introduces GAWR mechanism to balance the two advantage terms.

**Strengths:**

The proposed GAWR mechanism and sub-trajectory labeling provide a simple yet effective way to integrate style supervision into offline RL. Empirical results on Circle2D and HalfCheetah environments show that SCIQL consistently achieves higher style alignment scores compared to the baselines.

**Weaknesses:**

1. The problem formulation is conceptually unclear. If style alignment and task reward are inherently conflicting, the object should be to balance the trade-off between the two. However, the current formulation seems to sacrifice task reward to increase style conformity, which raises the question of whether this trad-off is explicitly modeled.

2. Given that style alignment and task reward clearly conflict as shown in Section 5.3, the evaluation might be better framed in a Pareto optimality context rather than using single averaged metrics. Without such discussion, it is difficult to interpret whether improving style alignment at the cost of lowered reward constitute genuine progress.

3. The paper defines style labels as discrete categories obtained via predefined labeling functions. Could the authors clarify why a discrete formulation was chosen over a continuous style ? Using continuous representations might allow smoother interpolation between styles and potentially improve generalization to unseen or mixed style combinations.

4. The evaluation is restricted to toy circle 2d and halfcheetah environments., which are relatively simple and low-dimensional. It would strengthen the work to include results on more diverse environments, such as other MuJoCo or Atari tasks or humanoids-tyle control demands where stylistic variations are more naturally expressed.

5. It would be valuable to assess whether the proposed method can extrapolate (or interpolate) to unseen style labels or novel combinations of style labels that were not encountered during training.

**Questions:**

1. Is z a multi-dimensional vector aggregating multiple criterion-specific labels, or a single discrete label ? If it is the former, the description around lines 180-190 should be revised to clarify how multiple criterion labels are annotated and used in z.

2. Minor typos. Line 453, Twhile -> while. Line 169, " is reversed.

---

> ### Author Response · Authors · 2025-11-21
> **Answer to Reviewer KRET (1)**
>
> We thank **R-KRET** for the review and constructive feedback. We appreciate that the efficient integration of style supervision and the higher performance of our approach is considered valuable.
>
> > The problem formulation is conceptually unclear. If style alignment and task reward are inherently conflicting, the object should be to balance the trade-off between the two. However, the current formulation seems to sacrifice task reward to increase style conformity, which raises the question of whether this trade-off is explicitly modeled.
>
> Our work operates within the framework of stylized offline RL, building on baselines like BCPMI [1] and SORL [2]. We address two distinct objectives: (1) maximizing **style alignment** (robustly matching a target behavior), and (2) **style-conditioned task optimization** (maximizing reward within that style). We view the first objective as a self-contained Style-Conditioned RL problem, where we achieve superior alignment compared to BCPMI (Table 1). However, acknowledging that style constraints can hinder task performance, we introduce the GAWR mechanism specifically to optimize the second objective of balancing the trade-off more effectively than SORL by filtering detrimental transitions while preserving style conformance (Table 2).
>
> Thus, while our work is related to the literature on Multi-Objective Reinforcement Learning (MORL) which aims to adapt policy behavior to user preferences towards multiple and potentially conflicting objectives, our context differs from the MORL context because of the asymmetric role played by our two metrics of style alignment and task performance. Our goal is **not to find a trade-off between alignment and performance**, but to **optimize task performance under the constraint of preserving the style alignment** of our stylized policies. To illustrate this, Table 2 shows that across our experiments, going from SCIQL(λ) to SCIQL (λ > r) always increases the task performance, while significantly decreasing the style alignment only in halfcheetah-stitch-v0. This is the reason for our choice to not frame our setting in the context of Pareto optimality in the first place, although we agree such visualizations can provide useful comparative information (see next answer).
>
> > Given that style alignment and task reward clearly conflict as shown in Section 5.3, the evaluation might be better framed in a Pareto optimality context rather than using single averaged metrics. Without such discussion, it is difficult to interpret whether improving style alignment at the cost of lowered reward constitutes genuine progress.
>
> Despite our different positioning than MORL, we understand the interest of **R-KRET** to visualize such style-rewards tradeoffs. As such, we computed the Pareto fronts of both SORL and SCIQL and displayed the results in the updated manuscript in Figure 2. We additionally computed the following Hypervolumes:
>
>
> | Dataset            | SORL   | SCIQL  | improvement |
> | ------------------ | ------ | ------ | ----------- |
> | circle2d-inplace   | 4082.3 | 5913.5 | +44.9%      |
> | circle2d-navigate  | 4252.4 | 6283.4 | +47.8%      |
> | halfcheetah-fix    | 3833.4 | 5411.6 | +41.2%      |
> | halfcheetah-stitch | 3620.7 | 5415.9 | +49.6%      |
> | halfcheetah-vary   | 3383.2 | 5679.9 | +67.9%      |
>
>
> In the additional figures, we see that SCIQL’s front largely dominates SORL’s, meaning that SCIQL ensures better tradeoffs between task performance and style alignment across all environments. Also, Figure 2 showcases the preservation of style alignment from SCIQL when going from SCIQL(λ) to SCIQL (λ > r).

---

> ### Author Response · Authors · 2025-11-21
> **Answer to Reviewer KRET (2)**
>
> > The paper defines style labels as discrete categories obtained via predefined labeling functions. Could the authors clarify why a discrete formulation was chosen over a continuous style ? Using continuous representations might allow smoother interpolation between styles and potentially improve generalization to unseen or mixed style combinations.
>
> We choose discrete categories because they allow us to represent both styles derived from low-level concepts, such as the height of the cheetah or the radius of a circle, notably through discretization, as well as more conceptual styles, such as archetypes of gameplay behavior (aggressive play, passive play) or manners of movement (joyfully, or seriously). This enables our framework to extend, in potential future work, to different forms of labeling such as human annotation or VLM-based labeling. Moreover, discrete labeling allows for easier learning of the estimators $p(z∣s,a)$, as well as a simpler and clearer definition of alignment compared to the continuous case. Nonetheless, we consider the extension to the continuous setting as an interesting direction for future work, and we thank **R-KRET** for this remark.
>
> > The evaluation is restricted to toy circle 2d and halfcheetah environments., which are relatively simple and low-dimensional. It would strengthen the work to include results on more diverse environments, such as other MuJoCo or Atari tasks or humanoids-style control demands where stylistic variations are more naturally expressed.
>
> While higher-dimensional evaluations are undeniably valuable, our selected environments offer the advantage of precise and reproducible measurements of both style alignment and task performance. We argue that these controlled experiments constitute strong evidence of SCIQL’s capabilities prior to scaling to more complex, real-world systems in future work. That said, we recognize **R-KRET**’s concerns and are already conducting experiments in additional environments, which we aim to share before the end of the rebuttal period.
>
> > Is z a multi-dimensional vector aggregating multiple criterion-specific labels, or a single discrete label ? If it is the former, the description around lines 180-190 should be revised to clarify how multiple criterion labels are annotated and used in z.
>
> As this work aims to improve on the previous work of BCPMI [1] and SORL [2] in style alignment and style conditioned task performance optimization, we placed ourselves within the same context of a style defined as a single discrete label.
>
> > It would be valuable to assess whether the proposed method can extrapolate (or interpolate) to unseen style labels or novel combinations of style labels that were not encountered during training.
>
> While we agree that the extrapolation to unseen style labels or novel combinations is a very exciting research direction, it lies beyond the scope of this work which focuses on improving on the line of work of SORL [2] and BCPMI [1] on single discrete style labels conditioning, while extrapolation requires continuous labels. Nevertheless, unlike previous work, our method can generalize to unseen state-style combinations thanks to the style relabeling allowed by the value learning component of SCIQL.
>
> > Minor typos. Line 453, Twhile -> while. Line 169, " is reversed.
>
> We thank **R-KRET** for notifying this typos which has been corrected.
>
> [1] Yang, Yao & al., Diverse Policies Recovering via Pointwise Mutual Information Weighted Imitation Learning, ICLR 2025.
>
> [2] Mao & al., Stylized Offline Reinforcement Learning: Extracting Diverse High-Quality Behaviors from Heterogeneous Datasets, ICLR 2024.

---

> ### Author Response · Authors · 2025-11-28
> **Answer to Reviewer KRET (3)**
>
> > The evaluation is restricted to toy circle 2d and halfcheetah environments., which are relatively simple and low-dimensional. It would strengthen the work to include results on more diverse environments, such as other MuJoCo or Atari tasks or humanoids-style control demands where stylistic variations are more naturally expressed.
>
> We are pleased to announce the addition of a new set of experiments on a higher-dimensional humanoid robot control task following the reviewers suggestions. We invite **R-KRET** to refer to the **Additional answer to all reviewers** section of the answers for more details about those experiments.

---

> > ### Author Response · Authors · 2025-12-03
> > **Answer to Reviewer KRET (4)**
> >
> > > The evaluation is restricted to toy circle 2d and halfcheetah environments., which are relatively simple and low-dimensional. It would strengthen the work to include results on more diverse environments, such as other MuJoCo or Atari tasks or humanoids-style control demands where stylistic variations are more naturally expressed.
> >
> > On top of the aforementioned additional experiments, we are pleased to announce the addition of a new set of experiments on a more challenging version of our higher-dimensional humanoid robot control task. We invite **R-KRET** to refer to the **Final answer to all reviewers** section of the answers for more details about those experiments.

---

### Official Review · Reviewer_uFXs · 2025-11-01

**Soundness:** 3
**Presentation:** 2
**Contribution:** 3
**Rating:** 6
**Confidence:** 3

**Summary:**

This paper propose a new view of the stylized policy learning problem as a generalization of the goalconditioned RL and introduce SCIQL algorithm which uses hindsight relabeling and Gated Advantage Weighted Regression mechanism to optimize task performance.

**Strengths:**

This paper provides a unified formulation of behavioral style learning via programmatic sub-trajectory labeling, and introduces the SCIQL+GAWR framework that effectively balances style alignment and task performance in the offline RL setting.

**Weaknesses:**

The reliance on hand-crafted style labeling functions constrains scalability to more abstract or subtle styles, and may require domain expertise when applied to complex environments. The algorithmic pipeline is relatively intricate, increasing implementation burden, and evidence on large-scale real-world or high-dimensional robotic systems remains limited

**Questions:**

The proposed approach relies on hand-crafted sub-trajectory labeling functions; how scalable and generalizable is this design to tasks where styles are abstract, high-level, or difficult to encode programmatically?

While the method demonstrates strong performance in simulated benchmarks, there is no evaluation on real-world systems or higher-dimensional robot control tasks. Can the authors comment on the expected practicality and robustness of SCIQL in real settings?

The overall pipeline introduces multiple components and optimization stages; how sensitive is the method to hyperparameters, and can the authors provide an ablation isolating the contributions of each module to ensure that improvements are not due to increased model complexity?

The approach assumes accurate style labels from the labeling functions. How does performance degrade under noisy or imperfect style annotations, and can the method handle ambiguous or overlapping style categories?

The paper positions programmatic style labeling as scalable, but could the authors discuss potential avenues for extending the framework to automatically learn style representations, or integrate human feedback when labeling heuristics are insufficient?

---

> ### Author Response · Authors · 2025-11-21
> **Answer to Reviewer uFXs (1)**
>
> We thank **R-uFXs** for their review and constructive feedback. We were pleased to see that **R-uFXs** judged our method to be effective at balancing style alignment and task performance.
>
> ## On labeling functions and SCIQL's scalability
>
> > The reliance on hand-crafted style labeling functions constrains scalability to more abstract or subtle styles, and may require domain expertise when applied to complex environments.
>
> We agree that hand-crafted functions have limitations regarding abstract styles. However, **SCIQL is agnostic to the source of the labels**. We utilized programmatic functions because they are accurate, cheap, and interpretable, facilitating rigorous benchmarking. Furthermore, implementing these functions is often easier than reward engineering, as it avoids the complexities of reward shaping. For applications requiring abstract or subtle styles, our framework can seamlessly integrate labels from other sources, such as human annotators, VLMs, or learned classifiers, without modification.
>
> > The proposed approach relies on hand-crafted sub-trajectory labeling functions; how scalable and generalizable is this design to tasks where styles are abstract, high-level, or difficult to encode programmatically?
>
> To scale to more abstract styles, one can leverage the flexibility of the labeling framework in two ways: (1) Composition: High-level personas (e.g., "aggressive") can be constructed by composing sets of low-level labeling functions, (2) Alternative Sources: As noted above, SCIQL can ingest labels from VLMs or human feedback to capture semantics that are difficult to code programmatically. We believe that such directions constitute interesting future work to our method.
>
> > The paper positions programmatic style labeling as scalable, but could the authors discuss potential avenues for extending the framework to automatically learn style representations, or integrate human feedback when labeling heuristics are insufficient?
>
> We agree with **R-uFXs** that extending this framework to unsupervised learned styles, or integrating human-feedback, corresponds to interesting avenues. Regarding human feedback, a promising avenue is **active learning**: since SCIQL is built on IQL, it supports **human-in-the-loop** finetuning frameworks such as DAgger methods [1,2,3], where human corrections during online play could be labeled and added to the replay buffer to iteratively refine the policy.

---

> ### Author Response · Authors · 2025-11-21
> **Answer to Reviewer uFXs (2)**
>
> ## On additional environments
>
> > Evidence on large-scale real-world or high-dimensional robotic systems remains limited.
> > While the method demonstrates strong performance in simulated benchmarks, there is no evaluation on real-world systems or higher-dimensional robot control tasks.
>
> While we acknowledge the value of real-world evaluations, our considered environments allow for precise, reproducible measurements of style alignment and task performance. We believe these experiments provide strong evidence of SCIQL's effectiveness before moving to more complex, real-world systems in future work. Nevertheless, we understand **R-uFXs**’s concerns and are currently running experiments on additional environments, aiming to share them before the end of the rebuttal period.
>
> > Can the authors comment on the expected practicality and robustness of SCIQL in real settings?
>
> The applicability of our method depends on two main factors: the feasibility of obtaining style labels and the ability to transfer an offline-trained policy to a real environment, a general challenge for all offline RL algorithms.
>
> First, obtaining style labels can be particularly difficult in real-world robotics due to high dimensionality and partial observability. A practical way to adapt our full pipeline, including labeling functions, is to perform data collection, labeling, and training in simulation, where rich state information is available. The learned policy can then be deployed in the real world using sim-to-real transfer techniques. This workflow is widely used beyond style learning, for instance, humanoid robotics typically rely on motion-capture tracking in simulation before deploying policies on physical robots.
>
> Second, SCIQL is well suited for deployment in real environments thanks to its foundation in IQL, which provides several beneficial robustness properties. Because SCIQL replaces the max operator in the Bellman update with expectile regression, it naturally avoids querying out-of-distribution actions and prevents value overestimation, two common pitfalls in offline RL. This design reduces sensitivity to environment stochasticity and limits distributional mismatch during inference, helping the policy maintain stable performance when transferred from offline training to real-world execution.

---

> ### Author Response · Authors · 2025-11-21
> **Answer to Reviewer uFXs (3)**
>
> ## On the algorithmic pipeline and ablations
>
> > The algorithmic pipeline is relatively intricate, increasing implementation burden.
> [...]The overall pipeline introduces multiple components and optimization stages;
>
>
> We appreciate the feedback on implementation complexity and would like to clarify that our method does not require intricate procedures. Our SCIQL method can be easily built on top of a standard goal conditioned variant of IQL [4] by:
>
>  1. **Data Labeling**: Labeling trajectories by using any preferred method.
>
>  2. **Value Learning**: In addition to IQL’s task value functions we also learn value functions (Q and V)  conditioned on styles.
>
>  3. **Policy Extraction**: Replacing the standard advantage with our Gated Advantage during the Advantage Weighted Regression (AWR) [5] step.
>
> These components can be trained simultaneously in a **single optimization loop** for efficiency. We have updated the manuscript to feature this high-level description.
>
> > how sensitive is the method to hyperparameters, and can the authors provide an ablation isolating the contributions of each module to ensure that improvements are not due to increased model complexity?
>
> **Ablations** Our comparative study in Tables 1 and 2 acts as an additive ablation study, explicitly isolating the contribution of core mechanisms of SCIQL:
>  - **Trajectory Labeling (BC → CBC)**: The performance improvement from BC to CBC in Table 1 isolates the benefit of basic style conditioning.
>  - **Style Relabeling (CBC → SCBC)**: SCBC is an Imitation Learning variant of our method that uses hindsight relabeling (using future styles) but no value functions. The performance gap between CBC and SCBC isolates the benefit of relabeling for trajectory stitching.
>  - **Value Learning (SCBC → SCIQL)**: SCIQL adds value functions to SCBC. This allows training on randomly sampled styles (via value generalization) rather than just future styles. The performance gain in Table 1 highlights the necessity of RL over pure IL for robust style alignment.
>  - **Gated Advantage Weighted Regression (GAWR)**: In Table 2, comparing SCIQL (lambda) which optimizes only style to SCIQL (lambda > r) isolates the specific contribution of the gating mechanism. This shows that GAWR significantly boosts task performance while maintaining style alignment.
>
> Additionally, we provide specific technical ablations in Appendix E: E.1 analyzes estimation strategies for the style distribution p(z|s,a), and E.2 isolates the impact of random relabeling versus current-style sampling.
>
> We acknowledge that our ablations are poorly referenced in the manuscript. In the revised manuscript we better highlight that part of our considered baselines are direct ablations of our method, and better point to appendix E.
>
> **Hyperparameter Sensitivity**  SCIQL inherits the stability of IQL, we utilized standard, fixed IQL hyperparameters for value learning across all tasks. Regarding style-conditioned task performance optimization, our GAWR method simplifies tuning compared to prior work like SORL, which requires careful balancing of the temperature parameter beta terms. As demonstrated in Table 2, the gating mechanism does not introduce sensitive hyperparameters but rather eliminates the need for complex weighting, allowing robust performance without extensive tuning.
>
> > The approach assumes accurate style labels from the labeling functions. How does performance degrade under noisy or imperfect style annotations, and can the method handle ambiguous or overlapping style categories?
>
> We thank **R-uFXs** for this valuable suggestion. In this regard, as reviewer **R-re3p** noted, we observe a substantial performance drop for the style-alignment baselines on halfcheetah-vary-v0. Since this dataset contains trajectories with varying styles, it includes heterogeneous behaviors and noisy transitions. Consequently, SCIQL responds more robustly to this type of noise, showing no performance drop unlike the baselines, which provides an indication of its superior robustness.
> Nevertheless, to examine this more thoroughly, we are conducting additional experiments to evaluate SCIQL’s performance under different levels of label noise. We will include a robustness analysis in the revised manuscript.
>
> [1]  Ross & al., A Reduction of Imitation Learning and Structured Prediction
> to No-Regret Online Learning, AISTATS 2011.
>
> [2] Hoque & al., LazyDAgger: Reducing Context Switching in Interactive Imitation Learning, IEEE CASE 2021.
>
> [3] Biré & al., Efficient Active Imitation Learning with Random Network Distillation, ICLR 2025.
>
> [4] Park & al., OGBench: Benchmarking Offline Goal-Conditioned RL, ICLR 2025.
>
> [5] Peng & al., Advantage-Weighted Regression: Simple and Scalable Off-Policy Reinforcement Learning, Arxiv 2019.

---

> ### Author Response · Authors · 2025-11-28
> **Answer to Reviewer uFXs (4)**
>
> ## On additional environments
>
> > Evidence on large-scale real-world or high-dimensional robotic systems remains limited. While the method demonstrates strong performance in simulated benchmarks, there is no evaluation on real-world systems or higher-dimensional robot control tasks.
>
> We are pleased to announce the addition of a new set of experiments on a higher-dimensional humanoid robot control task following the reviewers suggestions. We invite **R-uFXs** to refer the **Additional answer to all reviewers** section of the answers for more details about those experiments.
>
> ## On the algorithmic pipeline and ablations
>
> > The approach assumes accurate style labels from the labeling functions. How does performance degrade under noisy or imperfect style annotations, and can the method handle ambiguous or overlapping style categories?
>
> To answer **R-uFXs**’s question more thoroughly, we performed a set of experiments to test the evolution of the style alignment of SCIQL compared to each baseline on a range of noise levels $\zeta$ spanning from 0.0 (no noise, all labels are right), to 1.0 (all labels are uniformly sampled among wrong labels). The results of this ablation are discussed in Appendix E.3, with two key takeaways:
>
>  - **For noise labels ranging from 0.0 to 0.6**: SCIQL maintains a very good style alignment of above 60 on all tests. This is in contrast to the baselines which perform similarly in alignment to BC beyond small noise levels such as 0.2, meaning that they simply ignore the conditioning and lose all controllability. More precisely, in our tests, SCIQL with a noise level of 0.6 performs on average better than the other baselines with no noise. Hence, SCIQL is significantly more robust to noisy labels than any other baseline.
>  - **For noise labels ranging from 0.6 to 1.0**: SCIQL’s alignment drops significantly towards 0. In fact, this phenomenon is a good feature. Our intuition behind it is that above the 0.6 threshold, the right labels are no longer the majority label for their given transition. As such, SCIQL learns to avoid them completely, in contrast to the baselines which as before ignore label conditioning. We encourage the reviewers to refer to the update Appendix E.3 for further details.
>
> In conclusion, on top of being a significant improvement of robustness compared to previous baselines, SCIQL is very promising towards the use of alternative and more noisy labeling methods such as manual or VLM labeling, or settings where labeling is more difficult to obtain.

---

> > ### Author Response · Authors · 2025-12-03
> > **Answer to Reviewer uFXs (5)**
> >
> > > Evidence on large-scale real-world or high-dimensional robotic systems remains limited. While the method demonstrates strong performance in simulated benchmarks, there is no evaluation on real-world systems or higher-dimensional robot control tasks.
> >
> > On top of the aforementioned additional experiments, we are pleased to announce the addition of a new set of experiments on a more challenging version of our higher-dimensional humanoid robot control task. We invite **R-uFXs** to refer to the **Final answer to all reviewers** section of the answers for more details about those experiments.

---

### Author Response · Authors · 2025-11-21
**Answer to all reviews**

We thank the reviewers for their time and valuable feedback.

**Manuscript Updates (Updated on December 3 to match the revised manuscript)**: Following their suggestions, we performed the following modifications to the manuscript:
 - l.190: Following **R-re3p**’s recommendation, we added a reference to the environment description part of the appendix to help the reader better understand what “style” means.
 - l.350: Following **R-uFXs**’s remarks, we clarified that our training pipeline is contained in practice under one loop, highlighting the simplicity of our training procedure.
 - l.397: Following **R-uFXs**'s and **R-KRET**’s suggestions, we present two newly integrated higher-dimensional humanoid settings called Humenv-Simple and Humenv-Complex based from the HumEnv environment from [1].
 - l.410: Following **R-uFXs**’s remarks on the need to isolate components, we modified the paragraph introducing the baselines to highlight that BC, CBC, SCBC and SCIQL are built on top of each other, and consequently can serve as a basis for ablation studies.
 - l.445: Following **R-re3p**’s remark, we added a discussion about the drop in performance of the baselines for the halfcheetah-vary-v0 dataset, to which SCIQL appears to be robust. We also added a reference to a more complete discussion about the results in Appendix D.
 - l.460: Following **R-uFXs**'s and **R-KRET**’s suggestions, we present new style alignment results on the HumEnv settings.
 - l.476: Following **R-KRET**’s suggestions, we added further discussions on the tradeoffs between the style alignment and task performance, plotting the Pareto fronts and computing additional metrics such as Hypervolumes.
 - l.495: Following **R-uFXs**'s and **R-KRET**’s suggestions, we present new style-conditioned task performance optimization results on the HumEnv settings.
 - l.1169: Following **R-uFXs**'s and **R-KRET**’s suggestions, we detail the new HumEnv settings in terms of environments, tasks, datasets, criteria and labels.
 - l.1224:  Following **R-uFXs**'s and **R-KRET**’s suggestions, we added details about the hyperparameters used in the HumEnv settings.
 - l. 1470: Following **R-uFXs**'s and **R-KRET**’s suggestions, we added details about the HumEnv settings experiment complexity.
 - l.1486: Following **R-re3p**’s suggestion, we added a more in-depth explanation of the different performance across datasets.
 - l. 1536: Following **R-uFXs**'s and **R-KRET**’s suggestions, we added details about the HumEnv settings performances in the full style alignment table.
 - l.1542: Following **R-re3p**’s suggestion, we also discuss the impact of SORL’s temperature sensitivity issue and SCIQL’s robustness to such tuning.
 - l. 1610: Following **R-uFXs**'s and **R-KRET**’s suggestions, we added details about the HumEnv settings performances in the full style-conditioned task performance table.
 - l. 1728: Following **R-re3p**’s question, we added an ablation study to demonstrate SCIQL's higher robustness to labeling noise.
 - We also fixed some typos throughout the paper.

**Website (Updated on December 3 to match the updated website)**: Furthermore, we added to our website https://sciql-iclr-2026.github.io/:
 - Videos of various stylized rollouts of trained SCIQL agents to help visualize the diversity and alignment of our stylized policies.
 - Comparative videos about SORL and SCIQL style-conditioned task performance optimization performances.
 - A summary of the main contributions, designs and results of our paper.

**Additional Experiments (Updated on December 3 to take into account additional experiments)**: We added additional experiments on the new HumEnv settings (following **R-uFXs**'s and **R-KRET**'s suggestions) as well as ablations on the robustness to noisy labels (following **R-uFXs**'s suggestions), both illustrating the superiority of SCIQL.

[1] Tirinzoni, Touat & al., Zero-shot Whole-Body Humanoid Control via Behavioral Foundation Models, ICLR 2025.

---

> ### Author Response · Authors · 2025-11-28
> **Additional answer to all reviewers (1)**
>
> We thank the reviewers for their patience. To further answer the reviewers’s questions, we introduce several additions to complete our first answers.
>
> ## Additional Experiments on Noisy Labels
>
> To answer **R-uFXs** more thoroughly, we performed a set of experiments to test the evolution of the style alignment of SCIQL compared to each baseline under a range of labeling noise levels. We updated the manuscript by discussing the results in Appendix E.3, highlighting the significantly superior robustness of SCIQL compared to the baselines. We also provided a more detailed answer in the answers to **R-uFXs**.

---

> ### Author Response · Authors · 2025-11-28
> **Additional answer to all reviewers (2)**
>
> ## Additional Experiments on a Humanoid Environment
> **(Updated on December 3 to change the name humenv to humenv-simple.)**
>
> Following **R-uFXs**’s and **R-KRET**’s suggestions,  we performed a new set of experiments on a higher-dimensional humanoid in Mujoco obtained from the HumEnv environment [1]. This environment is built on the SMPL skeleton [2], which consists of 24 rigid bodies, among which 23 are actuated. This SMPL skeleton is widely used in character animation and is well suited for expressing natural human-like stylized behaviors. HumEnv’s observations consist in the concatenation of the body poses (70 D), body rotations (144 D) and angular velocities (144D) resulting in a 358-dimensional vector. It moves the body using a proportional derivative controller resulting in a 69-dimensional action space. **This task has consequently a higher dimensionality of (358, 69) compared to HalfCheetah’s (17, 6) dimensionality.**
>
> We generated a stylized dataset using the Metamotivo-M1 model provided in [1], generating various ways of moving at different heights and speeds, and focused our study on head height labels, while defining the RL tasks rewards as the norm of the body velocity. Since, the Metamotivo-M1 model was trained with a regularization towards the AMASS motion-capture dataset [3], **it provides more natural and human-like stylized behaviors.**
>
> We display below the result table of each baseline on style alignment. We see that both SCBC and SCIQL perform near perfect style alignment on both head_height labels, while other methods struggle. BC, without conditioning, is more aligned towards label 0, which corresponds to low head positions, while sometimes reaching higher head positions. This is natural given that failure cases result in the falling of the humanoid. CBC, BCPMI and SORL are well aligned with label 0, which is easier to attain, but struggle to align with label 1 which corresponds to harder standing positions. Those results **highlight the importance of stitching and style relabeling when optimizing for style alignment**.
>
> | Dataset                        | BC          | CBC            | BCPMI          | SORL           | SCBC           | SCIQL          |
> | ------------------------------ | ----------- | -------------- | -------------- | -------------- | -------------- | -------------- |
> | humenv-simple - head_height - label 0 | 79.4 ± 44.4 | **99.2 ± 0.2** | **99.0 ± 0.4** | **99.3 ± 0.1** | **99.3 ± 0.0** | **99.3 ± 0.1** |
> | humenv-simple - head_height - label 1                               |  20.6 ± 44.4           | 79.0 ± 43.7               |   59.4 ± 52.             |    59.6 ± 53.7            |           **100.0 ± 0.0**     |     **100.0 ± 0.0**           |
> | humenv-simple - head_height - average | 50.0 ± 44.4 | 89.1 ± 22.0 | 79.2 ± 26.7 | 79.4 ± 26.9 | **99.6 ± 0.0** | **99.6 ± 0.0** |
>
> Additionally, we display the style conditioned task performance optimization table below. We can see that all methods preserve their style alignment while taking into account reward signals. However, while SORL (β=0) struggles to maintain standing positions with label information, the presence of speed reward signals, highly correlated to the standing positions required by the sprint, helps SORL (β>0) align to label 1. **SCIQL however presents near perfect alignment in any configurations. Overall, SCIQL obtains better task performance than any SORL variant while preserving its near perfect alignment.**
>
> | Dataset | Metric | SORL (β=0) | SORL (β=1) | SORL (β=3) | SCIQL (λ) | SCIQL (λ > r) | SCIQL (r > λ) |
> | :--- | :--- | :--- | :--- | :--- | :--- | :--- | :--- |
> | humenv-simple - head_height - label 0 | Style | 99.3 ± 0.1 | 99.2 ± 0.1 | 99.2 ± 0.1 | 99.3 ± 0.1 | 99.3 ± 0.1 | 99.3 ± 0.1 |
> | | Task | 13.2 ± 14.6 | 5.7 ± 4.2 | 13.4 ± 13.5 | 19.8 ± 14.0 | 33.6 ± 0.2 | 34.2 ± 0.3 |
> | humenv-simple - head_height - label 1 | Style | 59.6 ± 53.7 | 99.0 ± 1.6 | 99.7 ± 0.8 | 99.9 ± 0.1 | 100.0 ± 0.0 | 99.7 ± 0.3 |
> | | Task | 16.0 ± 14.5 | 26.4 ± 10.8 | 26.5 ± 11.5 | 18.5 ± 0.3 | 29.8 ± 9.5 | 38.8 ± 0.5 |
> | humenv-simple - head_height | Style | 79.4 ± 26.9 | 99.1 ± 0.9 | 99.4 ± 0.4 | 99.6 ± 0.0 | 99.6 ± 0.1 | 99.5 ± 0.2 |
> | | Task | 14.6 ± 14.5 | 16.0 ± 7.5 | 20.0 ± 12.5 | 19.1 ± 7.1 | 31.7 ± 4.8 | 36.5 ± 0.4 |
>
> Those results **demonstrate the capabilities of our method to scale in higher dimensional settings**. Also, we invite the reviewers to visit our updated anonymous website https://sciql-iclr-2026.github.io/ to visualize SCIQL (λ)’s rollouts on HumEnv, as well as a comparison of the style conditioned task performance optimization of both SORL (β=0), SORL (β=3), SCIQL (λ) and SCIQL (r > λ).
>
> [1] Tirinzoni, Touat & al., Zero-shot Whole-Body Humanoid Control via Behavioral Foundation Models, ICLR 2025.
>
> [2] Loper & al., SMPL: a skinned multi-person linear model, ACM Transactions on Graphics 2015.
>
> [3] Mahmood & al., AMASS: archive of motion capture as surface shapes, ICCV 2019.

---

> ### Author Response · Authors · 2025-12-03
> **Final answer to all reviewers (1)**
>
> As the discussion period comes to a close, we would like to take this opportunity to first thank the reviewers for their review, for which we have done our best to address the points. We add bellow details about the final additional experiments we performed.
>
> ## Follow up on additional experiments
> Following **R-uFXs**’s and **R-KRET**’s suggestions, we further enriched our experiments to strengthen the work.
>
> The initial provided **HumEnv experiments** were performed under one **head_height** criteria of two labels, corresponding to **low** and **high head positions** and for an **initial standing position**, resulting in an equal or superior performance of our method compared to the baselines.
>
> However, to further demonstrate its benefits, we tested our method against the baselines on a more complex HumEnv setup where we:
>  1) Changed the initialization from a **standing position** to a **lying down position**. In this new position, reaching high head positions, but also high speeds, would require more planning as lying down positions are not aligned with such styles. This setup would consequently better highlight the benefits of style relabeling.
> 2) We added a new **speed** criteria of 3 labels, for **immobile**, **slow** and **fast** respectively, and complexified the **head_height** criteria by adding a new label for a total of 3 labels, **low**, **medium** and **high** respectively. The **speed_label** has a particularity of being strongly correlated with the reward, allowing for testing the efficiency of our gating mechanism to stay with a certain style despite task optimization. Also, the finer head height labels allow for testing a more precise control of our policies regarding the head position.
> 3) We also modified the data generation process by changing styles along the trajectory, similarly to the halfcheetah-vary datasets.
>
> We call the resulting environment **HumEnv-Complex**, and consequently renamed the simpler version **Humenv-Simple**.
>
> We display below the result table of each baseline on style alignment. We see that contrary to the humenv-simple experiments, SCIQL displays a **significantly higher overall performance than all baselines**. In particular, SCBC’s performance largely decreased, equalling BC’s performance, in a similar fashion as in halfcheetah-vary. We can explain this decrease by the fact that SCBC relabels styles by uniformly sampling them in the future trajectory, considering the current action as a good action to reach any future label. However, when styles change often along the trajectory, this strategy leads to the learning of wrong actions overall. SCIQL on the other hand performs advantage weighting to solve this problem. SCIQL also largely outperforms every other baseline on all style criteria. This **highlights the benefits of style relabeling in association to value learning** to allow for **efficient planning towards style aligned state-actions pairs**.
>
> | Dataset | BC | CBC | BCPMI | SORL | SCBC | SCIQL |
> | :--- | :--- | :--- | :--- | :--- | :--- | :--- |
> | humenv-complex - head_height - label 0 | 0.4 ± 0.0 | 93.0 ± 1.9 | 80.7 ± 18.3 | 90.8 ± 4.0 | 0.5 ± 0.0 | **99.9 ± 0.2** |
> | humenv-complex - head_height - label 1 | **97.8 ± 4.0** | 53.8 ± 22.6 | 47.2 ± 32.6 | 69.0 ± 13.8 | **96.9 ± 4.0** | **93.3 ± 4.6** |
> | humenv-complex - head_height - label 2 | 1.8 ± 4.0 | 37.9 ± 31.0 | 43.3 ± 19.0 | 23.5 ± 9.9 | 0.0 ± 0.0 | **56.6 ± 15.0** |
> | humenv-complex - head_height - all | 33.3 ± 2.7 | 61.6 ± 18.5 | 57.1 ± 23.3 | 61.1 ± 9.2 | 32.4 ± 1.3 | **83.3 ± 6.6** |
> | humenv-complex - speed - label 0 | 22.8 ± 7.8 | 17.9 ± 6.4 | 13.3 ± 17.2 | 6.7 ± 3.7 | 16.1 ± 9.7 | **96.3 ± 1.9** |
> | humenv-complex - speed - label 1 | 77.2 ± 7.8 | 76.7 ± 10.1 | 82.1 ± 21.8 | **93.2 ± 3.8** | 86.2 ± 7.8 | **91.8 ± 5.4** |
> | humenv-complex - speed - label 2 | 0.0 ± 0.0 | 3.1 ± 4.7 | 0.8 ± 1.8 | 3.0 ± 6.7 | 0.0 ± 0.0 | **62.9 ± 10.5** |
> | humenv-complex - speed - all | 33.3 ± 5.2 | 32.6 ± 7.1 | 32.1 ± 13.6 | 34.3 ± 4.7 | 34.1 ± 5.8 | **83.7 ± 5.9** |

---

> ### Author Response · Authors · 2025-12-03
> **Final answer to all reviewers (2)**
>
> Additionally, we display the style conditioned task performance optimization table below. We can see that incorporating task optimization (i.e. going from SORL (β=0) to SORL (β>0)) results in a decrease of the average style alignment, while it increases it for SCIQL (i.e. going from SCIQL (λ) and SCIQL (λ >r)). We can explain this by the fact that SCIQL can exploit beneficial rewards signals to perform better style alignment when possible. Here, it is for instance the case for the high head_height label, head_height - label 2, which is highly correlated to a running position and as such to a high reward. Consequently, not only can SCIQL **maintain style alignment when optimizing for task performance by filtering non-aligned actions**, it can also **benefit from rewards signals to perform better alignment overall.**
>
> | Dataset | Metric | SORL (β=0) | SORL (β=1) | SORL (β=3) | SCIQL (λ) | SCIQL (λ > r) | SCIQL (r > λ) |
> | :--- | :--- | :--- | :--- | :--- | :--- | :--- | :--- |
> | humenv-complex - head_height - label 0 | Style | 90.8 ± 4.0 | 42.3 ± 24.4 | 56.1 ± 26.4 | 99.9 ± 0.2 | 100.0 ± 0.0 | 7.3 ± 7.2 |
> | | Task | 2.6 ± 2.0 | 26.9 ± 10.4 | 19.9 ± 12.5 | 0.1 ± 0.0 | 0.1 ± 0.0 | 42.2 ± 3.4 |
> | humenv-complex - head_height - label 1 | Style | 69.0 ± 13.8 | 14.0 ± 10.4 | 3.2 ± 1.4 | 93.3 ± 4.6 | 82.9 ± 16.3 | 1.7 ± 0.1 |
> | | Task | 5.9 ± 5.4 | 25.2 ± 2.9 | 33.8 ± 2.8 | 4.2 ± 1.8 | 6.6 ± 2.6 | 41.3 ± 4.9 |
> | humenv-complex - head_height - label 2 | Style | 23.5 ± 9.9 | 9.7 ± 6.1 | 13.5 ± 28.7 | 56.6 ± 15.0 | 87.3 ± 11.5 | 90.9 ± 7.32 |
> | | Task | 5.1 ± 4.0 | 6.7 ± 2.6 | 7.7 ± 12.7 | 25.6 ± 6.4 | 40.5 ± 5.1 | 42.3 ± 3.2 |
> | humenv-complex - head_height - all | Style | 61.1 ± 9.2 | 22.0 ± 13.6 | 24.3 ± 18.9 | 83.3 ± 6.6 | 90.1 ± 9.3 | 33.3 ± 4.9 |
> | | Task | 4.5 ± 3.8 | 19.6 ± 5.3 | 20.5 ± 9.3 | 10.0 ± 2.8 | 15.7 ± 2.6 | 41.9 ± 3.8 |
> | humenv-complex - speed - label 0 | Style | 6.7 ± 3.7 | 1.8 ± 3.7 | 13.5 ± 11.1 | 96.3 ± 1.9 | 99.5 ± 0.0 | 6.6 ± 3.0 |
> | | Task | 5.0 ± 0.2 | 42.8 ± 2.3 | 37.7 ± 5.9 | 0.4 ± 0.1 | 0.1 ± 0.0 | 40.6 ± 1.7 |
> | humenv-complex - speed - label 1 | Style | 93.2 ± 3.8 | 4.5 ± 2.5 | 3.9 ± 2.3 | 91.8 ± 5.4 | 86.9 ± 18.0 | 7.2 ± 4.3 |
> | | Task | 5.7 ± 1.6 | 38.8 ± 5.1 | 38.4 ± 10.5 | 6.1 ± 0.7 | 7.2 ± 3.2 | 39.0 ± 4.5 |
> | humenv-complex - speed - label 2 | Style | 3.0 ± 6.7 | 80.0 ± 18.9 | 50.4 ± 19.9 | 62.9 ± 10.5 | 88.5 ± 8.6 | 86.1 ± 3.6 |
> | | Task | 6.4 ± 3.1 | 37.3 ± 8.1 | 24.8 ± 8.3 | 29.5 ± 4.1 | 41.3 ± 3.9 | 40.4 ± 1.7 |
> | humenv-complex - speed - all | Style | 34.3 ± 4.7 | 28.8 ± 8.4 | 22.6 ± 11.1 | 83.7 ± 5.9 | 91.6 ± 8.9 | 33.3 ± 3.7 |
> | | Task | 5.7 ± 1.6 | 39.7 ± 5.2 | 33.6 ± 8.2 | 12.0 ± 1.6 | 16.2 ± 2.3 | 40.0 ± 2.6 |
>
> Those results again demonstrate the **capabilities of our method to scale in harder and higher dimensional settings** which require the **association of both style relabeling and value learning.**
>
> We reported those additional results in the dedicated sections of our updated manuscript.

---

### Author Response · Authors · 2025-12-03
**Summary of Discussion for the Area Chair**

Dear Area Chair,

As the discussion period comes to a close, we would like to thank you for overseeing this submission. To assist in your assessment, we have summarized below the principal points of the discussion. We leveraged the reviewers' helpful feedback to significantly strengthen the paper, which we updated. In particular, the main reviewers' concerns and associated updates were:

1. **Adding a high-dimensional humanoid environment (R-uFXs, R-KRET):** We generated two stylized locomotion datasets using the Metamotivo-M1 policy in the HumEnv environment from [1], based on the widely used SMPL humanoid skeleton [2] (358D observation, 69D action). On this challenging testbed, **SCIQL significantly outperformed all studied baselines and ablations** both on pure style alignment (Table 1) and on **style-conditioned task performance** (Table 2).

2. **Pareto optimality visualization and positioning (R-KRET):** We clarified that our problem formulation differs from symmetric Multi-Objective RL, as we optimize task performance under style constraints (details in **Answer to Reviewer KRET (1)**). Nevertheless, to better visualize this trade-off, we added Pareto front plots (Figure 2). These results demonstrate that **SCIQL’s frontier strictly dominates baselines across all environments** (including our added humanoid experiments), yielding hypervolume improvements ranging from **+41% to +163%.**

3. **Need for ablations & sensitivity (R-uFXs):** We updated the manuscript to mention explicitly that **our comparative analysis includes an additive ablation study (section 5.1.2)**, isolating the specific contributions of **style conditioning (BC → CBC), hindsight relabeling (CBC → SCBC), and value learning (SCBC → SCIQL).** We further clarified that our **Gated Advantage mechanism (GAWR) eliminates the need for sensitive temperature tuning** required by prior work (SORL [3]), acting instead as a robust mechanism to balance style and task objectives.

 4. **Presentation:** Based on all reviewers’ suggestions, we extensively revised the manuscript to improve clarity (see post **Answer to all reviewers** for an exhaustive list of manuscript updates). Finally, we launched a companion website featuring videos of our trained SCIQL agents to better visualize our results: https://sciql-iclr-2026.github.io/.  In particular, we provide a visualization of learned humanoid policies using our SCIQL vs the SOTA baseline SORL, showcasing the superiority of our approach in video.

**Conclusion:** We believe these updates comprehensively address the reviewers' concerns regarding scalability and experimental scope, confirming SCIQL as a robust and high-performing framework for stylized offline RL. We hope these additions further support the acceptance of our work.

[1] Tirinzoni, Touat & al., Zero-shot Whole-Body Humanoid Control via Behavioral Foundation Models, ICLR 2025.

[2] Loper & al., SMPL: a skinned multi-person linear model, ACM Transactions on Graphics 2015.

[3] Mao & al., Stylized Offline Reinforcement Learning: Extracting Diverse High-Quality Behaviors from Heterogeneous Datasets, ICLR 2024.

---

### Meta-Review · Area_Chair_WPMW · 2026-01-03

**Summary:**

In this paper, the authors considered style-conditioned policy learning in offline setting. By style-conditional task learning, the authors meant to optimize the policy (maximizing the task return) while following a certain behavior style. For this the authors considered a set of discrete styles induced by the criterion function lambda, and then defined a "style-alignment reward" in Eq. (3). The authors then derived a policy optimization criterion, referred to as Gated AWR, based on IQL both on the task advantage estimation and the style advantage estimation. Finally, the authors shows performance gains of their method.

**Reviewer Concerns:**

In addtion to the minor concerns such as hyperparameter sensitivity, implementation complexity, applicability to more realistic environments, etc, the major concerns are as follows:

- As mentioned by Reviewer KRET, the policy optimization formulation in equations (12) and (13) is questionable when considering the confliction between the task performance and the style performance. As mentioned, a more meaning formulation can be a multi-objective RL formulation.

- (AC comment) More importantly, there seems a technical fallacy in the formulation. Eq. (2) should be the true style alignment metric, and eq. (3) can be a good substitute. Note here that in eq. (3), one should use p_\pi^\lambda (z| s_t,a_t), which is the probability that conditioned on s_t, a_t, the policy \pi under optimization  generates the style (discrete variable) z. The discounted sum of these probabilities forms S^p(pi, lambda, z), a style aligment metric. We are optimizing policy pi so that it generates the target style z. But, in line 216 and line 311, the authors replaced p_pi^lambda (z|s,a) with p_piD^lambda (z|s,a), where piD is a sampling distribution. Note that this sampling distribution is not the policy under optimization. So, the final style alignment metric is not one about the policy under optimization. It is undertandable that computing p_\pi^\lambda (z| s_t,a_t) for z in trajectory is not easy, but when replaced, it is solving a different problem. The authors could consider how to correct this issue. For example, the importance sampling ratio -based modification could be useful, although it is not certain. Or some other techniques might be used.

- The writing should be improved significantly. Currently, the contents of the paper do not smoothly flow.

- It is recommended that the authors further improve the paper by incorporating the reviewers and AC's comments and submit to a future venue.

**Reviewer Scores:**

The original scores are 6, 6, 2. It is not likely that Reviewer KRET of score 2 would increase the score.

---

### Decision · Program_Chairs · 2026-01-26

Reject